# A comprehensive sensitivity and uncertainty analysis for discharge and nitrate-nitrogen loads involving multiple discrete model inputs under future changing conditions

Christoph Schürz[1], Brigitta Hollosi[2], Christoph Matulla[2], Alexander Pressl[3], Thomas Ertl[3], Karsten Schulz[1], and Bano Mehdi[1,4]

[1]Institute for Hydrology and Water Management, University of Natural Resources and Life Sciences, Vienna (BOKU), Vienna, Austria
[2]Department of Climate Research, Zentralanstalt für Meteorologie und Geodynamik (ZAMG), Vienna, Austria
[3]Institute of Sanitary Engineering and Water Pollution Control, University of Natural Resources and Life Sciences, Vienna (BOKU), Vienna, Austria
[4]Division of Agronomy, Department of Crop Sciences, University of Natural Resources and Life Sciences, Vienna (BOKU), Tulln, Austria

**Correspondence:** Christoph Schürz (christoph.schuerz@boku.ac.at)

**Abstract.** Environmental modeling studies aim to infer the impacts on environmental variables that are caused by natural and human-induced changes in environmental systems. Changes in environmental systems are typically implemented as discrete scenarios in environmental models to simulate environmental variables under changing conditions. The scenario development of a model input usually involves several data sources and perhaps other models, that are potential sources of uncertainty. The setup and the parametrization of the implemented environmental model are additional sources of uncertainty for the simulation of environmental variables. Yet, to draw well-informed conclusions from the model simulations it is essential to identify the dominant sources of uncertainty.

In impact studies in two Austrian catchments the eco-hydrological model Soil and Water Assessment Tool (SWAT) was applied to simulate discharge and nitrate-nitrogen ($NO_3^-$-N) loads under future changing conditions. For both catchments the SWAT model was set up with different spatial aggregations. Non-unique model parameter sets were identified that adequately reproduced observations of discharge and $NO_3^-$-N loads. We developed scenarios of future changes for land use, point source emissions, and climate and implemented the scenario realizations in the different SWAT model setups with different model parametrizations, which resulted in 7000 combinations of scenarios and model setups for both catchments. With all model combinations we simulated daily discharge and $NO_3^-$-N loads at the catchment outlets.

The analysis of the 7000 generated model combinations of both case studies had two main goals; i) to identify the dominant controls on the simulation of discharge and $NO_3^-$-N loads in the two case studies, and ii) to assess how the considered inputs control the simulation of discharge and $NO_3^-$-N loads. To assess the impact of the input scenarios, the model setup and the parametrization on the simulation of discharge and $NO_3^-$-N loads we employed methods of global sensitivity analysis (GSA). The uncertainties in the simulation of discharge and $NO_3^-$-N loads that resulted from the 7000 SWAT model combinations

were evaluated visually. We present approaches for the visualization of the simulation uncertainties that support the diagnosis of how the analyzed inputs affected the simulation of discharge and $NO_3^-$-N loads.

Based on the GSA we identified climate change and the model parametrization to be the most influential model inputs for the simulation of discharge and $NO_3^-$-N loads in both case studies. In contrast, the impact of the model setup on the simulation of discharge and $NO_3^-$-N loads was low and the changes in land use and point source emissions were found to have the lowest impact on the simulated discharge and $NO_3^-$-N loads. The visual analysis of the uncertainty bands illustrated that the deviations in precipitation of the different climate scenarios to historic records dominated the changes in simulation outputs, while the differences in air temperature showed no considerable impact.

## 1 Introduction

Environmental systems are under constant change. Predicting the development of natural resources in a changing system involves large uncertainties (Milly et al., 2008). Climate change, in concurrence with other dynamic processes such as population growth, land use change or economic development pose challenges to the management of water supply and water quality (Duran-Encalada et al., 2017; Yates et al., 2015). Human disturbances can exacerbate the impacts of climate and amplify consequences to water quality (Jiménez et al., 2014) on one hand. On the other hand, stakeholders in environmental systems have to respond to future changes, for instance adapting farm management practices due to changes in temperatures and precipitation patterns (Schönhart et al., 2018). Ideally, an impact assessment considers all future changes that can affect the development of the environment of interest as well as those future changes that can introduce uncertainties in the simulation of the environmental variables of interest.

Changes in environmental systems are typically represented by discrete scenarios in impact studies. Preferably, the set of scenarios representing a dynamic change covers the full range of trajectories along which the development is plausible (Clark et al., 2016). Scenario development involves different data sources and models, which can introduce and propagate uncertainties. For example, climate scenarios have several sources of uncertainty and may include several socioeconomic scenarios (e.g. the current "Representative Concentration Pathways" (RCP; Moss et al., 2010; van Vuuren et al., 2011)) that drive an array of global climate models (GCM) (Knutti and Sedláček, 2013). However, the GCMs also have inherent uncertainty and they provide the boundary conditions for regional climate models (RCM) (e.g. Jacob et al., 2014). Further, the downscaling (Wilby et al., 1998; Wood et al., 2004) of the RCM simulations and the bias correction (Teutschbein and Seibert, 2013, 2012) are associated with their own uncertainty and are a standard procedures in climate scenario development. Eventually, it is essential to characterize the uncertainties inherent in all processes that affect the simulation of an environmental variable.

To simulate the development of hydrological variables under changing conditions, the developed scenarios are implemented as boundary conditions in hydrological models that are calibrated for historic observations. Yet, often different model setups and different sets of parameters in a model can perform equally well to reproduce historical observations of the variables of interest. Equifinality is a well-known issue in hydrologic modeling that has been extensively addressed in the literature (e.g. Schulz et al., 1999; Beven, 2006; Beven and Freer, 2001; Beven, 1996), where multiple model structures (e.g. Clark et al.,

2008) and model parametrizations (e.g. Schulz et al., 1999) represent observations equally well and thus cannot be rejected (Beven, 2006). An adequate representation of historical data does not necessarily assure that different model setups agree when extrapolating to future conditions (Chiew and Vaze, 2015; Milly et al., 2008). Thus, differences in the model setup are a source of uncertainty in the simulation of an environmental variable under future conditions.

Altogether, an impact study comprises an abundance of combinations of trajectories of system changes and model setups to describe an environmental system that ultimately characterize the uncertainties in a simulation. Hence, a comprehensive description of the uncertainties in model simulations is a major challenge of any impact study.

Model sensitivity analysis (SA) can be used to derive the impact of different input variables on hydrological target variables. SA investigates the response of a modeled variable to the variation of model input variables (Saltelli et al., 2004). For a local sensitivity analysis (LSA) the model inputs are varied around a point (often an 'optimum' point) in the model input space. Global sensitivity analysis (GSA) assesses the sensitivity of a model output for the entire feasible range of model inputs (Gupta and Razavi, 2017; Pianosi et al., 2016). Compared to LSA, GSA usually requires a larger number of computations. Thus, a substantial part of recent GSA literature focuses on the computational efficiency and the robustness of GSA methods (e.g. Pianosi and Wagener, 2015; Razavi and Gupta, 2016a; Sarrazin et al., 2016; Cuntz et al., 2015; Rakovec et al., 2014), but also on increasing the insight into modeled systems from a certain number of model evaluations (e.g. Borgonovo et al., 2017; Dai et al., 2017; Guse et al., 2016a; Massmann et al., 2014; Razavi and Gupta, 2016a).

The complexity and computational demand of a model determine the feasible number of model evaluations and thereby the applicability of a SA method (Razavi and Gupta, 2015). Large atmospheric model applications, for instance, only allow a LSA with a few model evaluations (Gupta and Razavi, 2017; Pianosi et al., 2016). Environmental model applications are usually less computationally expensive and allow a more extensive GSA, illustrated in many environmental modeling studies (e.g. Guse et al., 2016b; Haghnegahdar et al., 2017; Massmann and Holzmann, 2015; Razavi and Gupta, 2016b; Sarrazin et al., 2016). Most applications utilize GSA to identify influential model parameters and to rank model parameters according to their influence on model outputs. Model parameters are usually continuous model inputs. (Saltelli et al., 2008; Baroni and Tarantola, 2014).

Although it is possible to implement composite model inputs (e.g. climate scenarios that affect several climate variables at the same time, or land use scenarios that can impact the entire model setup) in a GSA and therefore to employ GSA in impact studies, a consideration of discrete and composite model inputs can constrain the applicability of GSA and complicate the implementation (Baroni and Tarantola, 2014). In impact studies, the response of an environmental variable to a (future) change in a model input is usually inferred by implementing a scenario realization of the respective model input in a model setup. From an SA perspective, this approach is equivalent to a local 'one-at-a-time' (OAT) assessment of the model input sensitivity (Saltelli and Annoni, 2010; Baroni and Tarantola, 2014). A local OAT analysis however presumes linear models and non-correlated inputs which are hardly true for any environmental model application (Rosolem et al., 2012; Baroni and Tarantola, 2014). Thus, to account for interactions of model inputs and model non-linearities the application of GSA is recommended instead (Saltelli and Annoni, 2010; Saltelli and Tarantola, 2002; Baroni and Tarantola, 2014).

Yet, a few studies implemented discrete and composite model inputs in GSA. With the Generalized Probabilistic Framework, Baroni and Tarantola (2014) rendered a solid basis for the implementation of correlated, non-continuous model inputs in GSA and applied the variance-based SA method of Sobol (1993) to assess the response of soil moisture, evapotranspiration, and soil water fluxes to uncertainties in meteorological input data, crop parameters, soil properties, model structure, and observation data. In a synthetic example, Dai and Ye (2015) performed model and scenario averaging to assess the impact of different model structures and scenarios of precipitation on groundwater flow and reactive transport in the soil. In a more recent study, Dai et al. (2017) employed the method of Sobol to identify the relevant system processes for groundwater flow and reactive transport represented in different model structures. Savage et al. (2016) applied GSA to identify the dominant controls in the calculation of flood inundation, to assess whether a high spatial resolution of the flood inundation model or the model parametrization is dominating the simulation. The mentioned studies illustrate the use of GSA with discrete and composite model inputs. Anderson et al. (2014) and Butler et al. (2014) highlight the importance of assessing the uncertainty of future climate change impacts and the identification of relevant drivers and their interactions for climate policy making.

In this paper we demonstrate the utility of GSA and uncertainty analysis in a comprehensive setting of an environmental model impact study and address the following points:

– We apply GSA in two environmental modeling impact studies to identify the dominant sources of uncertainties for the simulation of discharge and nitrate-nitrogen ($NO_3^-$-N) loads. We analyze the impacts of different spatial aggregations of the model setup and different model parametrizations and assess the effects of changes in the land use, point source emissions, and the future climate.

– We analyze the resulting uncertainties in the simulation of the long-term monthly mean discharge and monthly sums of $NO_3^-$-N loads, as well as flow duration curves (FDCs) of daily discharge and daily $NO_3^-$-N loads visually. We present ways to visualize the discrete model inputs that provide further insights into the relationships of uncertainties in the simulations and different properties of the discrete realizations of the model inputs.

– Based on the GSA and the visual analysis of the simulated uncertainties we are able to draw conclusion on the simulation of discharge and $NO_3^-$-N loads as impacted by the model setup, model parametrization and the future scenarios of land use, point source emissions and climate. These conclusions are of course limited to assumptions made in the model setup and in the development of the scenarios.

The paper is structured in the following way: Section 2 contains an overview of the two investigated catchments, the Soil and Water Assessment Tool (SWAT, Arnold et al., 1998) that we implemented in this study, and the preparation of the model input data that we used in the model setup. In Section 2.4 we describe the setup of the SWAT model with different spatial aggregations and illustrate the pre-processing of the SWAT model setups that was necessary to identify the sensitive SWAT model parameters and to define non-unique parameter sets for all model setups. The scenarios of land use, point source emissions and the climate together with the input data and pre-processing to develop the individual scenarios are specified in Section 2.5. Section 2.6 combines the SWAT model setups, the defined non-unique model parametrizations and the developed scenarios of land use,

point source emissions and climate in the GSA and explains the methods we applied to analyze the sources of uncertainties for the simulation of discharge and $NO_3^-$-N loads. The results of the combined GSA framework and the visual analysis are provided in Section 3. We discuss the findings of the GSA application and the visual analysis of the simulation uncertainties for the two case studies in Section 4 and address the specific assumptions that we made during the model setup and the development of the scenarios.

## 2   Materials and Methods

### 2.1   Study sites

The two investigated catchments (Schwechat and Raab) are representative examples for river systems for the eastern region of Austria. Both rivers have their origin in the forested foothills of the limestone Alps with a pre-alpine character and a low anthropogenic impact. The lower parts of both catchments are characterized by human activities, with primarily urban settlements and agricultural uses in the plains of the Schwechat catchment and dominant industrial activities and agricultural land uses in the valley bottom of the Raab catchment (Fig. 1 and Tables A3 and A4).

The Schwechat river has its source in the Vienna woods at the northeastern boundary of the Northern Limestone Alps with a maximum altitude of 893 m a.s.l. After a natural flow section in the narrow and dominantly forested valley of the "Helenental" (70% of the total catchment area. See Table A3), the Schwechat drains into the Vienna basin with flat topography and a predominance of agriculture, viniculture and settlement areas. The main agricultural crops are winter wheat and summer wheat. Larger areas in the upper part of the catchment are used as pastures (~10% of the total area). The largest settlement is the city of Baden with a population of approximately 26000 inhabitants, while smaller settlements are scattered over the catchment. All municipal wastewaters are collected in three wastewater treatment plants (WWTP, black triangles in Fig. 1), where the the WWTP Baden is the most relevant one with a capacity of 45000 population equivalents (PE). All WWTPs perform carbon removal, nitrification, denitrification and enhanced phosphorus removal. Due to the close proximity to the city of Vienna population growth is a likely prospect for the settlement areas in the lower part of the catchment. The part of the catchment considered in this study has its outlet next to the city of Traiskirchen at an altitude of 185 m a.s.l. and covers an area of approximately 275 km². The long term mean annual precipitation in the Vienna Basin is around 620 mm/yr and the mean annual temperature is 9.9°C.

The Raab river originates at the edge of the southeastern Alps. These are characterized by low mountain ranges with a maximum altitude of 1547 m a.s.l., mostly covered by forests (~42% of the total catchment area. See Table A4). The Raab flows through the southern part of Austria and crosses the boarder to Hungary close to the city of Neumarkt an der Raab at an altitude of 232 m a.s.l. The case study encompasses the Austrian part of the Raab with a catchment area of approximately 998 km². The long-stretched river valley is dominated by agricultural activities (~25 % of the total area), with urban areas in between. The slopes along the Raab are covered with heterogeneous patterns of forests, pasture areas and agricultural land use. The main agricultural crops are corn and oil seed pumpkins, but also wheat and vegetable production are common. While the urban areas are of similar small structure as in the Schwechat catchment, leather industries are present in the catchment

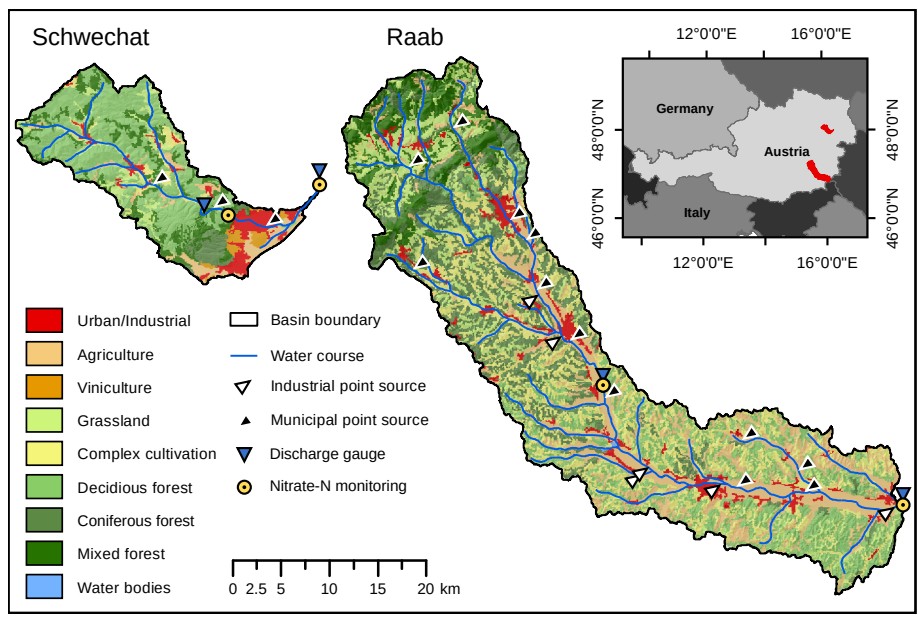

**Figure 1.** Study sites Schwechat (left) and Raab (right).

that release substantial nutrient inputs into the receiving waters, which has resulted in trans-boundary conflicts (Ruzicka et al., 2009). Municipal wastewaters in the Raab catchment are collected in 12 relevant WWTPs (black triangles in Fig. 1) that all have the same standards for wastewater treatment as in the Schwechat catchment, but have almost three times the total capacity (approximately 150000 PE). Six relevant industrial emitters are located along the main reach of the Raab river (white

triangles in Fig. 1) that all perform internal waste water treatment following the respective industry-specific regulations for wastewater treatment (e.g., BGBl. II Nr. 10/1999, 1999; BGBl. II Nr. 12/1999, 1999). The average annual precipitation in the Raab catchment is approximately 800 mm/yr and the long term annual mean temperature is 9.0°C.

## 2.2 The Soil and Water Assessment Tool (SWAT)

The SWAT model (Arnold et al., 1998) is a continuous, process based, semi-distributed eco-hydrological model. In this study

we implemented SWAT2012 (Rev.622) to simulate daily time series of discharge and $NO_3^-$-N loads at the catchment outlets. The models' spatial reference to a catchment is given by a subdivision of the basin into subbasins. Areas containing the same land use, soil type and lying in the same slope range are lumped together in each subbasin to form hydrologic response units (HRUs). All processes on the land phase of each subbasin are calculated at the HRU scale and are further propagated into the water phase of each subbasin. The processes calculated on the land phase include water balance components such

as interception, infiltration, shallow and deep percolation, surface runoff, lateral flow, groundwater flow, plant uptake and evapotranspiration, or the pathways of nutrients such as the input through atmospheric deposition, or fertilizer application, the transformation into other forms of a nutrient and the transport through surface runoff, percolation, lateral flow and return flow

in the groundwater (Neitsch et al., 2011). In the water phase, the nutrients budgets are calculated. Following the calculation of the water balance and the nutrient budgets, the discharge, the nutrient loads and other substances are routed through the linked subbasins to the defined catchment outlet (Neitsch et al., 2011). The required input data to set up a model with SWAT are a digital elevation model (DEM), a raster land use map including the model parametrization and the performed management operations for each land use, a raster soil map with soil physical and chemical parameters for all soil layers, and meteorological input data.

## 2.3 Model input data and data preparation

A DEM with a 10 m resolution was available for Austria from an airborne laser scan (Geoland.at, 2015). Based on the DEM we defined three slope classes with slopes of 0-3%, 3-8%, and >8% in the HRU definition step.

CORINE land cover (EEA, 2015) served as the base land use map to which more detailed agricultural data was added. CORINE does not classify agricultural land uses into crop types. Therefore, tabular data of agricultural land uses at the municipal level derived from the 2010 Austrian agronomic census (Statistik Austria, 2015b) was superimposed onto CORINE data by randomly distributing crops according to the crops' areal share at the municipal level to CORINE pixels containing agricultural and complex cultivation land use. Typical time windows for planting, fertilizer application, tillage and harvest were derived from field experiment records for the individual crops (Land NÖ, 2015) and written to the HRU management files. The management dates were randomized for all HRUs within the time windows derived for a management operation. Dates with strong rainfall or a high soil moisture potential were not used for scheduling management operations. With 70.0% and 42.3% forest land uses were the most dominant land uses in the Schwechat and the Raab catchments, respectively. The SWAT model setups differentiated between deciduous forests, coniferous forests and mixed forests, derived from CORINE land cover (see Tables A3 and A4). All HRUs with one of the three forest types as land use were parameterized with an initial biomass and an initial leave area index to simulate intact forests in both catchments.

The SoilGrids data base (Hengl et al., 2017) is a consistent global soil information system that provides soil physical and chemical parameters at a 250m grid resolution and seven soil depths. We utilized the available soil parameters from SoilGrids and estimated further required soil parameters with pedo-transferfunctions provided by the R package euptf (Tóth et al., 2015). The seven available soil depths from the SoilGrids data were aggregated to three soil depths (0-30cm, 30-100cm, and 100-200cm), and the gridded data were clustered into soil classes applying kmeans clustering (R Core Team, 2017, Hartigan and Wong (1979)) resulting in 14 and 8 "optimum" soil classes for the rivers Schwechat and Raab respectively.

Meteorological input data was available from the INCA system developed and operated by the Central Institute for Meteorology and Geodynamics of Austria (ZAMG; Haiden et al., 2011). INCA provides reanalysis data of precipitation and temperature on 1km grid resolution for Austria with a temporal resolution of 15 minutes for precipitation and 60 minutes for temperature in the period from 2003 to 2015. For all SWAT model setups, daily precipitation sums and daily minimum and maximum temperatures were temporally and spatially aggregated for the model subbasins.

Point source emission data was available from external emission monitoring of municipal WWTP greater than 2000 PE according to BGBl. 1996/210 (1996) for both catchments. Municipal WWTP larger than 2000 PE are responsible for 99.2%

**Table 1.** Input data for the SWAT model setup, the data sources, and data processing steps.

| Input data set | Data source | Data preparation |
|---|---|---|
| Topography | DEM Austria (Geoland.at, 2015) | Digital Elevation Model for Austria in 10m resolution. |
| Land use | CORINE Landcover (EEA, 2015), 2010 Austrian agronomic census (Statistik Austria, 2015b) | Basis: CORINE Land cover, Agricultural areas resampled with statistical information from 2010 Austrian agronomic census. |
| Soil data | soilgrids.org (Hengl et al., 2017), euptf (Tóth et al., 2015) | Basis: SoilGrids 250m resolution in 7 depths. Clustered in space and and aggregated over depth. Further SWAT soil parameters derived using pedotransfer functions. |
| Meteorology | INCA (Haiden et al., 2011) | Preciptation and temperature data in 1km resolution. |
| Agricultural practices | Statistik Austria (2015b), Land NÖ (2015) | Derive time periods and sequences of field management practices from field experiments. |
| Point source emissions | External monitoring, Internal records of WWTPs | Time series and point measurements of discharge and $NO_3^-$-N concentrations. |

and 86.3% of municipal point source emissions in the Schwechat and the Raab catchments respectively. Thus, these data cover a substantial part of the municipal emissions. Additionally, daily and weekly internal monitoring data was available for some large WWTP schemes. In most cases however, only information on $NO_3^-$-N emissions was provided. A general budgeting of nitrogen emissions however showed, that the substantial share of total nitrogen is emitted in form of $NO_3^-$-N (87% in the Schwechat catchment and 89% in the Raab catchment). For industrial emitters monthly and annual records from internal and external monitoring agencies were available and only allowed an estimation of industrial emissions with coarse temporal resolution, while covering the annual budgets. Again, mainly data for $NO_3^-$-N emissions were available. Although, nitrogen is emitted in different forms the available data basis only allowed to consider $NO_3^-$-N loads contributed by point sources.

Table 1 provides an overview of the model input data that was used for the SWAT model setup.

Hourly observations of discharge were available for the period from 2003 to 2015 at two gauges for the Schwechat and the Raab each (Fig. 1). $NO_3^-$-N concentration readings with varying time intervals of 5 to 15 minutes were available at two stations in both catchments (yellow circles in Fig. 1) for selected time periods resulting from monitoring campaigns at the rivers Schwechat (BMLFUW, 2013) and Raab (BMLFUW, 2015a, b). SWAT simulates output variables with daily time steps. To compare the observations with the modeled SWAT outputs of discharge and $NO_3^-$-N loads, daily $NO_3^-$-N loads and daily mean discharge were calculated from the observation data.

**Table 2.** SWAT model setups for the Schwechat and the Raab catchment including the numbers of subbasins and the number of HRUs for each setup.

| | Schwechat | | | Raab | |
| Setup | # Subbasin | # HRU | Setup | # Subbasin | # HRU |
| --- | --- | --- | --- | --- | --- |
| sw_14_full | 14 | 1434 | rb_54_full | 54 | 5349 |
| sw_14_thru | 14 | 196 | rb_54_thru | 54 | 954 |
| sw_03_full | 3 | 606 | rb_30_full | 30 | 3516 |
| sw_03_thru | 3 | 64 | rb_30_thru | 30 | 584 |
| | | | rb_04_full | 4 | 755 |
| | | | rb_04_thru | 4 | 115 |

## 2.4 Model setup, parameter selection and identification of non-unique parameter sets

Graphical GIS user interfaces such as ArcSWAT (Winchell et al., 2015) or QSWAT (Dile et al., 2016) facilitate the setup of SWAT models. Yet, a model setup requires the modeler to define the number of subbasins as well as the number of HRUs (e.g. by removing HRUs with areas below a certain threshold from the setup and apportion their areas to the remaining HRUs). The size and the number of subbasins in a model setup can affect the process simulations and the resulting model outputs (Jha et al., 2004; Momm et al., 2017; Tripathi et al., 2006). Removing small HRUs from the model setup and allocating their areas to the remaining HRUs affects the distribution of land use, soil types, and slope classes and thus can impact the model simulations substantially (Jha et al., 2004).

We used the ArcSWAT plugin (Version2012.10_1.14) together with ArcGIS 10.1 (ESRI, 2012) for the model setup. For both case studies we set up the SWAT model with different numbers of subbasins, whereby we prepared model setups with the full number of HRUs and respective setups with a reduced number of HRUs for each catchment.

In total, we set up four SWAT models, two with 3 and two with 14 subbasins for the Schwechat catchment and six models for the Raab catchments with two each of 4, 29, and 54 subbasins. For the full HRU setups we kept the resulting HRUs unmodified. For the model setups with a reduced number of HRUs we eliminated small HRUs. We determined thresholds for land use, soil, and slope classes to remove HRUs that have an area below these found thresholds. The thresholds were determined using the R package 'topHRU' (Strauch et al., 2016). 'topHRU' enables to find thresholds that minimize the number of HRUs of a SWAT model setup while minimizing the aggregation error (sum of changes in the areas of land uses, soils and slope classes of the reduced set of HRUs compared to the full HRU setup). To maintain a comparability between the reduced HRU setups thresholds were selected that result in an aggregation error of maximum 5% in all reduced HRU model setups. Table 2 gives an overview of the final model setups for both case studies.

In a parameter screening, we applied a GSA to the simulations of discharge and $NO_3^-$-N loads at the catchment outlets of all SWAT model setups to identify influential model parameters. Initially, 42 model parameters were selected that are frequently calibrated in SWAT model setups to simulate discharge and $NO_3^-$-N loads (see e.g. Arnold et al. (2012) and Abbaspour et al.

(2007) for a general overview of relevant model parameters, Mehdi et al. (2018) and Haas et al. (2016) for parameters control-ling the water balance and nutrient cycles, or Haas et al. (2015) for a review on the dominant nitrogen parameters). The SWAT model setup initializes the model parameters using values obtained from the SWAT data bases (either standard values or user defined, e.g. by pedotransfer functions). The selected initial ranges to modify the model parameters and the selected types of parameter changes (e.g. replace parameter values globally or modify a spatially distributed parameter field by a fraction of a parameter) reflect typical procedures often found in SWAT model calibration studies. An overview of the model parameters that were identified as influential and that were further used in the model impact study is provided in Table A1.

We employed the STAR VARS approach (Razavi and Gupta, 2016a, b) to screen and rank the model parameters. STAR VARS utilizes variograms along each model input dimension of the input space to infer each model inputs influence on a target variable over different scales (where short lag distances approximate the derivative based method of Morris (Morris, 1991) and long distances the method of Sobol (Sobol, 1993)). The calculation of the variograms is based on the tailored STAR sampling design where "star center" points are randomly sampled in the input space. For each center point cross sections are sampled along the input factor dimensions with an equally spaced interval. For each sampled input combination the model is evaluated and variograms along the response surface are calculated. Razavi and Gupta (2016a) proposed integrated measures of the variograms as measures of sensitivity, where the measures $IVARS_{10}$, $IVARS_{30}$, and $IVARS_{50}$ represent the integrals over 10%, 30%, and 50% of each input dimension respectively and therefore provide the sensitivity of a target variable to a model input over different scales. A detailed description of the method is provided in Razavi and Gupta (2016a) and the STAR sampling is outlined in Razavi and Gupta (2016b). The method proved to be robust and computationally efficient for high dimensional problems (e.g., Razavi and Gupta, 2016b; Haghnegahdar et al., 2017; Sheikholeslami et al., 2019; Haghnegahdar and Razavi, 2017).

We drew STAR samples (Razavi and Gupta, 2016b) with 50 center points and ten parameter samples per parameter di-mension that resulted in 18950 parameter combinations per model setup. The Nash Sutcliffe Efficiency criterion (NSE, Nash and Sutcliffe, 1970), the Kling Gupta Efficiency criterion (KGE), including its three components (Gupta et al., 2009), and a refined version of the Index of Agreement (Willmott et al., 2012) were used to evaluate the simulated time series of daily mean discharge and daily sums $NO_3^-$-N loads. Additionally, we applied the ratio of the root mean square error and standard deviation (RSR, (Moriasi et al., 2007)) to evaluate different segments of the FDCs of daily discharge and daily $NO_3^-$-N load simulations (Pfannerstill et al., 2014; Haas et al., 2016). All calculated criteria were included in the parameter sensitivity analysis as target variables. A model parameter was considered to be sensitive if it showed a relative sensitivity of 10% compared to the most sensitive parameter with respect to a specific objective criterion for at least one of the employed objective criteria.

The performed GSA for the model parameters of the different model setups of the Schwechat catchment and the Raab catchment respectively showed very similar results independent of the number of subbasins and HRUs of the individual model setups (Fig. A1). Therefore, for the impact study the same set of model parameters was considered as influential for all model setups of the Schwechat and the Raab, respectively. In total, 19 parameters for the Schwechat and 16 parameters for the Raab were identified to be influential for the analyzed target variables (Table A1). The majority of parameters were identified as influential parameters in the Schwechat and the Raab case study. The parameters SNO50COV, CANMX, CDN, and SDNCO

were only relevant for the model setups in the Schwechat and the parameter OV_N was only influential for in the Raab. For the majority of these parameters it is a matter of the selected threshold that defines a parameter to be influential or not. The most dominant parameters were however identified as highly relevant in both case studies.

To represent the model parametrization as an input in the subsequent sensitivity and uncertainty analysis of the environmental impact study, non-unique parameter sets were identified for the Schwechat and the Raab catchments, respectively. The preceding parameter SA revealed that changes in the model parameter values influenced the simulations similarly independent of the subbasin and HRU configurations in the Schwechat and the Raab catchment, respectively. As a consequence, but also to facilitate the separation of the effects of the model setup and the model parametrization in the analysis, we selected parameter combinations as non-unique ones that result in simulations of daily discharge and $NO_3^-$-N loads that fulfill certain objective criteria together with all model setups of the Schwechat and the Raab, respectively. For the respective 19 and 16 influential model parameters we randomly sampled 100000 parameter combinations and simulated daily discharge and $NO_3^-$-N loads with all model setups of the Schwechat and the Raab catchments. We evaluated the simulations with the following criteria to accept a parameter set: KGE > 0.5 for daily discharge at the catchment outlets, KGE > 0.4 for daily $NO_3^-$-N loads at the gauges with longer continuous records (in both case studies the gauging point within the catchment and not at the catchment outlet), percentage bias (Gupta et al., 1999) < 50% for $NO_3^-$-N loads, and absolute RSR < 1 for different discharge and $NO_3^-$-N load (according to Pfannerstill et al., 2014; Haas et al., 2016). In total, we identified 43 and 52 behavioral parameter combinations for the Schwechat and the Raab catchments, respectively. The ability of the selected parameter sets used with the different model setups to reproduce the observed data is illustrated in Fig. A2. The initial and final ranges of parameter changes are shown in Table A2. The 43 and 52 parameter combinations are additionally illustrated in parallel coordinate plots for the Schwechat and the Raab in Fig. A3 to show any clustering of individual parameters and interactions between parameters. The majority of parameters are scattered randomly and do not show any clustering or interaction with other parameters. The parameters RCN and NPERCO in the Schwechat catchment show a clear inverse relationship. This implies that the parameters compensate each other in the behavioral model setups. This finding seems plausible for the Schwechat catchment where the $NO_3^-$-N transport into the receiving waters is strongly groundwater driven and a surplus of $NO_3^-$-N input is reduced by a decrease in $NO_3^-$-N percolation. The parameters SLSOIL, SURLAG, and SOL_AWC show a clear bimodal pattern for the Raab catchment. The bimodal patterns of these parameters are strongly related and a compensation effect between these parameters is visible. Model setups with increased slope values (SLSOIL) and longer lag-times of the surface runoff (SURLAG) together with an increased soil available water content (SOL_AWC) resulted in behavioral model and were able to reproduce historic discharge and $NO_3^-$-N records, similar to the model setups where such clear relationship is not visible.

## 2.5 Scenario definition

The study involves future changes of the land use, point source emissions, and the climate. The uncertainties of these variables are expressed as discrete scenarios.

For the land use change scenarios, two scenario story lines (Rounsevell and Metzger, 2010) were developed for the Schwechat and the Raab catchments. A "business-as-usual" scenario extrapolates the trends that we determined for the dominant crops

in the time period 1970 - 2010 (Statistik Austria, 2015b) to the future (2071 to 2100), while a second "extensive" scenario assumes an extensification of agricultural activities and other intensive land uses in both catchments (Table A5).

In the Schwechat catchment population growth is the strongest factor for a future change in land use (Statistik Austria, 2015a, 2016). Hence, a transformation from extensive pasture land (-35%) to urban land use and an increase of dense urban
areas describe the "business-as-usual" scenario. The "extensive" scenario assumes no change in population and a shift of half of the wheat producing area to extensive pastures.

Since 1970, the areas for corn production increased by 220% in the Raab catchment, mostly for biogas production and at the expense of sugar beets and cereals (Statistik Austria, 2017). For the "business-as-usual" scenario, an increase in the corn area by a further 100% until the end of the century was assumed, replacing extensive pastures (-75%), sugar beets (-80%), legumes
(-70%), and winter wheat (-30%).

Groundwater protection measures lead to strict regulations for fertilizer application in the Leibnitzerfeld region adjacent to the Raab catchment (LGBl. Nr. 39/2015, 2015). Therefore, the "extensive" scenario assumes an adoption of similar nitrogen regulations in the Raab catchment. Thus, decreasing areas with intensive fertilizer application, such as corn by 50% and transforming these areas to extensive pasture land was carried out in this scenario.

Two municipal point source emission scenarios for both case studies (Table A6) and two industrial point source emission scenarios for the Raab catchment (Table A7) were developed. The future change in municipal emissions was assumed to be directly related to the change in population. For all provinces in the Schwechat basin future scenarios predict an average population growth of 32% (Statistik Austria, 2015a, 2016). The predictions of the population development in the provinces of the Raab are contradicting, with predicted changes between +2.3% (Statistik Austria, 2015a) and -20.4% (Amt d. Stmk LReg,
20   2016).

In the Raab catchment 94% of the industrial point source emissions stem from the leather industry and almost 70% of the industrial point source emissions are caused by one leather manufacturing company. Thus, industrial emission scenarios were developed for that particular manufacturer. As boundaries for the production, we defined an upper environmental boundary and a lower economical boundary for the prediction of future industrial emissions. Based on an assessment of effluent dilution
(ÖWAV, 2010), current environmental regulations (BGBl. II 2010/99, 2010; and BGBl. II 2006/96, 2006) allow an increase of 30% in emissions from that leather producer, resulting in a total increase in industrial emissions of 22.6%. Assuming a relocation of the two manufacturing sites of that leather producer to outside of the catchment would stop their emissions into the Raab, reducing the total industrial point emissions by 75.2%.

Future climate change was considered with 22 downscaled and bias corrected climate change scenarios (Table A8). Regional
climate simulations were obtained from the EU-CORDEX project (Jacob et al., 2014), providing 11 GCM-RCM simulations for the emission scenarios RCP4.5 (Smith and Wigley, 2006; Wise et al., 2009) and RCP8.5 (Riahi et al., 2007) respectively. In this study we utilized daily precipitation sums and daily minimum and maximum temperatures for the time period 2071 to 2100. The EURO-CORDEX climate simulations are available at a spatial resolution of 12.5 km (EUR-11) (Jacob et al., 2014). Statistical downscaling (Zorita and Von Storch, 1999) was applied to prepare all climate simulations at a resolution of 1 km.
To correct downscaling errors (e.g. Haslinger et al., 2013; Muerth et al., 2013), bias correction (Teutschbein and Seibert, 2013)

**Table 3.** SWAT inputs implemented in the sensitivity analysis case studies and their numbers of discrete realizations for the Schwechat and the Raab catchments.

| Input | # Values | | Details on values |
| --- | --- | --- | --- |
| | Schwechat | Raab | |
| Land use scenario | 2 | 2 | one "extensive", one "business-as-usual" |
| Point source scenario | 2 | 4 | Population growth: optimistic/pessimistic , Industry Raab: production increase/resettlement |
| Climate scenario | 22 | 22 | 11 RCP4.5, 11 RCP8.5, period: 2071-2100 |
| Model setup | 4 | 6 | Raab: 54, 30, 4 subbasins with/without HRU reduction, Schwechat: 14, 3 subbasins with/without HRU reduction |
| Parametrization | 43 | 52 | KGE discharge >0.5, KGE $NO_3^-$-N >0.4, pbias $NO_3^-$-N <50% |

was applied to the climate simulations employing quantile mapping (Hempel et al., 2013). Downscaling and bias correction were performed for the historical period 1971 to 2000, involving the reanalysis datasets SPARTACUS (Hiebl and Frei, 2016) for minimum, mean and maximum temperature and GPARD (Hofstätter et al., 2013) for daily precipitation sums.

## 2.6 Analysis

Table 3 summarizes the land use change, point source emissions, and climate change and the model setups and model parametrizations that were used for the analysis of simulated discharge and $NO_3^-$-N loads in the Schwechat and the Raab catchments. In total, 7000 combinations of land use, point source emissions, climate, model setups and model parametrizations were drawn for both case studies applying a quasi-random sampling Saltelli and Tarantola (2002). The number of combinations results from previous experiments that applying the SA method of Sobol (results not shown) using the sampling strategy proposed by Saltelli and Tarantola (2002). A base sample size of $N_b = 1000$ was used to meet the suggestions shown in Saltelli et al. (2008). Thus, the total sample size of 7000 is defined as $N = N_b(k+2)$, where $k$ is the number of model inputs ($k = 5$). Although (Sarrazin et al., 2016) report publications that required substantially larger base sample sizes (e.g. $N_b = 12000$ in Nossent et al. (2011), or $N_b = 8192$ in Tang et al. (2007)) for convergence of the ranking of influential continuous model parameters, a sample size of 7000 includes 46% and 12% of all possible model input combinations in the Schwechat and the Raab case studies, respectively. All sampled combinations were assembled to executable SWAT models. Daily discharge and daily $NO_3^-$-N loads at the outlets of the Schwechat and the Raab catchments were simulated for the period from 2071 to 2100.

The analysis of discharge and $NO_3^-$-N loads follows two main goals i) to identify the dominant controls on the simulation of discharge and $NO_3^-$-N loads in the two case studies and ii) to assess how the considered inputs control the simulation of discharge and $NO_3^-$-N loads.

### 2.6.1 Global sensitivity analysis

To measure the relative importance of the developed model input scenarios, the model setup and the parametrization on the simulation of daily discharge and daily $NO_3^-$-N loads, we employed GSA using the PAWN sensitivity index (Pianosi and Wagener, 2015). PAWN employs the empirical cumulative distribution function (CDF) of a target variable to infer the model input influence (Pianosi and Wagener, 2015). PAWN is moment-independent and was found to be a robust measure for sensitivity of non-symmetrically distributed outputs of environmental models (Pianosi and Wagener, 2015; Zadeh et al., 2017).

PAWN expresses the sensitivity of a target variable $y$ to a model input $x$ by computing a distance measure between the unconditional CDF $F_y(y)$ (where all model inputs are perturbed) and the conditional CDF $F_{(y|x_i)}(y)$ (where the model input of interest is fixed and all others are perturbed). Pianosi and Wagener (2015) proposed is the Kolmogorov-Smirnov test statistics as a distance measure. The distance $KS_j(x_i^j)$ between the CDFs for the model input $x_i$ fixed at a value $x_i = x_i^j$ is defined as:

$$KS_j(x_i^j) = \left\| F_y(y) - F_{y|x_i, x_i = x_i^j}(y) \right\|_y \tag{1}$$

To assess the overall sensitivity considering all fixed values of $x_i$, the values of $KS_j(x_i^j)$ are summarized for all $j$ sampling points. A summary statistics (Pianosi and Wagener (2015) suggested e.g. median or maximum) is applied to compute the PAWN index $T_i$ for the model input $x_i$. The model inputs that are analyzed in this study strongly differ in their numbers of discrete realizations. Further, the distribution of the resulting Kolmogorov Smirnov distances can be highly skewed (e.g. the majority of discrete realizations has a low impact, while a few realizations strongly influence the simulation). Therefore, the significance of an average sensitivity of a target variable $y_i$ to a model input $x_i$ is questionable. In a setting where the strongest impact of a model input $x_i$ on a target variable $y_i$ is of major interest the application of a maximum statistics is beneficial. Hence, the PAWN sensitivity index is defined here as:

$$T_i = \max_{x_i = x_i^1 \dots x_i^{n_i}} \left( KS_j(x_i^j) \right) \tag{2}$$

The values $x_i = x_i^1, \dots, x_i^j, \dots, x_i^{n_i}$ are the $n_i$ discrete realizations of the input $x_i$. The resulting PAWN sensitivity index varies between 0 and 1 where a lower value of $T_i$ implies a lower influence of the input $x_i$ on the target variable $y$.

Pianosi and Wagener (2015) introduced the PAWN sensitivity method using a specifically tailored sampling design to infer the PAWN indices $T_i$ for continuous model inputs $x_i$. The proposed sampling scheme suggests to draw $N_c$ conditional samples at $n$ randomly sampled points of each influencing variable $x_i$, where $x_i$ is fixed at a value $x_i = x_i^j$ while all others are perturbed. Recently, Pianosi and Wagener (2018) extended the applicability of the PAWN sensitivity method to estimate $T_i$ from a generic random sample of continuous model inputs. To approximate $T_i$ the generic sample $N$ is split into $n$ segments along each model input dimension resulting in conditional samples $N_c$ with an approximate size of $N/n$. We employed the proposed updated sampling strategy and adapted it for the use with discrete model inputs. A sample of the size $N$ was drawn. For each model input combination every model input was sampled randomly from its discrete realizations. To infer $KS_j(x_i)$ for all discrete

values $x_i^j$ of a model input $x_i$ the sample $N$ was split into subsets for all $n_i$ discrete values, resulting in subsets of the size $N/n_i$ on average. It is important to consider, that the subset size depends on the number of discrete values $n_i$ of a model input $x_i$, while the subsets resulting from the sampling scheme proposed by Pianosi and Wagener (2018) had an average size of $N/n$ for all model inputs $x_i$.

To account for the effect of different numbers of discrete realizations of the analyzed inputs, but also to assess whether the number of drawn samples of input combinations ($N = 7000$) was sufficient to perform a GSA with PAWN, confidence intervals were calculated for the PAWN indices applying bootstrapping (Hinkley, 1988; Efron, 1987) using the R package 'boot' (Canty and Ripley, 2017). To calculate the bootstrap mean and the 95% confidence intervals, 1000 bootstrap replicates were drawn (as demonstrated in Sarrazin et al. (2016)).

Signature measures of discharge and $NO_3^-$-N loads were used as target variables $y$. Signature measures are measures that describe specific characteristics of simulated time series (Euser et al., 2013) (in this case of daily mean discharge and daily sums of $NO_3^-$-N loads). We calculated quantile values (0.01, 0.05, 0.20, 0.70, 0.95, and 0.99) of daily discharge and daily $NO_3^-$-N loads, long-term mean discharges and long-term mean sums of $NO_3^-$-N loads on an annual basis and for the meteorological seasons spring, summer, autumn, and winter, and mean $NO_3^-$-N concentrations for different ranges of discharge quantiles
(very high discharge (above 0.95 quantile), high discharge (0.95 to 0.70 quantile), medium discharge (0.70 to 0.20 quantile), low discharge (0.20 to 0.05 quantile), and very low discharge (below 0.05 quantile)).

### 2.6.2    Visual analysis of the simulation uncertainties

To investigate how the inputs of land use change, changes in point source emissions, climate change, the model setup or the model parametrization control the simulation of discharge and $NO_3^-$-N loads, we analyzed the simulation outputs and their as-
sociated uncertainties visually. The 7000 assembled combinations of model inputs, model setups and parametrizations resulted in ranges of simulated discharge and $NO_3^-$-N loads. All executed model setups represent plausible realizations of the future conditions in both catchments to simulate future discharge and $NO_3^-$-N loads. Thus, the overall simulation uncertainties of simulated discharge and $NO_3^-$-N loads comprise all 7000 simulations of the Schwechat and the Raab catchments, respectively.

We visually analyzed the uncertainty bands (no thresholds were set) of the simulations of the long-term mean monthly
specific discharge, the long-term mean monthly sums of $NO_3^-$-N loads and the FDCs of daily discharge and daily $NO_3^-$-N loads. These variables are related to a wide range of the signature measures that were analyzed in the GSA and thus allow a comparison of the GSA results with the results of the visual uncertainty analysis.

The low number of possible values taken by each input allowed a more detailed analysis of their effect on the simulated uncertainties, by grouping the uncertainty bands of the discharge and $NO_3^-$-N load simulations with respect to the individual
realizations of the analyzed model input. The separated simulation uncertainty bands were additionally colored with respect to the specific properties of an input, such as the temperature or precipitation deviations of each climate scenario compared to historical records. These color ranges greatly facilitated identifying the dominant controls of the simulation.

# 3 Results

## 3.1 Sensitivity analysis

Fig. 2 summarizes the influence of the implemented land use, point source emission, climate scenarios, the different model setups and the model parametrizations for the simulation of future discharge and $NO_3^-$-N loads in the Schwechat (left) and the
Raab (right) catchments. Each plot panel shows the calculated PAWN indices for the analyzed target variables for one model input in a catchment. Related target variables are grouped by colors to support the interpretability (e.g. to identify changes in sensitivity from high to low discharge). In its entity each panel provides a general overview of the importance of an input for the simulation of discharge and $NO_3^-$-N loads. Individual PAWN indices (a single bar in a plot panel) highlight the importance of an input for the simulation of specific characteristics of the time series of discharge and $NO_3^-$-N loads.

The white boxes on top of each bar show the bootstrap means and the 95% confidence intervals (CI) of each PAWN index and therefore provides an indicator for the adequacy of the sample size that was used to perform the analysis and the impact of differing numbers of discrete values of the analyzed input variables. In general the bootstrapping resulted in narrow confidence intervals (maximum +0.05 and -0.08) for all analyzed model inputs and all signature measures providing high confidence in the resulting sensitivities. Although the numbers of discrete realizations of the analyzed model inputs (e.g. only 2 land use
scenarios, but 43 and 52 model parametrizations) differ strongly and therefore result in different subset sizes to calculate the PAWN indices, no substantial differences in the confidence intervals is visible.

The land use scenarios applied to SWAT demonstrated a rather negligible impact on all signature measures, with mean PAWN indices below 0.05 and 0.07 and confidence intervals in the same range for the Schwechat and Raab respectively (first row Fig. 2). The point source scenarios, in contrast, showed a considerable influence on the signature measures of $NO_3^-$-N
loads and concentrations in the Raab case study, while the impacts of the point sources in the Schwechat case study were negligibly low (second row Fig. 2). Thus, based on the implemented point source emission scenarios, industrial emitters in the Raab catchment are relevant for the development of in-stream $NO_3^-$-N loads and concentrations, particularly for low discharges and low $NO_3^-$-N loads. The importance of the industrial point sources in SWAT increases when higher $NO_3^-$-N load quantiles (low $NO_3^-$-N loads, from dark yellow to light yellow in Fig. 2)) and $NO_3^-$-N concentrations for low discharges (from dark red
to light red in Fig. 2) are simulated, which is evident from an increase in the mean PAWN index from 0.11 to 0.49 and 0.22 to 0.43, respectively. The climate scenarios and the model parametrizations show respective decreases in their importance for the simulation of low $NO_3^-$-N loads and $NO_3^-$-N concentrations for low discharges (with decreases in the mean PAWN index from 0.71 to 0.28 for the climate scenarios' influence on $NO_3^-$-N loads and from 0.79 to 0.36 for model parametrization's influence on $NO_3^-$-N concentrations).

The implemented climate scenarios showed large impacts on all calculated signature measures of discharge and $NO_3^-$-N loads (third row Fig. 2). The mean PAWN indices range between 0.25 to 0.90 and 0.25 to 0.96 for the Schwechat and the Raab, respectively. The climate scenarios were the most relevant inputs for the simulation of seasonal mean discharges and seasonal sums of $NO_3^-$-N loads. For the simulation of low discharge quantiles (large daily discharges) climate scenarios showed the highest relevance. For the simulation of low discharges however, the importance of the climate scenarios decreases, while the

model parametrization becomes more relevant (from dark green to light green in Fig. 2). The mean PAWN indices of climate scenarios drop from 0.74 to 0.47 in the Schwechat catchment and from 0.82 to 0.51 for the simulation of lower discharges, while the mean PAWN indices for the model parametrization show respective increases from 0.43 to 0.87 and 0.44 to 0.80.

In general, the model parametrization was highly influential for all calculated signature measures and is comparable to that
of the climate scenarios, with mean PAWN indices ranging between 0.43 to 0.90 in the Schwechat and 0.36 to 0.80 in the Raab (fifth row Fig. 2). Particularly, for the simulation of $NO_3^-$-N concentrations the model parametrization was the most dominant control of the variable simulated. In contrast to the large impact of the model parametrization, the relevance of the model setup was much lower for the simulation of discharge and $NO_3^-$-N loads and concentrations. Overall, values of the PAWN index for the choice of the model setup did not exceed 0.37, and were much smaller (two to five times) compared to
the model parametrization. The model setups yielded insignificantly low PAWN indices for the majority of signature measures with values below 0.1 in the Raab case study (2.5 % CI almost 0 for many signature measures), indicating that the model setup had a low influence on most of the analyzed processes. Only for high discharges and large $NO_3^-$-N loads a mean value for the PAWN index above 0.1 is visible.

### 3.2  Analysis of the simulation uncertainties of discharge and $NO_3^-$-N loads

Using all 7000 combinations of land use, point source emissions, climate, model setups, and model parametrizations, the simulated discharges and $NO_3^-$-N loads deviated by up to 350% (grey bands in Fig. 3) from the simulations of discharge and $NO_3^-$-N loads in the reference period 2003 to 2015 (dashed line in Fig. 3). In the Schwechat (left column in Fig. 3) wider uncertainty bands are visible for the spring and early summer months. The results for the Raab catchment (right column) show wider uncertainty bands emerged for summer as well as for winter/early spring. A notable difference between the two case
studies is how the simulations of long term monthly discharges and $NO_3^-$-N loads in the reference period compare to the ranges of future simulations. While the majority of model combinations for the Schwechat simulated larger discharges and $NO_3^-$-N loads for all months in the future, for the Raab catchment the simulations of discharge and especially $NO_3^-$-N loads are lower in comparison to the reference period.

The analyses of the uncertainty bands with respect to the implemented land use scenarios and the point source scenarios
fully confirm the results from the SA (Fig. 4). The attributed uncertainty bands for the two land use scenarios almost entirely overlap and show only minor deviations. A similar result is illustrated for the two point source scenarios in the Schwechat case study. The scenarios in the Raab catchment involved industrial point source emissions. The grouped uncertainty bands that include scenarios with an increase in industrial production (red) and the uncertainty bands that include a decrease in industrial production (blue) show similar patterns. Yet, the blue and red uncertainty bands show a clear shift to each other. On average
the scenarios with an increase in industrial production show long-term monthly sums of $NO_3^-$-N loads that are 15 tons higher compared to the scenarios with a decrease in industrial production. The same scenarios show larger amplitudes for medium and low $NO_3^-$-N loads, while large $NO_3^-$-N loads remain uninfluenced by the two scenarios for the development of the leather industry.

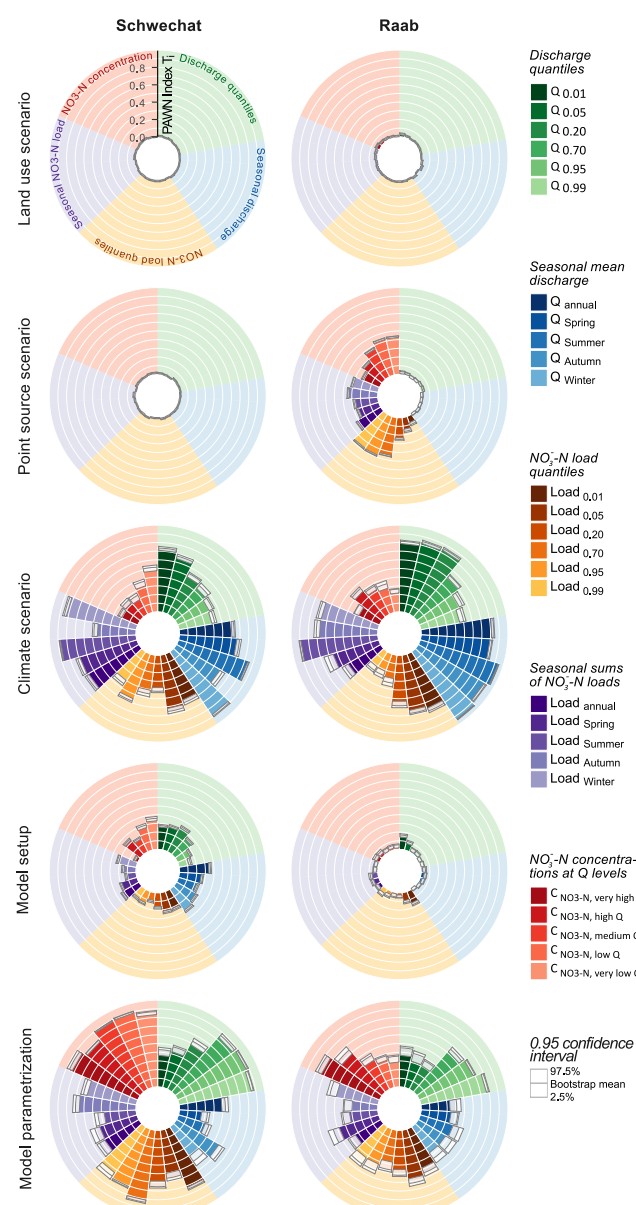

**Figure 2.** Sensitivities of signature measures of discharge and $NO_3^- - N$ loads in the Schwechat (left) and the Raab (right) catchment to the model inputs land use scenarios, point source scenarios, climate scenarios, the model setup, and the model parametrization. Each circle plot shows the set of PAWN indices calculated for the respective case study and model inputs. PAWN indices are illustrated in colored groups and clockwise order for discharge quantiles (green), seasonal long-term mean discharges (blue), quantiles of $NO_3^- - N$ loads (yellow), seasonal sums of $NO_3^- - N$ loads (purple), and mean $NO_3^- - N$ concentrations for discharge quantiles (red). The white boxes represent the bootstrap mean and the 95% confidence intervals for the calculated PAWN indices.

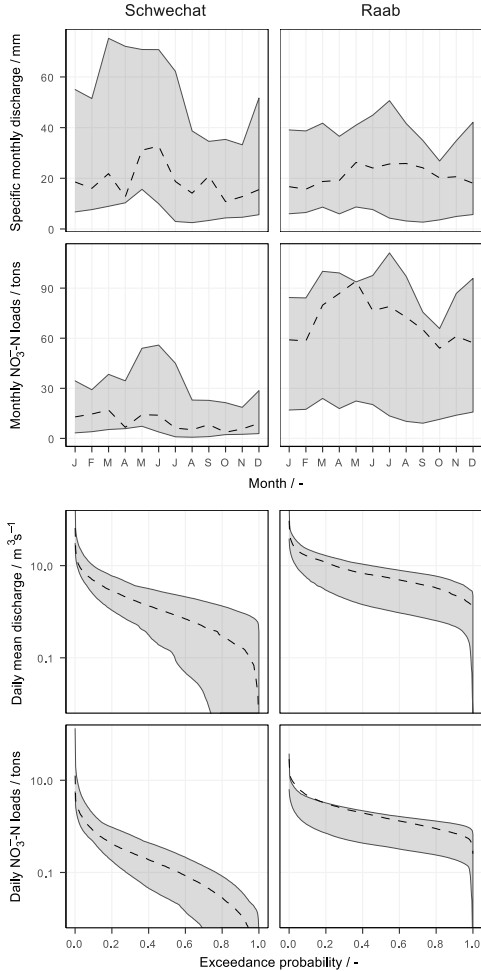

**Figure 3.** Simulated uncertainties resulting from the 7000 combinations of realizations of the influencing variables for the Schwechat (left) and the Raab (right). The grey bands illustrate the absolute ranges of simulated long-term mean monthly specific discharge (first row), long-term monthly sums of $NO_3^- - N$ loads (second row), FDCs of mean daily discharges (third row), and FDCs for daily sums of $NO_3^- - N$ loads (fourth row). The dashed lines show the best simulation of the historical reference period.

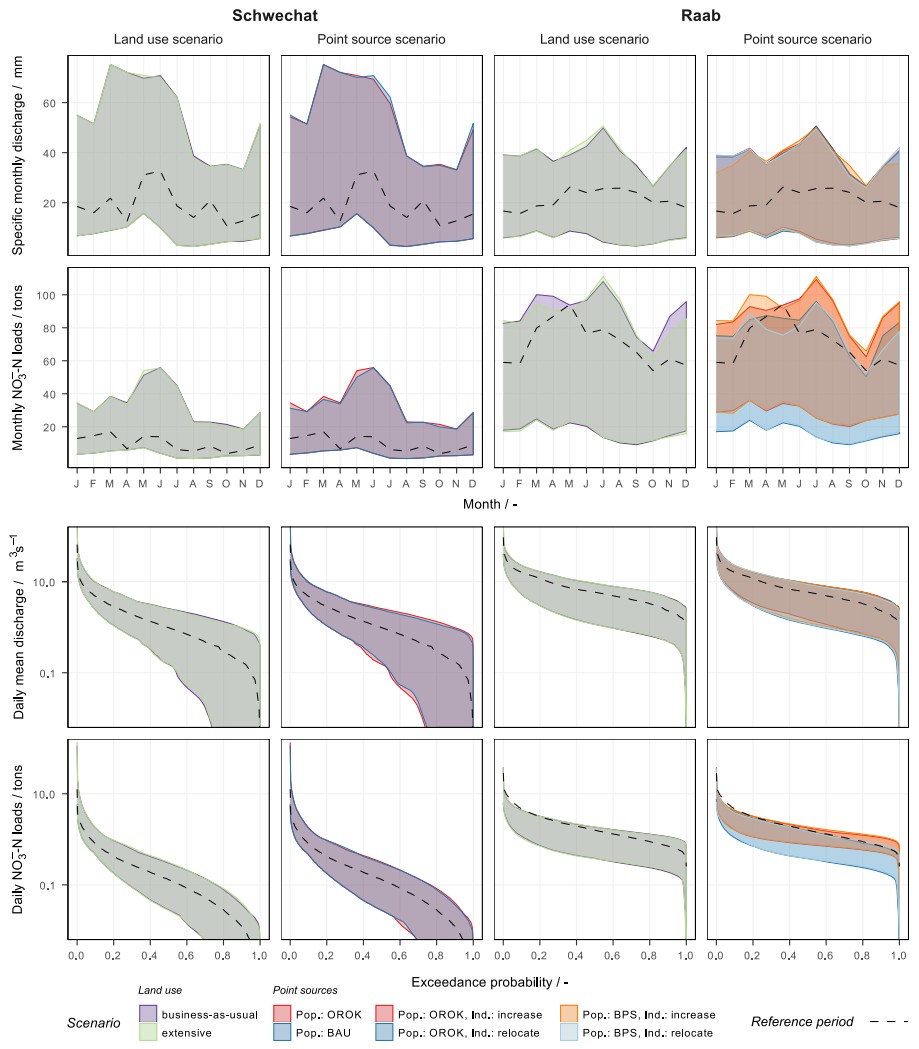

**Figure 4.** The influence of land use change and the development of point source emissions on the uncertainties resulting from the 7000 combinations of realizations of the influencing variables for the Schwechat (left) and the Raab (right). The uncertainties are illustrated for simulated long-term mean monthly specific discharge (first row), long-term monthly sums of $NO_3^- - N$ loads (second row), FDCs of mean daily discharges (third row), and FDCs for daily sums of $NO_3^- - N$ loads (fourth row). The uncertainty bands are attributed to the implemented land use scenarios (left panels per case study) and the point emission scenarios (right panels). The colors of the grouped uncertainty bands indicate the different scenarios. The dashed lines show the best simulation of the historical reference period. The corresponding land use changes are provided in Table A5. The corresponding population growth scenarios (Pop. in the legend) are listed in Table A6 and the corresponding industrial emission scenarios in the Raab catchment (Ind. in the legend) are listed in Table A7.

With the GSA we identified the climate scenarios to have a great influence on all signature measures of the simulated variables. Attributing the uncertainty bands to the individual GCM-RCM combinations unveils diverse outcomes for the future flow regime, the distribution and amplitude of monthly $NO_3^-$-N loads, as well as the appearance of high and low discharges and $NO_3^-$-N loads (Fig. 5). A visual analysis of the separated uncertainty bands identifies that the deviations of the mean

annual precipitation of the GCM-RCM combinations have a strong impact on the simulation of discharge and $NO_3^-$-N loads. In comparison to the reference period (dashed line), wetter future climate scenarios (blue) simulated larger discharge and $NO_3^-$-N loads, while drier future conditions lead to a drastic reduction in discharge and $NO_3^-$-N loads. These findings further imply that $NO_3^-$-N applied in fertilizers will remain in the upper soil layers and be transformed (mineralized or immobilized or denitrified) instead of being transported to the receiving waters. A comparison of the $NO_3^-$-N budgets of simulations with

dry and wet climate scenarios for the Raab shows a difference of up to +27% of $NO_3^-$-N accumulated in the soil, as well as a decrease of 43% and 38% in $NO_3^-$-N yield in the fast and slow runoff, respectively.

Half of the 22 implemented GCM-RCM combinations simulated an increase of more than 75 mm (dark blue) and for two GCM-RCM combinations, an increase of more than 25 mm (light blue) of precipitation for the Schwechat catchment was simulated. In contrast, for the Raab nine and four GCM-RCM combinations simulated a decrease in precipitation of more

than 75 mm (dark red) and 25 mm (light red), respectively. Consequently, a decrease in discharge and $NO_3^-$-N loads due to a decrease in precipitation is pronounced in the Raab catchment, while the majority of simulations of the Schwechat catchment show an increase in discharge and $NO_3^-$-N loads.

While a grouping of the individual climate scenarios with respect to their temperature deviations shows a more indefinite picture, all climate scenarios simulated an increase in temperature. Nevertheless, the expectation that an increase in annual mean

temperature increases evapotranspiration and thus reduces discharge and $NO_3^-$-N loads is not met in Fig. 6. A clear separation of warmer and cooler climate scenarios, as it is observable for precipitation is not the case with temperature. Consequently, the differences in precipitation predominantly account for the influence of the climate scenarios, rather than the differences in temperature.

Although the influence of the model setups was much lower compared to the influence of the climate scenarios or the model

parametrization, the analysis of the uncertainty bands for the different model setups provides interesting insights (Fig. 7). The uncertainty bands do overlap to a great extent, which confirms a low impact of the use of different model setups in the simulation of discharge and $NO_3^-$-N loads. Noteworthy is, that model setups that use the full set of HRUs agree much stronger in their simulations compared to the model setups where the number of HRUs was reduced. The difference between the full HRU and the reduced HRU model setups is distinct in the Schwechat case study. The uncertainty bands of the two full HRU

model setups almost completely overlap, although their numbers of subbasins are different (4 and 14 subbasins). The two model setups with a reduced number of HRUs (but also with 4 and 14 subbasins) show differences of up to 15 mm in the simulated monthly specific discharge and up to 7 tons in the monthly $NO_3^-$-N loads (~20 % of the uncertainty bandwidth).

The model parametrizations were relevant for all signature measures of discharge and $NO_3^-$-N loads and were most dominant for medium and low flows. The most dominant model parameters in both case studies were the parameters CNOP_till and

SOL_AWC. Both parameters control the water retention and thus the immanent contribution of rainfall to the river discharge.

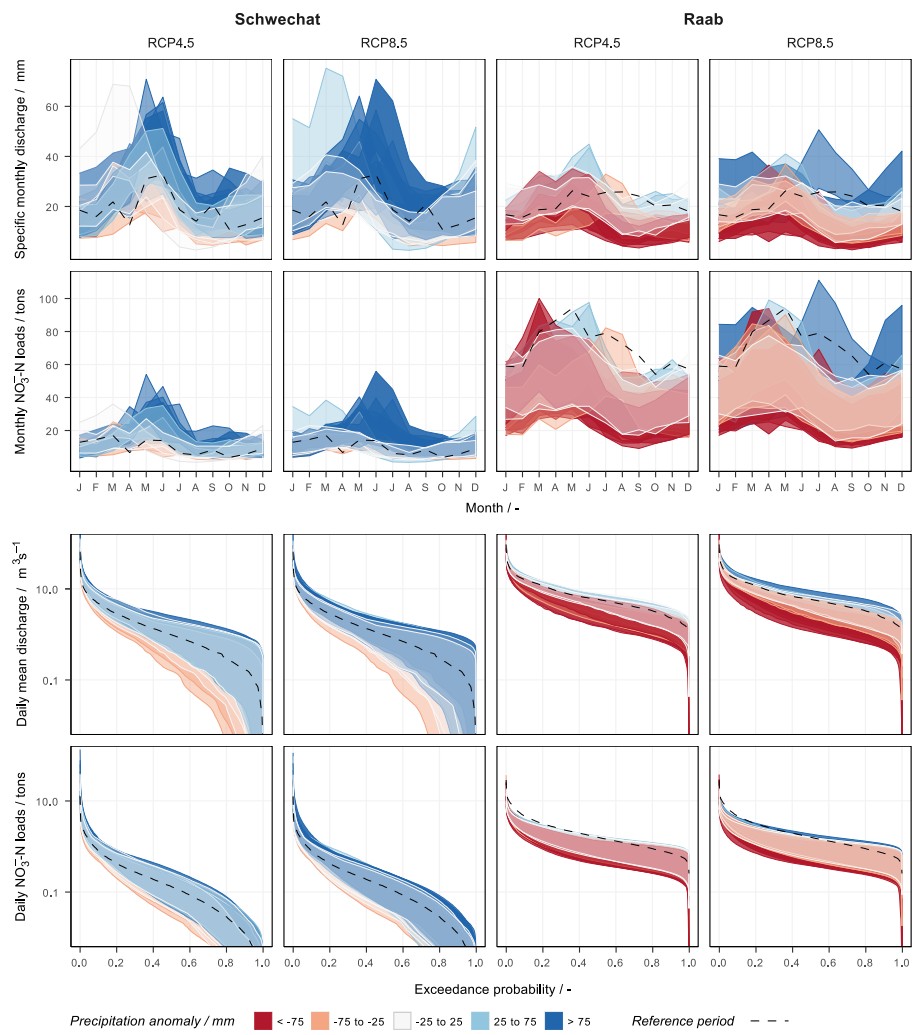

**Figure 5.** The influence of deviations in precipitation on the uncertainties resulting from the 7000 combinations of realizations of the influencing variables for the Schwechat (left) and the Raab (right). The uncertainties are illustrated for simulated long-term mean monthly specific discharge (first row), long-term monthly sums of $NO_3^- - N$ loads (second row), FDCs of mean daily discharges (third row), and FDCs for daily sums of $NO_3^- - N$ loads (fourth row). The uncertainty bands are attributed to the individual implemented climate scenarios. The colors of the uncertainty bands show the deviations in long-term mean annual precipitation of each climate scenario, where blue represents wetter conditions compared to the reference period and red dryer conditions. The dashed lines show the best simulation of the historical reference period.

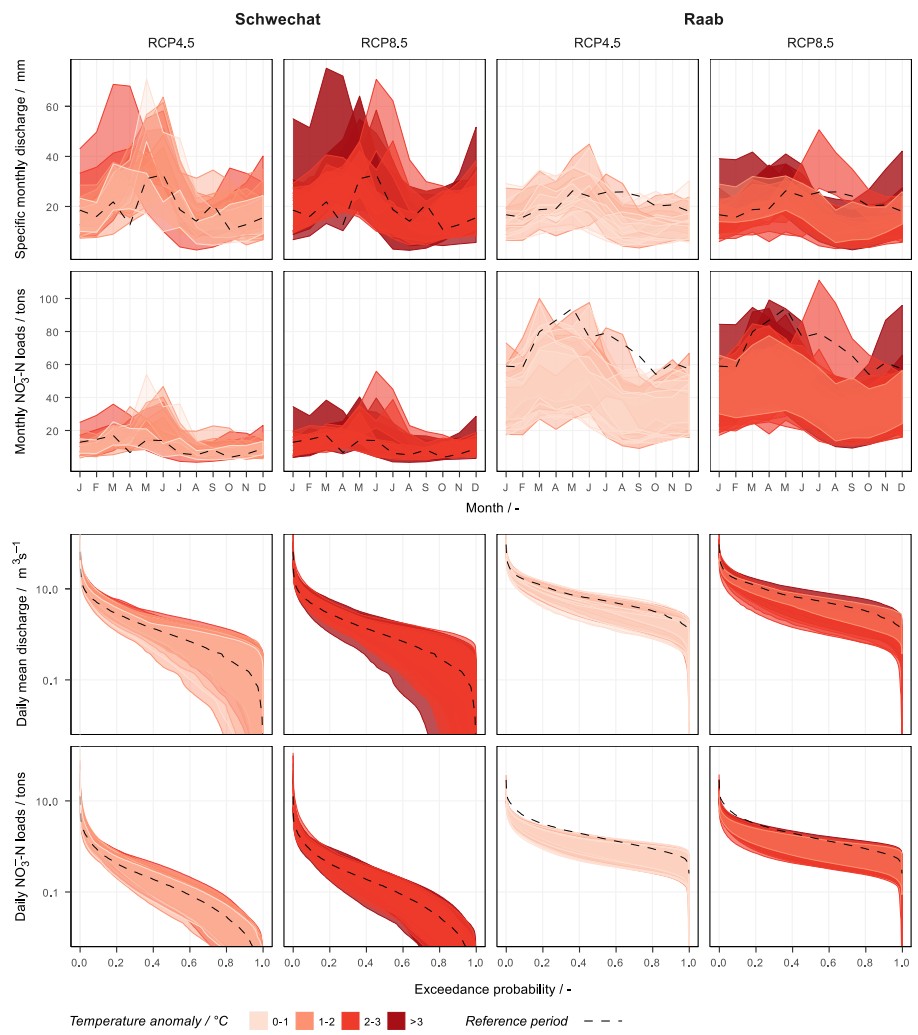

**Figure 6.** The influence of deviations in air temperature on the uncertainties resulting from the 7000 combinations of realizations of the influencing variables for the Schwechat (left) and the Raab (right). The uncertainties are illustrated for simulated long-term mean monthly specific discharge (first row), long-term monthly sums of $NO_3^- - N$ loads (second row), FDCs of mean daily discharges (third row), and FDCs for daily sums of $NO_3^- - N$ loads (fourth row). The uncertainty bands are attributed to the individual implemented climate scenarios. The colors of the uncertainty bands show the deviations in long-term mean annual air temperature of each climate scenario, where a darker red represents hotter conditions compared to the reference period. The dashed lines show the best simulation of the historical reference period.

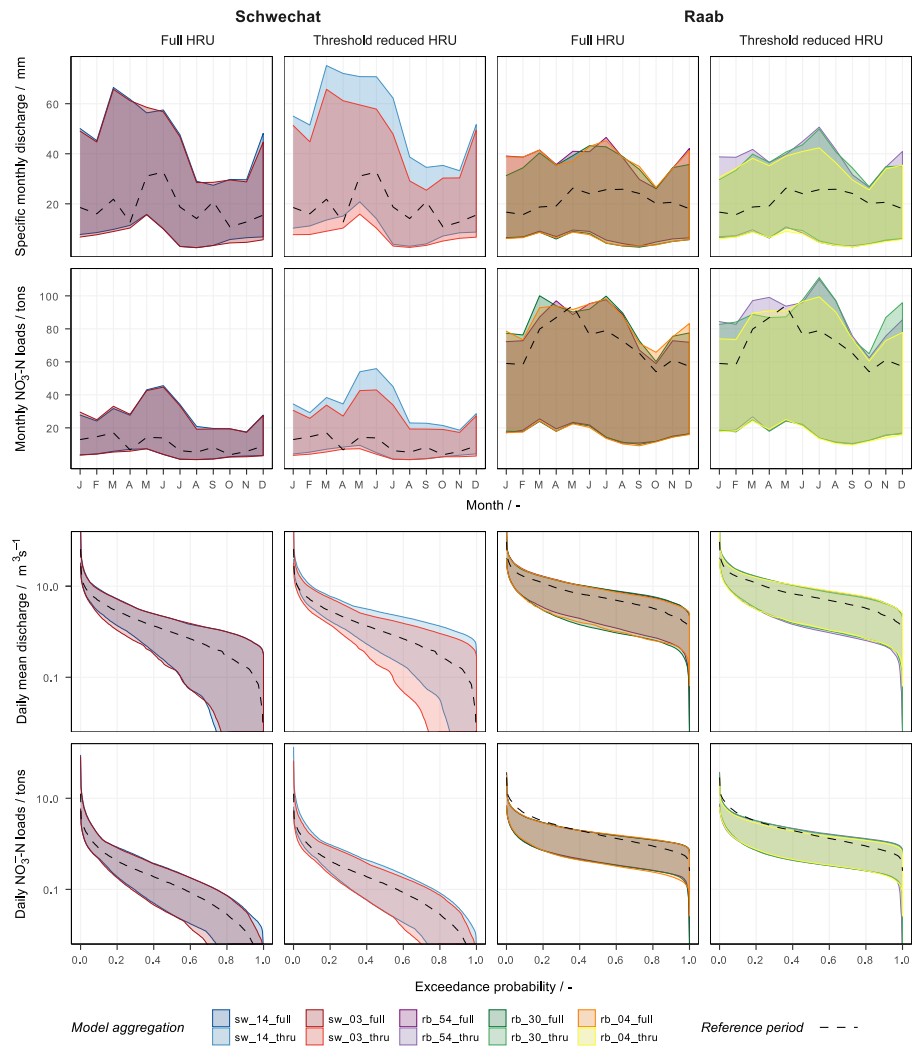

**Figure 7.** The influence of model setup on the uncertainties resulting from the 7000 combinations of realizations of the influencing variables for the Schwechat (left) and the Raab (right). The uncertainties are illustrated for simulated long-term mean monthly specific discharge (first row), long-term monthly sums of $NO_3^- - N$ loads (second row), FDCs of mean daily discharges (third row), and FDCs for daily sums of $NO_3^- - N$ loads (fourth row). The uncertainty bands are attributed to the individual SWAT model setups. The results are separated for model setups where the full set of HRUs was used (left panels per case study) and for setups with a reduced set of HRUs (right panels). The colors of the uncertainty bands show the different model setups with varying numbers of subbasins. The dashed lines show the best simulation of the historical reference period.

Large values of CNOP_till and small values of SOL_AWC reduce the water retention capacity and increase the amplitude of medium and low discharges (third row in Fig. 8). A similar but inverse behavior is visible with medium $NO_3^-$-N loads (last row in Fig. 8), where a higher water retention results in an increase of $NO_3^-$-N loads. For the long-term monthly mean discharges and sums of $NO_3^-$-N loads two effects are observable in Fig. 8. First, smaller values of CNOP_till and larger values of SOL_AWC decrease the upper boundary of the uncertainty bands. Second, selected model parametrizations with large values of CNOP_till and small values of SOL_AWC cause considerably larger discharges in spring and a strongly reduced runoff in the autumn months in the Schwechat case study.

## 4  Discussion

### 4.1  What can we as modelers learn from such analysis

The illustrated case studies emphasized the necessity to characterize, identify and explicitly communicate the uncertainties in a modeling chain, particularly for future simulations of environmental variables where large uncertainties are inherent in several modeling inputs. While the sensitivity analysis of signature measures related to discharge, $NO_3^-$-N loads and $NO_3^-$-N concentrations provided a comprehensive overview of the dominant influencing inputs on specific modeled variables, the analysis of the uncertainty bands for the simulation of the modeled variables provided insights into which properties of the model inputs (e.g. mean annual precipitation or mean air temperature of a climate scenario) control the uncertainties and how these control the simulation. The analyses allow to draw conclusions that are beneficial to consecutive steps of an impact study, for instance to refine the impact study setup and to focus on the most influential components and ultimately to reduce the uncertainties in the modeling simulation chain.

The land use scenarios showed an almost negligible impact on the simulation of discharge and $NO_3^-$-N loads. The discharge and the $NO_3^-$-N loads at the catchment are however integrated signals for the entire catchment and changes in land use may have a greater importance for particular points in a catchment. Many case studies have applied the SWAT model to assess the impact of land use change on different variables of the water cycle (Wagner et al., 2017; Mehdi et al., 2015b), water quality (Guse et al., 2015; Mehdi et al., 2015a; Teshager et al., 2016), or sediment yield (Bieger et al., 2013). Bieger et al. (2013) found very low land use change induced increases in discharge for a catchment in China. Only an assumed strong intensification of the agriculture led to a 4% increase in discharge. At the same time however, a strong increase in sediment yield of up to 450% for the summer months was simulated due to the intensification of agriculture. Guse et al. (2015) also found only small changes in simulated discharge caused by future land use change in a German lowland catchment. In absolute numbers the simulated future $NO_3^-$-N loads showed small differences between the baseline scenario and the two applied methods of land use change presented by Guse et al. (2015). Yet, the temporal patterns in $NO_3^-$-N loads caused by the different approaches of changing the land use were the major observable difference. Mehdi et al. (2015b) however found that including agricultural land use change into the impact assessment of a southern German watershed strongly increased the $NO_3^-$-N and total phosphorus loads. Teshager et al. (2016) support the findings of Mehdi et al. (2015b) and also found that corn intensive scenarios lead to an increase in discharge and significant water quality problems while an extensive scenario where mainly switchgrass is planted

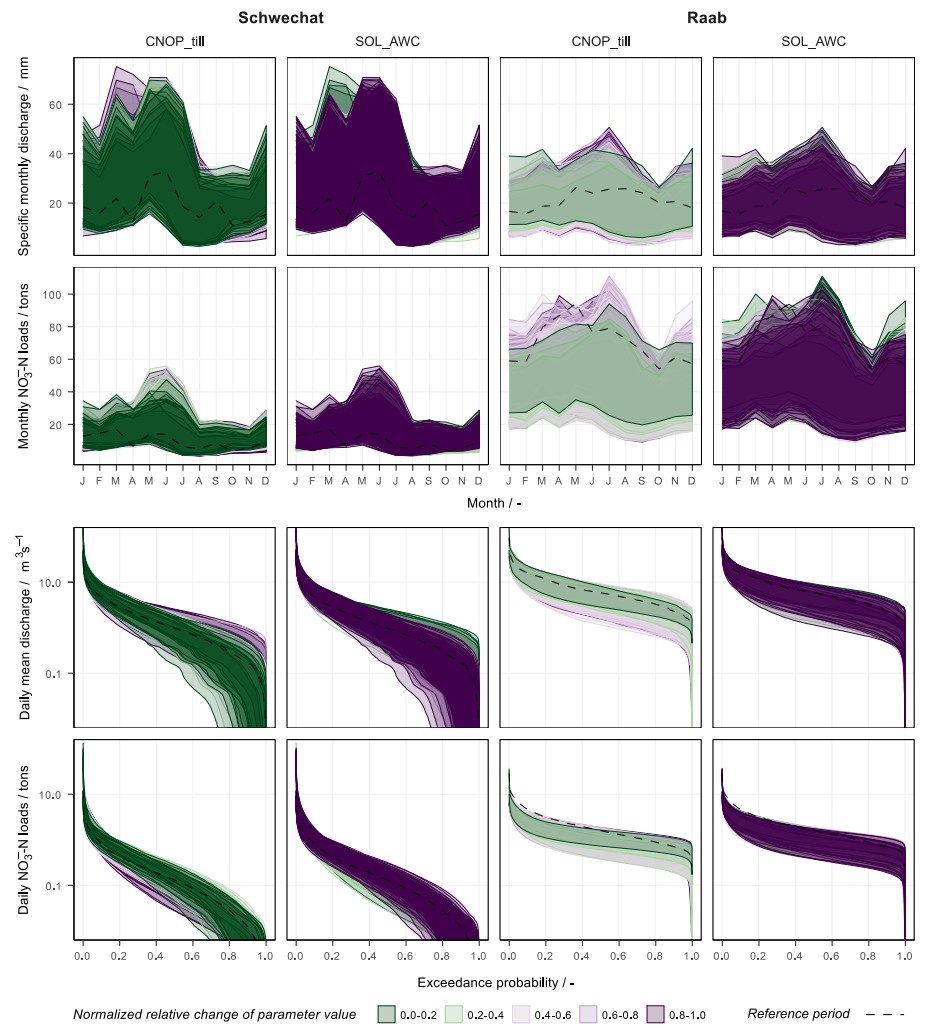

**Figure 8.** The influence of model parametrization on the uncertainties resulting from the 7000 combinations of realizations of the influencing variables for the Schwechat (left) and the Raab (right). The uncertainties are illustrated for simulated long-term mean monthly specific discharge (first row), long-term monthly sums of $NO_3^- - N$ loads (second row), FDCs of mean daily discharges (third row), and FDCs for daily sums of $NO_3^- - N$ loads (fourth row). The uncertainty bands are attributed to the individual 'behavioral' SWAT model parameter sets. The effect of the two dominant model parameters CNOP_till (left panels for each case study) and SOL_AWC (right panels) is shown. The subsetted uncertainty bands are colored with respect to the changes of the parameter values, shown as normalized values for comparability. The dashed lines show the best simulation of the historical reference period.

lead to water quality improvements under future climate change. Consequently, the low impact of land use change found in the present study seems reasonable with respect to other literature, particularly as no extreme scenarios were implemented. This does however not generally imply a low importance of land use change in environmental impact assessments. Land use change or changes in the management can be the most relevant input, particularly when strong future changes, such as possible bans of emittents are considered (Honti et al., 2017).

Industrial emitters were the main cause for the impact of point sources on medium to low $NO_3^-$-N loads. The future scenarios of the development of industrial emitters were however highly uncertain. The developed scenarios are based on expert knowledge. Yet, there is no reliable basis available on status of the industrial emitters by the end of the century. Therefore, the developed scenarios should be noted as feasible futures, rather than e.g. politically realizable futures (Godet and Roubelat, 1996). To set a feasible range as boundaries for the future development of industrial emitters can lead to an overestimation of their impact in comparison to other influencing variables. Nevertheless, the visualization of the $NO_3^-$-N FDC of the Raab case study highlights the effect of the industrial emissions for medium and small $NO_3^-$-N loads. Large $NO_3^-$-N loads however, are hardly affected by the implemented scenarios, indicating that large $NO_3^-$-N emissions are mainly driven by agricultural activities.

The selection of climate scenarios had a strong influence on the simulation of discharge and $NO_3^-$-N loads in both case studies. The analysis of the uncertainties bands identified the differences in precipitation between the GCM-RCM combinations as being the main control, while the differences in air temperature had a low impact on the simulation outcome. This finding stands in contrast to other studies. Milly and Dunne (2011) and Sheffield et al. (2012) for example, identified empirical approaches for the calculation of evapotranspiration as the main source for overestimation of the climate's influence on hydrological processes, particularly when evapotranspiration is a function of air temperature (Clark et al., 2016; Shaw and Riha, 2011; Roderick et al., 2014). In the climate scenarios used in this study, the impact of large differences in mean annual precipitation on the simulated outputs exceeded the impact of the differences in air temperature.

The effect of the model setup, with different watershed subdivisions, on the simulation of discharge or water quality variables has been investigated in various studies (e.g. Jha et al., 2004; Momm et al., 2017; Pignotti et al., 2017). Jha et al. (2004) emphasize the greater impact of changes during the HRU definition over the defined number of subbasins, as a consequent change in the distribution of land use, soil, or topography strongly affect runoff and the nutrient budget in a catchment. The analysis of the uncertainties bands with respect to the different model setups clearly confirmed the study by Jha et al. (2004), especially in the case of the Schwechat. Nevertheless, the impact of the model setup was lower than the effect of the model parametrization by a factor of up to five in the Schwechat study and up to eight in the Raab case study. Yet, the model setup strongly affects the computation time. In the present case, where aggregated discharge and $NO_3^-$-N loads at the catchment outlets were the variables of interest a strong focus on the model parametrization is of higher priority than the spatial distribution of the model setup. Therefore, to maintain short computation times (and at the same time to maintain the distributions of land use, soil, or topography) a model setup with a low number of subbasins without any reduction of the number of HRUs is beneficial.

The impact of parameter non-uniqueness on the simulation of hydrological and water quality variables has been demonstrated previously (e.g.; Wilby, 2005; Mehdi et al., 2018). The importance of the model parametrization for the simulation of discharge and $NO_3^-$-N loads was confirmed in the present study as well. Large sensitivities of all signature measures of discharge and $NO_3^-$-N loads to the different model parametrizations were identified . Although all selected parameter sets

represented historical observations of discharge and $NO_3^-$-N loads with a certain goodness of fit (based on defined objective criteria), the colored grouping of the uncertainty bands illustrated that the selected model parameter sets control the simulation of future discharge and $NO_3^-$-N loads in different ways. Thus, the large impact of the model parametrization and the distinctive patterns identified in the uncertainty bands suggest a great potential to further refine the model parametrization and consequently reduce simulation uncertainties with a more intensive model calibration. Additional information on the time series of

observations can help to constrain the model parameters and adequately describe the relevant processes (e.g. Hrachowitz et al., 2014; Pfannerstill et al., 2017).

## 4.2 How to attribute subjectivity inherent in the scenarios

Scenarios always reflect subjective assumptions made by the modeler. Assumptions that are made in the scenario development however, can strongly influence a simulation and thus affects a comparison of different model inputs and their impacts on

the simulation. All steps in a scenario development involve subjective assumptions and can lack plausibility (Mahmoud et al., 2009; van Vuuren et al., 2012), regardless of whether the process involves expert knowledge, the input of stakeholders in an participatory process, or an exploratory approach that extrapolates trends, these practices potentially introduce uncertainties in the definition of scenarios. Technical aspects such as how the scenario is represented in the model are also strongly biased by the modelers decision and represent an additional source of uncertainty (Mahmoud et al., 2009). The communication of the

potential uncertainties inherent in the developed scenarios and the boundaries of the explanatory power of an scenario ensemble is essential for the integrity of any impact study (Mahmoud et al., 2009; Jones et al., 2014).

In the present study, several assumptions were made in the development of scenarios that are highly subjective, such as the extrapolated gradient of future land use changes, the drastic changes in future industrial emissions, and also the selection of objective criteria that define a behavioral SWAT model setup. Scenarios must cover a broad range of possible futures and have

to be adequately represented in the model setup. An explicit delineation of the implemented scenarios and their limitations is essential to clearly illustrate the limitations of an impact study's conclusions. An immanent risk in any impact study is that the model representation of a future change, or the uncertainties in a model input fail to reproduce the response of a simulated variable that would have taken place in the real environmental system. Hence, a detailed analysis of the simulation uncertainties perfectly complements a SA to identify possible shortcomings in the study setup. Attributing the uncertainty bands resulting

from the simulation of an environmental variable to individual model inputs prove to be a useful visual analysis tool that gives the power to illustrate the uncertainties in a transparent way. Furthermore, the colored differentiation provides a visual guidance to judge the impacts of different implemented scenarios.

### 4.3 Sensitivity analysis or hydrologic storylines

The presented approach implements large samples combining scenarios for different model inputs and different model setups and parametrizations in a GSA to identify the dominant contributors of uncertainties in the simulated outputs. The utilization of SA with large sample sizes however, raises the following issues: i) compared to a standard approach to perform an impact assessment, where a few different future scenarios are implemented into a model, the computational demand of a GSA requiring hundreds or thousands of model executions is larger by several orders of magnitude. Thus, a practical implementation of the presented procedure in impact studies is questionable and a strong cooperation between research and the practitioners is essential. ii) scenarios of different model inputs are often interrelated (Mahmoud et al., 2009). A change in one model input therefore for example expects the change of another model input into one direction and makes a change into another direction unlikely. While the implementation of input dependencies, althouh challenging is feasible for continuous model inputs, for instance by a transformation of the input space (e.g., Tarantola and Mara, 2017; Mara and Tarantola, 2012), or the determination of input distribution functions (Hart and Gremaud, 2018), the dependencies of composite model inputs are usually difficult to express mathematically. To identify the dependencies between composite model inputs, expert knowledge is required to properly constrain the model input combinations and therefore complicates the implementation in approaches, such as the presented one.

Clark et al. (2016) therefore suggest to identify consistent hydrologic story lines that result in least severe, most likely, and most severe responses of the modeled system. Such an approach would tremendously reduce the number of necessary model evaluations, but also establish consistency between the considered influencing variables. Nevertheless, the feasible combinations of influencing variables that lead to extreme or likely responses of the modeled system are hardly known a priori. Consequently, a sensitivity analysis with a constrained sampling space, to avoid infeasible combinations of influencing variables might be a pragmatic compromise.

### 5 Conclusions

In this study we utilized methods for GSA in environmental impact studies to identify the dominant sources of uncertainties for the simulation of environmental variables under future changing conditions. In two Austrian case studies for the rivers Schwechat and Raab, we simulated the river discharge and the $NO_3^-$-N loads from the catchments under the condition of future changes in climate, land use, and emissions from urban and industrial point sources implementing different SWAT model setups with various model parametrizations.

Both case studies identified climate change and the model parametrization to be the most important (influential) model inputs for the simulation of discharge and $NO_3^-$-N loads, based on performing a GSA and on the resulting analysis of signature measures of discharge and $NO_3^-$-N loads (quantiles of discharge and $NO_3^-$-N loads, seasonal mean discharge and seasonal sums of $NO_3^-$-N loads and $NO_3^-$-N concentrations for discharge quantiles). The impact of the model setup on simulated variables of discharge and $NO_3^-$-N loads was found to be considerably lower than the impact of the model parametrization for the Schwechat and even more distinct for the Raab. The impact of the implemented scenarios for land use and municipal

point source emissions were negligible for all analyzed signature measures. Because of a large leather industry in the Raab catchment, the future development of industrial emission in the Raab catchment was found to be relevant for low $NO_3^-$-N loads and $NO_3^-$-N concentrations during low discharge.

Accompanying the GSA, a detailed analysis of the simulation uncertainties provided additional insights on how the uncertainties in the model inputs control simulated discharge and $NO_3^-$-N loads. The visualizations we developed supported the identification of the relevant properties of the model inputs that control the simulation uncertainties and provide insight how individual realizations of a model input can affect the simulations. In the climate simulations, we found the precipitation to dominate the simulation outputs, rather than changes in air temperature. Although the impact of the model setup on the simulation of discharge and $NO_3^-$-N loads was low, the visual analysis of the uncertainty bands illustrated that the HRU definition is an important step in the model setup. The use of the full set of HRUs was identified as the preferred setup in the two case studies. In contrast the effect of using different numbers of subbasins in the model setup was low for the simulation of discharge and $NO_3^-$-N loads at the catchment outlets.

The drawn conclusions are the result of specific conditions and the assumptions made for each individual catchment in the two case studies. The conclusions cannot be extrapolated with ease to other catchments. Nevertheless, the presented work provides an approach to identify and analyze the dominant sources of simulation uncertainties in environmental impact studies that can easily be generalized and that can act as a template for further impact studies. The analyses advocate for a stronger focus on the communication of uncertainties in model simulation and their sources in environmental impact studies. Although a variety of tools to perform SA are available for different programming languages (e.g., Pianosi and Wagener, 2015; Reusser, 2015; Iooss et al., 2018; Houska et al., 2015), the main constraint for a practical application remains the development of a comprehensive set of discrete input realizations, the computational costs of such analysis, and the lack if straight forward methods to implement composite inputs into SA. This might detain the practical application of such methods. To facilitate the implementation of composite model inputs in SA, we plan to implement the demonstrated procedures and tools for visualization into a user friendly programming environment.

**Appendix A: Supplementary figures and tables**

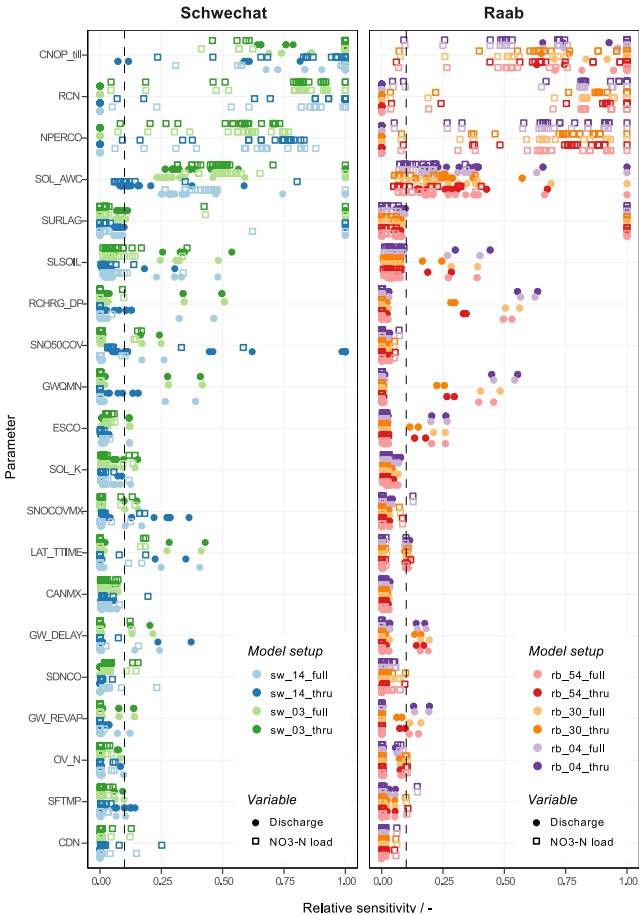

**Figure A1.** Identification of the influential SWAT model parameters for the case studies Schwechat (left) and Raab (right). The y-axis illustrates model parameters that showed an impact on at least one of the analyzed objective criteria. The x-axis shows the relative sensitivities of analyzed objective criteria (in relation to the most influential parameter for an objective criterion). The colors indicate the different SWAT model setups. The circles show the sensitivities for objective criteria related to discharge, while the hollow squares show parameter sensitivities for $NO_3^- - N$ loads. The dashed line indicates the 0.1 value of relative sensitivity. A parameter is considered to be sensitive if it resulted in a relative sensitivity above this threshold for the objective criteria.

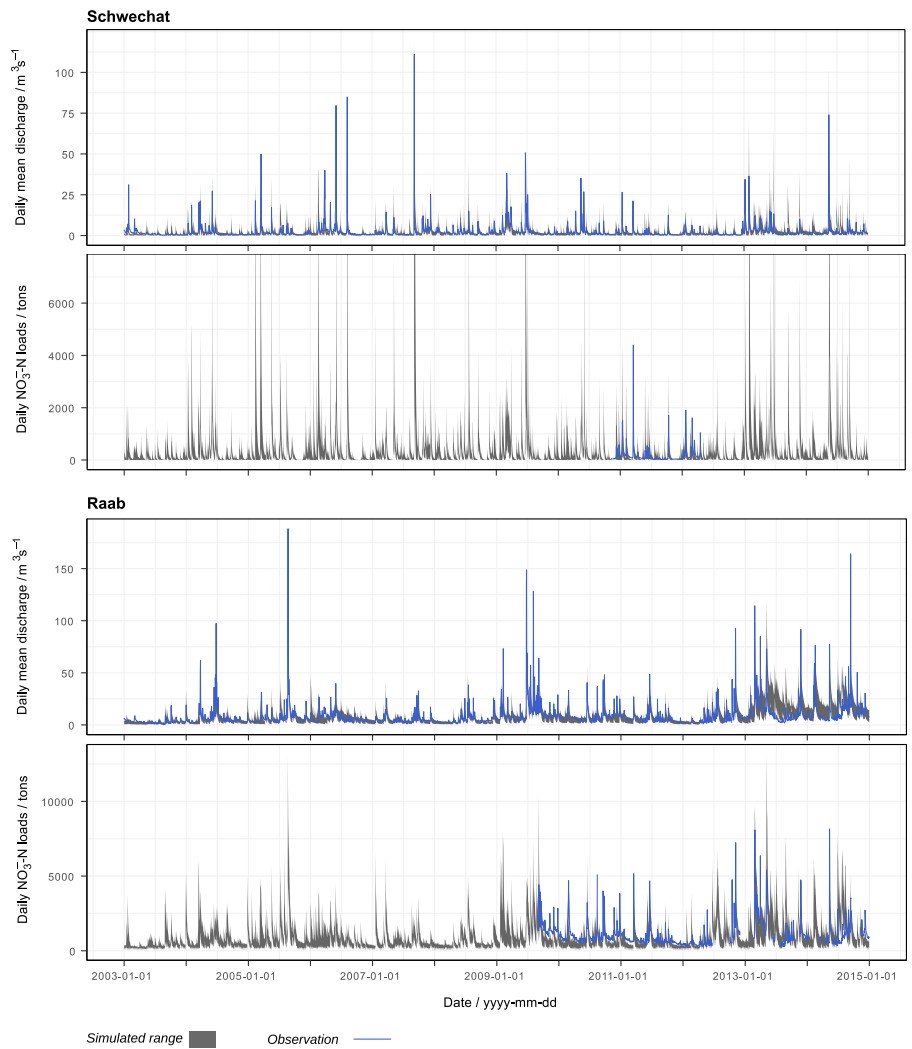

**Figure A2.** Simulated time series of daily mean discharge and daily $NO_3^- - N$ loads for the Schwechat (top) and the Raab (bottom) catchments for the time period 2003 to 2015. The gray bands show the ranges simulated using the selected model parameter sets with the different SWAT model setups. The blue solid lines indicate available observations of discharge and $NO_3^- - N$ loads for the respective time periods.

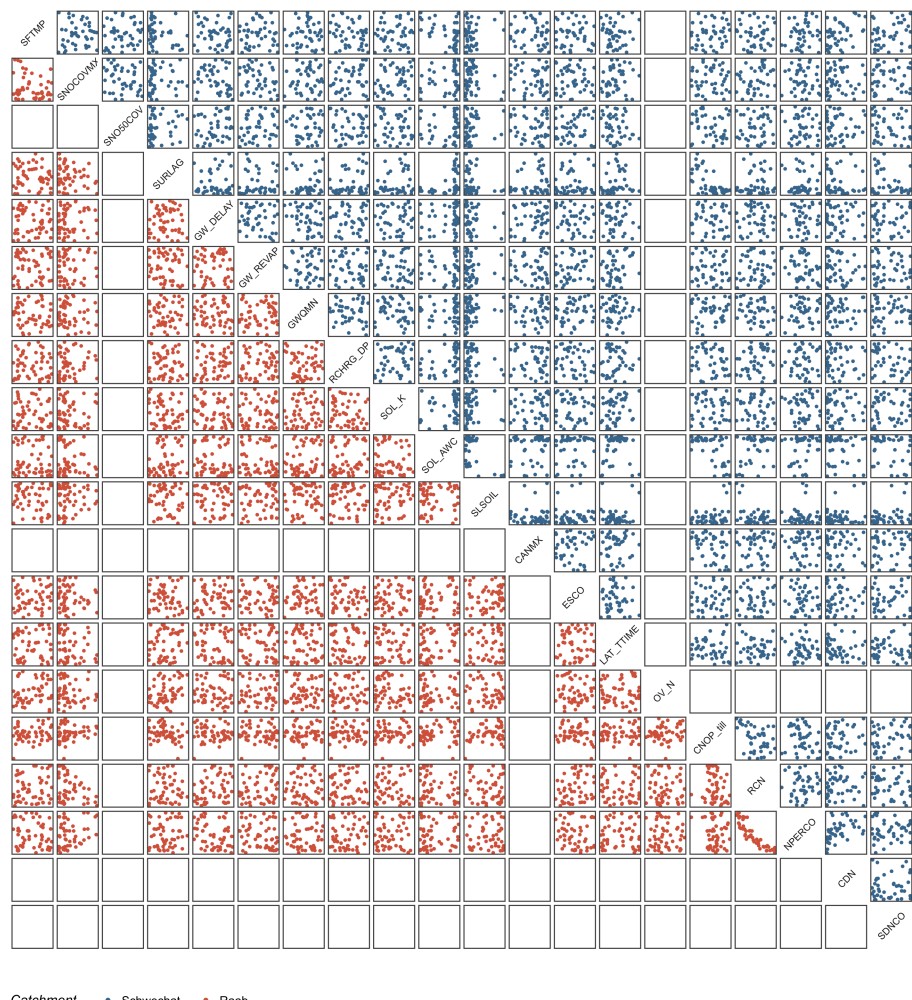

*Catchment* • Schwechat • Raab

**Figure A3.** Parallel coordinate plot of the 43 and 52 behavioral SWAT model parameter combinations that were used with the model setups of the Schwechat and the Raab, respectively. Each panel illustrates the interaction of two model parameters. The parameter combinations for the Schwechat are illustrated in red (below the diagonal) and the combinations for the Raab are given in blue (above the diagonal). The x and y axes of each panel show the range of the respective parameter plotted along the x or y dimension. The corresponding parameter ranges for all illustrated parameters are provided in Table A2.

Table A1: Influential and non-influential SWAT model parameters for the model setups of the Schwechat and the Raab.

| Parameter | Description | Influential for discharge | | Influential for NO$_3^-$-N loads | |
|---|---|---|---|---|---|
| | | Schwechat | Raab | Schwechat | Raab |
| SFTMP | Snowfall temperature ($degC$) | X | | | X |
| SNOCOVMX | Minimum snow water content that corresponds to 100% snow cover | X | | X | X |
| SNO50COV | Snow water equivalent that corresponds to 50% snow cover | X | | X | |
| SURLAG | Surface runoff lag time ($h$) | X | X | X | X |
| GW_DELAY | Groundwater delay ($d$) | X | X | X | |
| GW_REVAP | Groundwater revaporation coefficient | X | X | | |
| GWQMN | Threshold depth of water in shallow aquifer for return flow ($mm$) | X | X | | |
| RCHRG_DP | Deep aquifer percolation fraction | X | X | | |
| SOL_K | Saturated hydraulic conductivity ($mm \cdot h^{-1}$) | X | | X | |
| SOL_AWC | Available water capacity of the soil layer | X | X | X | |
| SLSOIL | Slope length for lateral subsurface flow ($m$) | X | X | X | X |
| CANMX | Maximum canopy storage ($mm$) | | | X | |
| ESCO | Soil evaporation compensation factor | X | X | | |
| LAT_TTIME | Lateral flow travel time | X | X | X | X |
| OV_N | Manning's n-value for overland flow | | X | | X |
| CNOP_till | SCS runoff curve number for the tillage operation | X | X | X | X |
| RCN | Concentration of nitrogen in rainfall | | | X | X |
| NPERCO | Nitrogen percolation coefficient | | | X | X |
| CDN | Denitrification exponential rate coefficient | | | X | |
| SDNCO | Denitrification threshold water content | | | X | |
| SMTMP | Snow melt base temperature ($degC$) | | | | |
| SMFMX | Melt factor for snow on June 21 ($mm \cdot degC^{-1}$) | | | | |
| SMFMN | Melt factor for snow on Dec. 21 ($mm \cdot degC^{-1}$) | | | | |
| TIMP | Snowmelt temperature lag factor | | | | |
| CH_N1 | Manning's n value for the tributary channels | | | | |

Table A1: Influential and non-influential SWAT model parameters for the model setups of the Schwechat and the Raab.

| Parameter | Description | Influential for discharge | | Influential for $NO_3^-$-N loads | |
|-----------|-------------|---------------------------|---|----------------------------------|---|
| | | Schwechat | Raab | Schwechat | Raab |
| CH_N2 | Manning's n value for the main channel | | | | |
| CH_K1 | Effective hydraulic conductivity in tributary channel alluvium $(mm \cdot h^{-1})$ | | | | |
| CH_N2 | Effective hydraulic conductivity in main channel alluvium $(mm \cdot h^{-1})$ | | | | |
| ALPHA_BNK | Baseflow alpha factor for bank storage $(d)$ | | | | |
| ALPHA_BF | Baseflow alpha factor $(d^{-1})$ | | | | |
| REVAPMN | Threshold depth in the shallow aquifer for revap or percolation $(mm)$ | | | | |
| GW_SPYLD | Specific yield of the shallow aquifer $(m^3 \cdot m^{-3})$ | | | | |
| RCHRG_DP | Deep aquifer percolation fraction | | | | |
| SLSUBBSN | Average slope length $(m)$ | | | | |
| EPCO | Plant uptake compensation factor | | | | |
| CN2 | SCS Curve Number for soil moisture II | | | | |
| CNOP_plant | SCS runoff curve number for the planting operation | | | | |
| CNOP_hrvst | SCS runoff curve number for the harvesting operation | | | | |
| SHALLST_N | Initial concentration of nitrate in shallow aquifer $(mg \cdot L^{-1})$ | | | | |
| HLIFE_NGW | Half-life of nitrate in the shallow aquifer $(d)$ | | | | |
| N_UPDIS | Nitrogen uptake distribution parameter | | | | |
| CMN | Rate factor for humus mineralization of active organic nutrients | | | | |

**Table A2.** Ranges of parameter changes for the behavioral model parameter sets. The type of change indicates whether a model parameter was replaced by absolute values, altered by adding an absolute to the initial parameter value, or changed by a relative fraction of the initial parameter value. The initial ranges of parameter changes and the ranges of parameter ranges of the behavioral parameter combinations in the model setups of the Schwechat and the Raab are shown.

| Parameter | Type of change | Range of parameter change | | |
| --- | --- | --- | --- | --- |
| | | Initial range | Schwechat | Raab |
| SFTMP | replace value | [-1.00, 1.00] | [-0.69, 0.93] | [-0.98, 0.88] |
| SNOCOVMX | replace value | [100.0, 500.0] | [0.9, 177.0] | [100.8, 447.5] |
| SNO50COV | replace value | [0.20, 0.50] | [0.21, 0.49] | |
| SURLAG | replace value | [0.00, 18.00] | [0.02, 0.99] | [0.01, 0.10] |
| GW_DELAY | replace value | [0.0, 300.0] | [5.5, 25.0] | [2.1, 283.3] |
| GW_REVAP | replace value | [0.02, 0.20] | [0.05, 0.15] | [0.02, 0.20] |
| GWQMN | replace value | [0, 3000] | [567, 2472] | [109, 2925] |
| RCHRG_DP | replace value | [0.01, 1.00] | [0.31, 0.69] | [0.13, 0.97] |
| SOL_K | relative change | [-0.90, 10.00] | [0.00, 0.97] | [-0.79, 9.76] |
| SOL_AWC | relative change | [-0.90, 2.00] | [-0.86, 1.49] | [0.01, 1.98] |
| SLSOIL | replace value | [0.0, 150.0] | [0.9, 27.6] | [14.7, 148.2] |
| CANMX | relative change | [-0.90, 2.50] | [0.34, 2.40] | |
| ESCO | replace value | [0.00, 0.90] | [0.05, 0.9] | [0.05, 0.89] |
| LAT_TTIME | replace value | [0.0, 180.0] | [0.8, 6.8] | [5.5, 176.3] |
| OV_N | absolute change | [-0.09, 0.60] | | [0.07, 0.58] |
| CNOP_till | relative change | [-0.20, 0.10] | [-0.19, -0.06] | [-0.18, 0.01] |
| RCN | replace value | [2.00, 10.00] | [5.05, 9.97] | [2.30, 8.45] |
| NPERCO | replace value | [0.00, 1.00] | [0.24, 0.99] | [0.18, 0.7] |
| CDN | replace value | [0.00, 1.50] | [0.01, 1.44] | |
| SDNCO | replace value | [0.00, 0.50] | [0.02, 0.49] | |

**Table A3.** Area and percentage of the land uses in the Schwechat catchment. The land use groups are the respective land uses shown in Fig. 1 and are derived from CORINE. With a higher thematic resolution the land uses that were implemented in the SWAT models are listed providing their areas and their percentages in the catchment.

| Land use group | CORINE Level 3 | Land use | SWAT Land use | Area / ha | Percentage / % |
|---|---|---|---|---|---|
| Urban/Industrial | 11X, 14X | Urban medium density | URMD | 154.2 | 0.6 |
| | 11X, 14X | Urban medium/low density | URML | 2388.3 | 8.7 |
| | 12X | Industrial | UIDU | 209.5 | 0.8 |
| Agriculture, Complex Cultiv. | 221, 222, 242 | Winter wheat, winter grains | WWHT | 667.6 | 2.4 |
| | | Spring wheat, summer grains | SWHT | 317.8 | 1.2 |
| | | Corn, Maize | CORN | 111.5 | 0.4 |
| | | Vegetables grouped | SGBT | 74.1 | 0.3 |
| | | Sunflower | SUNF | 30.0 | 0.1 |
| | | Soybean | SOYB | 19.7 | 0.1 |
| | | Orchard, Fruit trees | ORCD | 25.6 | 0.1 |
| | | Vineyard | GRAP | 699.5 | 2.5 |
| Grassland, Complex Cultiv. | 231, 242 | Pasture, extensive use | FESC | 2406.6 | 8.8 |
| | | Pasture, intensive use | FESI | 762.9 | 2.8 |
| | | Alfalfa, clover, etc. | ALFA | 400.7 | 1.5 |
| Deciduous forest | 311 | Forest, deciduous | FRSD | 12941.3 | 47.1 |
| Coniferous forest | 312 | Forest evergreen | FRSE | 1152.2 | 4.2 |
| Mixed forest | 312 | Forest, mixed | FRST | 5138.4 | 18.7 |
| | | | | 27499.9 | 100.0 |

**Table A4.** Area and percentage of the land uses in the Raab catchment. The land use groups are the respective land uses shown in Fig. 1 and are derived from CORINE. With a higher thematic resolution the land uses that were implemented in the SWAT models are listed providing their areas and their percentages in the catchment.

| Land use group | CORINE Level 3 | Land use | SWAT Land use | Area / ha | Percentage / % |
|---|---|---|---|---|---|
| Urban/Industrial | 11X, 14X | Urban medium/low density | URML | 11850.8 | 12.0 |
| Agriculture, Complex Cultivation | 221, 222, 242 | Corn, Maize | CORN | 11982.5 | 12.1 |
| | | Oil seed pumpkin | OELK | 3171.1 | 3.2 |
| | | Vegetables grouped | SGBT | 3035.9 | 3.1 |
| | | Winter wheat, winter grains | WWHT | 1855.6 | 1.9 |
| | | Spring wheat, summer grains | WWHT | 981.9 | 1.0 |
| | | Soybean | SOYB | 445.9 | 0.5 |
| | | Orchard, fruit trees | ORCD | 3036.1 | 3.1 |
| Grassland, Complex Cultivation | 231, 242 | Pasture, extensive use | FESC | 11635.7 | 11.8 |
| | | Pasture, intensive use | FESI | 8474.0 | 8.6 |
| | | Alfalfa, clover, etc. | ALFA | 598.0 | 0.6 |
| Deciduous forest | 311 | Forest, deciduous | FRSD | 15379.4 | 15.6 |
| Coniferous forest | 312 | Forest evergreen | FRSE | 7773.2 | 7.9 |
| Mixed forest | 312 | Forest, mixed | FRST | 18540.2 | 18.8 |
| Waterbodies | 41X | Wetlands, mixed | WETL | 55.4 | 0.1 |
| | | | | 98815.9 | 100.0 |

**Table A5.** Transformations of land uses (LUSE) in the implemented land use scenarios at the Schwechat and the Raab.

| | "business-as-usual" | | | "extensive" | |
| From LUSE | To LUSE | Change %/ha | From LUSE | To LUSE | Change %/ha |
|---|---|---|---|---|---|
| Schwechat: | | | | | |
| Urban, light | Urban, dense | 10 / 239 | Winter wheat | Ext. pasture | 27.5 / 184 |
| Ext. pasture | Urban, light | 15 / 361 | Winter wheat | Legumes | 27.5 / 184 |
| Ext. pasture | Winter wheat | 20 / 481 | | | |
| Raab: | | | | | |
| Ext. pasture | Corn | 75 / 8726 | Corn | Ext. pasture | 27.5 / 3595 |
| Sugar beet | Corn | 80 / 2429 | Corn | Legumes | 27.5 / 3595 |
| Legumes | Corn | 70 / 419 | | | |
| Winter wheat | Corn | 30 / 557 | | | |

**Table A6.** Municipal point source emissions and changes in the emissions due to different population growth scenarios in the Schwechat and the Raab catchments.

| District | Scenario BAU/BPS | | | Scenario OROK | | |
|---|---|---|---|---|---|---|
| | Change / % | Population | $NO_3^-$-N / $kg \cdot yr^{-1}$ | Change / % | Population | $NO_3^-$-N / $kg \cdot yr^{-1}$ |
| Baden (Schwechat) | 0.0 | 32058 | 39842 | +32.0 | 42317 | 52591 |
| Total Schwechat | 0.0 | 32058 | 39842 | +32.0 | 42317 | 52591 |
| Weiz (Raab) | +7.7 | 56982 | 44918 | -2.0 | 51529 | 40872 |
| Südoststeiermark (Raab) | +2.3 | 32296 | 16537 | -20.4 | 25117 | 12868 |
| Total Raab | +5.7 | 89278 | 61455 | -8.7 | 76646 | 53740 |

**Table A7.** Industrial point source emissions and implemented changes in the emissions at the Raab due to increase in production or relocation of the dominant leather producer.

| Industrial emitter | Relocation of leather industry | | Increase in production | |
| --- | --- | --- | --- | --- |
| | Change / % | $NO_3^-$-N / $kg \cdot yr^{-1}$ | Change / % | $NO_3^-$-N / $kg \cdot yr^{-1}$ |
| Agrana Fruit Austria GmbH | 0.0 | 1029 | 0.0 | 1029 |
| BOXMARK Leder/Feldbach | -100.0 | 0 | 30.0 | 88257 |
| BOXMARK Leder/Jennersdorf | -100.0 | 0 | 30.0 | 36442 |
| Fleischhof Raabtal GmbH | 0.0 | 292 | 0.0 | 292 |
| Johann Titz GmbH | 0.0 | 3774 | 0.0 | 3774 |
| WOLLSDORF Leder | 0.0 | 26572 | 0.0 | 26572 |
| Total | -75.20 | 31667 | 22.6 | 156366 |

**Table A8.** GCM-RCM combinations implemented in the study with their long-term mean annual precipitation sums and long-term mean annual temperatures for the Schwechat and the Raab.

| Model | Schwechat | | Raab | |
|---|---|---|---|---|
| | P / $mm \cdot gr^{-1}$ | T / °C | P / $mm \cdot gr^{-1}$ | T / °C |
| EUR-11_CNRM-CERFACS-CNRM-CM5_RCP45_CLMcom-CCLM4-8-17 | 845.6 | 10.5 | 1103.0 | 12.4 |
| EUR-11_CNRM-CERFACS-CNRM-CM5_RCP85_CLMcom-CCLM4-8-17 | 828.7 | 11.6 | 1075.6 | 13.7 |
| EUR-11_CNRM-CERFACS-CNRM-CM5_RCP45_SMHI-RCA4 | 911.9 | 10.9 | 1118.0 | 12.6 |
| EUR-11_CNRM-CERFACS-CNRM-CM5_RCP85_SMHI-RCA4 | 943.8 | 12.4 | 1091.0 | 14.4 |
| EUR-11_ICHEC-EC-EARTH_RCP45_CLMcom-CCLM4-8-17 | 813.3 | 10.6 | 967.0 | 12.5 |
| EUR-11_ICHEC-EC-EARTH_RCP85_CLMcom-CCLM4-8-17 | 809.2 | 12.1 | 941.5 | 14.4 |
| EUR-11_ICHEC-EC-EARTH_RCP45_SMHI-RCA4 | 915.8 | 11.2 | 1018.4 | 12.9 |
| EUR-11_ICHEC-EC-EARTH_RCP85_SMHI-RCA4 | 939.7 | 12.9 | 1036.1 | 15.1 |
| EUR-11_ICHEC-EC-EARTH_RCP45_KNMI-RACMO22E | 772.7 | 10.9 | 965.0 | 12.6 |
| EUR-11_ICHEC-EC-EARTH_RCP85_KNMI-RACMO22E | 779.0 | 12.6 | 925.6 | 14.6 |
| EUR-11_ICHEC-EC-EARTH_RCP45_DMI-HIRHAM5 | 925.8 | 10.4 | 962.8 | 12.4 |
| EUR-11_ICHEC-EC-EARTH_RCP85_DMI-HIRHAM5 | 912.9 | 12.1 | 976.8 | 14.4 |
| EUR-11_IPSL-IPSL-CM5A-MR_RCP45_IPSL-INERIS-WRF331F | 907.2 | 10.2 | 1046.7 | 13.0 |
| EUR-11_IPSL-IPSL-CM5A-MR_RCP85_IPSL-INERIS-WRF331F | 996.2 | 11.6 | 1202.2 | 14.6 |
| EUR-11_IPSL-IPSL-CM5A-MR_RCP45_SMHI-RCA4 | 899.8 | 11.7 | 1076.8 | 13.7 |
| EUR-11_IPSL-IPSL-CM5A-MR_RCP85_SMHI-RCA4 | 934.6 | 13.5 | 1217.3 | 15.9 |
| EUR-11_MPI-M-MPI-ESM-LR_RCP45_CLMcom-CCLM4-8-17 | 839.1 | 11.5 | 960.5 | 13.6 |
| EUR-11_MPI-M-MPI-ESM-LR_RCP85_CLMcom-CCLM4-8-17 | 867.9 | 13.3 | 913.2 | 15.7 |
| EUR-11_MOHC-HadGEM2-ES_RCP45_SMHI-RCA4 | 974.4 | 11.6 | 1108.5 | 13.6 |
| EUR-11_MOHC-HadGEM2-ES_RCP85_SMHI-RCA4 | 945.0 | 13.6 | 1117.4 | 15.9 |
| EUR-11_MOHC-HadGEM2-ES_RCP45_SMHI-RCA4 | 781.1 | 10.2 | 940.3 | 12.2 |
| EUR-11_MOHC-HadGEM2-ES_RCP85_SMHI-RCA4 | 813.2 | 12.0 | 1021.4 | 14.3 |

*Author contributions.* Christoph Schürz, Karsten Schulz, and Bano Mehdi developed the study framework and prepared the manuscript. Christoph Schürz designed and performed all analyses illustrated in the paper. Bano Mehdi and Christoph Schürz acquired all SWAT model input data, set up the models, and developed the land use change scenarios, Brigitta Hollosi and Christoph Matulla developed the future climate change scenarios, and Alexander Pressl and Thomas Ertl calculated present wastewater emissions and developed the future municipal and industrial emission scenarios.

*Competing interests.* The authors declare no competing interests.

*Acknowledgements.* This work is a result from the project UnLoadC$^3$ (Project No.: KR13AC6K11021) funded by the Austrian Climate and Energy Fund in the $6^{th}$ call of the ACRP program line. The open access publishing was supported by the BOKU Vienna Open Access Publishing Fund. We gratefully obtained time series data of $NO_3^-$-N concentration from Stefan Schuster (TBS Water Consult) and Roland Fuiko (IWR TU Wien) who manage the Raab monitoring data at the Stations Takern II and Neumarkt a.d. Raab. We want to thank Francesca Pianosi, Björn Guse, and the two anonymous reviewers for their detailed comments that helped to substantially improve the paper.

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
