# Peer review of "A comprehensive sensitivity and uncertainty analysis for discharge and nitrate-nitrogen loads involving multiple discrete model inputs under future changing conditions"

_Hydrology and Earth System Sciences, 2018_

## Referee Comment (RC1) · B. Guse (Referee) · 21 Sep 2018

In this manuscript by Schürz et al., a detailed analysis of the impact of scenario simulations on hydrological variables is presented. In a sensitivity analysis, the sensitivities of five groups are separated. These groups are three types of scenarios (land use, point source and climate) and two model-specific groups (model set-up, model parameterisation). In the analysis, the impact of the input variables and of the uncertainty of the selection of scenarios or model characteristics is presented.

[Figure]

Overall, I like the manuscript. However, I see still potential for improvements to increase the understanding of the manuscript.

Major comments:

From my perspective, the readability of the manuscript can be increased by a clear separation in the two major points of the article. First, the impact of the input variables is analysed to show which input variables are more relevant for discharge and nitrate. Second, it is analysed how the selection of a scenario or model characteristic controls the target variables (uncertainty analysis). I think that the article would be easier to understand if these two aspects are clearly separated. This comment is mainly related to abstract, introduction and discussion. In contrast, these two aspects are already clearly separated in the conclusion.

P.11, L: 1: Is it correct that you have identified 43 and 52 behavioural parameter sets out of 100.000 model simulations? If it is true than the number of behavioural parameter sets is rather low. How is than the impact on the sensitivity analysis meaning that most of the parameter sets are unbehavioural?

Figure 4: It is very hard to understand this figure. In my understanding the results from Fig. 3 are shown again and in addition to that the variations evoked by changes in land use or point emissions. Is it maybe better to present this as relative change to the lines in Figure 3? Or only as line and not as coloured area?

Discussion: One idea is to add a table or figure as an overview in the discussion to show which of the five criteria has a dominant impact on discharge and nitrate and which criteria are uncertain. I think that the article would benefit from a clear and easy understandable presentation at the end as a kind of take-home-message. I have in mind a figure which summarize all results in relative values. To understand the overall idea of summary figures see for example Figure 9 in Herman et al., 2013.

Specific comments:
P.1, L. 8: I suggest to modify to: "In impacts studies in two Austrian catchments, ...

P. 1, L.13: I suggest to write "for each catchments" instead of "for both catchments".

P.2, L: 17: I suggest to add: "using a set of different climate input data for hydrological models" at the end of this sentence (or a similar statement).

P.2, L.27: The discussion on equifinality is not well motivated. I miss a sentence to relate both paragraphs.

P.3, L. 5: I suggest to add a sentence at the beginning of the paragraph similar to "Sensitivity analysis can be used to derive the impact of different input variables on hydrological target variables" to make clear why you have selected this method.

P. 6, L. 11: fertilizer

P. 8, L. 5: Please avoid one-sentence-paragraphs

P. 9, L: 14: I suggest to write: "applied a GSA on discharge and nitrate...".

Table 3: Is the sensitivity related to discharge or nitrate or both?

P.15, L. 14: You may add that this result could be expected since the model structure is known to be of higher importance for low flows since high flows are strongly driven by the precipitation (observations).

P. 15, L. 31-34: For me, it seems to be that in Fig. 3, spring is the dominant season in the upper left subplot.

Figure 4: The legend needs to be explained in the figure caption.

P. 19, L. 5: Could you add in which subplot you can see this drastic change?

P. 19, L. 11: Have you an explanation for this?

P. 25, L. 3: I suggest to add "The selection of" before "climate scenarios".

P. 26, section 4.2: You may add a statement similar to "This analysis shows again that

a clear description of the selected scenarios is mandatory for impact studies."

References: Herman, J.D.; Kollat, J.B.; Reed, P.M.; Wagener, T. (2013): From maps to movies: high resolution time-varying sensitivity analysis for spatially distributed watershed models, Hydrol. Earth Syst. Sci., 17, 5109-5125.

---

## Referee Comment (RC2) · Anonymous Referee #2 · 24 Sep 2018

This manuscript by Schürz et al. gives a detailed sensitivity and uncertainty analysis for modelling of hydrology and nitrate export in two medium-size catchments. The sensitivity analysis is elaborated for three groups of input scenarios (land use, point sources, climate) and alternatives of model setup and model parameters. The uncertainty of the modelled flow and nitrate exports is done separately for these five model-specific groups, which enabled evaluations of their influence on the reliability of modelling outputs.

I like the study. It shows a well-designed example how to transparently present mod-

elling results. The methods are sound, using contemporary approaches, and sufficiently described. The results are suitably visualized and a discussed, and support conclusions.

Major comments:

From my view, more credibility can be given to the parametrization of model (which shows very high impact to simulated results and uncertainty) when the selected parameter values that were used in the uncertainty analysis are given, at least in the Appendix.

Specific comments:

p.5, l. 25: Shouldn't be the Raab catchment area 988 km2?

p.19, l. 12-13: I suggest to join the sentences: "While a grouping of the individual climate scenarios with respect to their temperature anomalies shows a more indefinite picture, all climate scenarios simulated an increase in temperature."

---

## Referee Comment (RC3) · Anonymous Referee #3 · 3 Oct 2018

I provide my comments below according to the HESS review criteria. Given some of my major comments below, it does not seem necessary to provide a more detailed line by line annotation at this point.

1. Does the paper address relevant scientific questions within the scope of HESS? Yes. Trying to quantify and attribute uncertainty from various sources in "eco-hydrological" modelling in the context of climate and environmental change.

2. Does the paper present novel concepts, ideas, tools, or data? I found the way

figures 2 to 7 very informative. I particularly found figure 2 very appealing in presenting SA results.

3. Are substantial conclusions reached? Given some of the discussions provided on the methodology below in NO.4, I am not sure if we can say conclusions are substantial.

4. Are the scientific methods and assumptions valid and clearly outlined? I very much liked how the manuscript tries to do a systematic and comprehensive approach, step-by-step, to set up the models, define scenarios, conduct SA/UA experiments, visualize (for better communication of) the results, and reach to conclusions. However, I have some major concerns about some of the methods and tools used in this study that I explain below:

a) Discrete PAWN SA: My most fundamental concern is related the way the main SA with PAWN is performed in this work, which also led to main conclusions in the paper. I strongly feel that the PAWN SA results (Figure 2) is largely impacted by the NUMBER of discrete realizations in each category (Table 4) and not by their CONTENT. In other words, it is intuitive that in this design of SA experiments, by default, the category with a higher number of members will always show a higher influence, because parameters sampled here will naturally have a much higher variability with respect to those categories. And this is exactly what we see in SA results and why some results are rather counter intuitive (e.g. negligible or small influence from land use changes or model setup, and very large influence from Climate and parameters). This is a fundamental issue that needs to be addressed by authors as it is the foundation for all conclusions.

b) Design of Experiments: Authors do a great job particularly in explaining a rather careful and detailed procedure to setup the model, process the required data, define HRUs, and layout future land, pollution, and climate scenarios. This is extensive amount of work. However, I feel that this breadth has caused insufficient scientific depth in places in the manuscript. For example, it is unclear to me why certain various metrics are chosen in the SA analysis with VARS? How are these metrics really differ-

ent from each other from an SA perspective (in particular, NSE and RSR are directly related, so why both are used?), Why this choice is not consistent with the metrics used in the next steps (e.g. what happened to KGE or RSR)? Perhaps strategically reducing some of the metrics can help in a more efficient way of conducting SA and presenting its results (e.g. some of the quantile classes presented in Figure 2 in each signature measure can be removed).

Or for example, what is the scientific reference or justification for the way UA is conducted here at the end using 7K simulations out of all possible combinations? Wouldn't a Latin Hypercube Sampling be a more effective choice than random sampling? These methods and choices (and other similar ones) must to be clearly justified in the manuscript.

5. Are the results sufficient to support the interpretations and conclusions? Please see my comments above in NO.4.

6. Is the description of experiments and calculations sufficiently complete and precise to allow their reproduction by fellow scientists (traceability of results)? No. Details of SA/UA experiments are missing. In particular, I found description of the VARS method somewhat short and there are important details that are missing (a more careful description from the original papers or some of newer applications is recommended). Another very important information that is missing is the ranges used for parameters, and an explanation of how these ranges are determined. These ranges can impact all the SA/UA results. Or it is unclear how parameters are tied to HRUs, and how all different setups, with different NO. of HRUs, in different basins have the same number of parameters (42) when doing SA with VARS?

7. Do the authors give proper credit to related work and clearly indicate their own new/original contribution? Yes.

8. Does the title clearly reflect the contents of the paper? Yes for the most part.

[Figure]

9. Does the abstract provide a concise and complete summary? Yes.

10. Is the overall presentation well-structured and clear? Yes for the most part.

11. Is the language fluent and precise? I feel the language needs to be modified a bit. Both in terms of English grammar (double check usage of "the" and "comma"), and in terms of being scientifically more precise (e.g. using "pollution" instead of "emission"; or using "most influential input" instead of "most relevant"; or page 3 line 4; or page 3 line 26). I recommend a more careful re-view of the manuscript in this regard.

12. Are mathematical formulae, symbols, abbreviations, and units correctly defined and used? Yes.

13. Should any parts of the paper (text, formulae, figures, tables) be clarified, reduced, combined, or eliminated? Some of the quantile classes presented in Figure 2 in each signature measure can be removed.

14. Are the number and quality of references appropriate? Yes.

15. Is the amount and quality of supplementary material appropriate? Yes.

---

## Referee Comment (RC4) · F. Pianosi (Referee) · 8 Oct 2018

The manuscript presents an interesting application of uncertainty and sensitivity analysis to the SWAT model. The aim is to assess the dominant controls of long-term discharge and nitrate-nitrogen load predictions under climate and land use change, while also taking into account the intrinsic uncertainty in the model, i.e. parameter and set-up uncertainty. The analysis is solid and provides interesting insights about the model behaviour. Although the specific findings are only relevant to the investigated model and case studies, their discussion is interesting for the wider community of SWAT users

and in general users of environmental impacts assessment models, as it demonstrates the type of findings yielded by GSA and their implications for the refinement and use of the model. The visual analysis introduced in Figure 4-8 is a simple and yet effective complement to quantitative GSA approaches.

Overall the paper is well structured and well written, and I think it should be accepted for publication.

Below are some points that could be addressed to improve the manuscript clarity before publication.

[1] Language is at times unclear - some examples are given below as Minor points. I also have a general comment about the use of the term "sensitive". The authors use it as interchangeable with "influential" however I find this confusing, because "sensitivity" is an attribute of the output, not of the inputs. I would say that "input x1 is influential on the output" or "the output is sensitive to input x1" but I would not say that "input x1 is sensitive" - this is confusing. Some examples of these unclear occurrences are also given below under Minor points, however if the authors accept my remark they should check the entire manuscript.

[2] The definition and use of the behavioural parameter sets is slightly unclear. I think the confusion started on P. 10 L. 6-7 with the sentence
"For all SWAT model setups of the Schwechat and the Raab catchments we identified non-unique parameter sets that adequately simulated daily observation of discharge and NO3-N loads".
Does it mean that you identified one behavioural parameter set for **each** model setup, or that you identified one behavioural parameter set to be applied **in all** the set-ups? If the former, then how is the dependency between parameterisations and model setups accounted for in the GSA? If the latter, then the underlying assumption is that the same parameter values can effectively represent processes at different aggregation scales (ie for different definitions of the subbasins and HRUs)? This should be clarified.
On a parallel note, I find it interesting that out of 100,000 sampled parameterisations

only 43 and 52 where found behavioural. This is not uncommon in calibration of complex hydrological models but still worth highlighting. It would also be interesting to see whether these behavioural parameterisations are clustered in specific regions of the parameter space or if they are scattered across the sampled ranges, which would indicate a certain amount of interactions between the parameters. This could be illustrated for example through a parallel coordinate plot.

[3] GSA was applied using 7000 samples of the input factors. How was this number chosen? Did the authors checked the adequacy of this sample size? The fact that the ranking based on the sensitivity indices in Figure 2 is confirmed by the visual analysis of Figure 4-8 is reassuring, yet formal methods exist to assess the robustness of the GSA results to the chosen sample size (for example, using bootstrapping confidence intervals as in Sarrazin et al. 2016 or a dummy parameter as in Zadeh et al 2017, both cited in the manuscript). It would be good to include more discussion of this point in the manuscript.

[4] The PAWN method was applied using a sampling scheme different from the one originally presented in Pianosi and Wagener (2015), in order to handle discrete-valued input factors. I understand the idea is to consider as fixed points $x_i^j$ all the possible values that the discrete input factor $x_i$ can take. Hence, for each input factor, the number of fixed points coincides with the number of possible values ($n_i$) that the input can take. If my interpretation is correct, then the text is misleading when it says (P. 13 L. 28) that "a generic random sample of the size $N$ was drawn and subsetted with $N/n_i$ subsets for all $x_i^j$"
as the generic sample is divided into $n_i$ (and not $N/n_i$) subsets. Is this right?
Also, if I understand the strategy correctly, then the inputs with small number of possible values (for instance the land use scenario) are associated with conditional distributions based on a very large number of samples (around $N/n_i$=7,500/2 in the case of land use scenarios), while the inputs with large number of possible values (for instance the parameterisation) are associated with conditional distributions based on much smaller

number of samples (around 7,000/43). Do you think using such different sample sizes could have had an impact on the estimation of the KS values and hence of the PAWN sensitivity indices?

Finally, a new sampling strategy was recently proposed for PAWN (Pianosi and Wagener, 2018). While this new strategy is still designed for continuous inputs, and hence could not be used here, it would be good to mention its existence for readers who may want to apply PAWN in the future (as for the case of continuous inputs this would be recommended over the strategy in the 2015 paper).

[5] I think the discussion in Section 4.2 is interesting but potentially slightly misleading. The authors clarify that "several assumptions were made in the development of scenarios that are highly subjective". I understand the importance of highlighting the subjectivity inherent in the scenario definition *if the goal of this study was to make projections* of the future evolution of the two catchments. However, this is not the objective when doing GSA. GSA answers the question: "how much output variation do we get if we vary the inputs *within certain ranges*?" The answer yielded by GSA (i.e. the sensitivity indices, the input ranking, etc.) is certainly conditioned upon the chosen ranges, however this is "intrinsic" to the question asked, regardless of how the choice is made - be it an "objective" calibration exercise (as done for the parameterisations) or a "storylines" approach. In other words, I think the point is to justify why certain scenarios are considered for the impacts assessment study; once they have been selected for that purpose, it follows that they would be used in the GSA too if one wants to know their relative influence with respect to other input factors of the model. So, I do not agree with the sentence (P. 26 L. 13-14) "For the SA of the simulated variables the diversity of the developed scenarios is essential.": diversity may be important for the impacts assessment (is it?) but not necessarily for the GSA. If a limited set of scenarios were selected for the impacts assessment, I would use that set for the GSA even if it is not diverse.

MINOR POINTS

P. 1 L. 15: "scenario inputs" should be "input scenarios"

P. 2 L. 5: "the precipitation of the climate scenarios" sounds a bit odd, maybe "precipitation projections"

P. 3 L. 3-4: "An assessment is only as good as the dominant contributors of uncertainty in such a modeling chain." Unclear. Something seems to be missing in this sentence: an environmental impacts assessment is only good if dominant contributors of uncertainty are... what? identified? removed? ...?

P. 3 L. 11-12: "model computations" should be "model evaluations" (or "runs" or "executions")

P. 3 L. 19: "Most applications utilize GSA to identify and rank continuous model parameters". Unclear: GSA does not "identify parameters" at most "identify influential parameters"

P. 3 L. 21: "Although," Comma should be removed

P. 3 L. 26-27: "An OAT analysis however presumes linear models and non-correlated inputs". Not sure OAT requires a linear model, for instance the Morris method uses a OAT approach and yet is typically applied to nonlinear models. More generally, why should GSA be applied to a linear model at all? If the model is linear than the effect of each input on the model output is simply proportional to the input variation, no need to do GSA to know that.

P. 4 L. 4: "complex". Unclear. What is the definition of a "complex" input?

P. 4 L. 5: "No study is known to us that takes advantage of GSA in the scope of environmental impact studies." What is the definition of "environmental impact studies" here? I would say that GSA has been applied to such studies before, e.g. Anderson et al (2014); Butler et al (2014); Le Cozannet et al (2015)

P. 4 L. 13-16: Very long sentence, consider splitting into two.

P. 8 L. 3: "Although," Comma should be removed.

P. 8 L. 14-15 "The SWAT model setups for the Raab and the Schwechat involved decisions for the selected number of subbasins of a model setup and the definition of the HRUs." Convoluted sentence.

P. 8 L. 15: "Both modifications": which modifications? Unclear

P. 9 L. 2: "involving". Unclear. Maybe "which requires"?

P. 9 L. 11: "to define of the thresholds". Remove "of"? In general, the entire sentence is a bit unclear. How is the "aggregation error" defined? Error in which variable, and with respect to what "correct" value?

P. 9 L. 14: "In a pre-analysis step," In the GSA literature, this kind of "pre-analysis step" is often called a "screening" analysis, as it aims at screening out the non-influential parameters. Maybe worth mentioning the term as it would be familiar to many readers.

P. 9 L. 14: "relevant parameters". Relevant to what? Maybe better "influential"

P. 9 L. 21: "FDCs". Explain the acronym

P. 13 L. 5: "To identify the impact of" maybe better "To measure the relative importance of"

P. 13 L. 7: "PAWN involves". Unclear what "involves mean. Maybe better "PAWN uses"

P. 13 L. 11: "the sensitivity of a model input x for a target variable y". Sensitivity is an attribute of the output, not the input. I would rephrase as "the sensitivity of a target variable y to a model input x".

P. 13 Eq. (1) and (2). The mathematical notation could be made clearer. I find it odd that in Eq. (1) KS takes as subscript the index of the fixed point ($j$) while its argument remains the generic input $x_i$. This choice also makes it more difficult to understand how maximisation occurs in Eq. (2). I think using the notation $KS(x_i^j)$ in both equations would make things much clearer.

P. 13 L. 23: "possible states". Why "states"? The term was never used with this meaning before. I would rather say "possible values".

P. 13 L. 24: "a lower sensitivity of the input $x_i$ on the target variable y". Again, rephrase as either "a lower sensitivity of the target variable y to the input $x_i$" or "a lower influence of the input $x_i$ on the target variable y".

P. 13 L. 28: "subsetted with" Not sure "subset" can be used as a verb. Maybe better "divided into"

P. 13 L. 29. "... were used for the sensitivity assessment". I would link this to the

mathematical notation just introduced above and say: "... were used as target variable y".

P. 14 L. 10-12: "In this study, we consider all execute model setups to be plausible..." I do not understand this clarification. What other approach would have been possible? To discard some simulations because deemed not plausible? And how would you define then what is plausible and what is not? Please clarify.

P. 14 L. 17: "low number of each input". Unclear. Do the authors mean "low number of inputs" (i.e. 5 inputs) or "low number of possible values taken by each input"?

P. 14 L. 24-28. This sounds like a repetition of what just said in the methodology section, I do not think is needed. I would rather use this opportunity to explain how to read Figure 2 (what is the difference between the panels and how to read each circle plot).

P. 15 L. 17: "highly sensitive" replace by "highly influential"

P. 15 L. 25-26: "their overall sensitivities follow the general trend of the climate scenarios to a large extent". Unclear, please rephrase.

P. 15 L. 33: "difference that is visible for the two". Unclear, do you mean "difference that is visible between the two"?

P. 15 L. 33-34: "how the reference period relates to the uncertainty bands in amplitude". Unclear what this means.

P. 16, caption of Fig. 2: "Model input sensitivities for signature measures". Replace by "sensitivities of signature measures to model inputs". And later on "sensitivities of" should be replaced by "sensitivities to"

P. 17, text and Figure 3: what does "specific discharge" mean? Why "specific"?

P. 17 L. 6: "show a difference". Does this mean "show an increase"? If so, I would use "increase", it makes it easier for the reader to follow.

P. 19 L. 12: "While a grouping...". Remove "While".

P. 24 L. 20: caused future land use change" Maybe "caused **by** future..." ?

P. 26 L. 31-32: "The application of sampling strategies for SA usually do not account for the circumstances that one model input constrains any other model input". I do not fully agree. There is an increasing literature on GSA methods applicable to the case of

dependent inputs, see for instance Mara and Tarantola (2012; 2017).

P. 27 L. 17-18: "by a factor of up to 5... up to 8". Do these numbers come out of a comparison of PAWN indices? If so, I am not sure I would draw such quantitative comparison. PAWN indices are (maximum) KS values: what is the practical interpretation of "a factor of 5" between KS values? I find it difficult to imagine.

P. 28 L. 3: "the lack of tool that allow the practitioners access to such methods". Not sure I understand what the authors mean here. Several GSA software tools are available (some are reviewed for example in Pianosi et al 2015). So what is the problem here? That they are not "friendly" enough for practitioners to use them? Or that they are not sufficiently tailored to SWAT applications? Pls clarify.

REFERENCES

Anderson, B., Borgonovo, E., Galeotti, M., Roson, R., 2014. Uncertainty in climate change modeling: can global sensitivity analysis be of help? Risk Anal. 34 (2).

Butler, M.P., Reed, P.M., Fisher-Vanden, K., Keller, K., Wagener, T., 2014. Identifying parametric controls and dependencies in integrated assessment models using global sensitivity analysis. Environ. Model. Softw. 59.

Le Cozannet et al (2015) Evaluating uncertainties of future marine flooding occurrence as sea-level rises, Environmental Modelling Software, 73.

Mara and Tarantola, 2012, Variance-based sensitivity indices for models with dependent inputs, Reliability Engineering and System Safety

Tarantola and Mara, 2017, VARIANCE-BASED SENSITIVITY INDICES OF COMPUTER MODELS WITH DEPENDENT INPUTS: THE FOURIER AMPLITUDE SENSITIVITY TEST, International Journal for Uncertainty Quantification

Pianosi and Wagener (2018), Distribution-based sensitivity analysis from a generic input-output sample, Environmental Modelling Software

Pianosi, F., Sarrazin, F., Wagener, T. (2015), A Matlab toolbox for Global Sensitivity

[Figure]

Analysis, Environmental Modelling Software, 70.
* * *

---

## Author Comment (AC1) · 9 Nov 2018

**Reply to the reviewer comments RC1: 'Review of the manuscript by Schürz et al.' by**

**Björn Guse**

Summary

*In this manuscript by Schürz et al., a detailed analysis of the impact of scenario simulations on hydrological variables is presented. In a sensitivity analysis, the sensitivities of five groups are separated. These groups are three types of scenarios (land use, point source and climate) and two model-specific groups (model set-up, model parameterisation). In the analysis, the impact of the input variables and of the uncertainty of the selection of scenarios or model characteristics is presented.*

*Overall, I like the manuscript. However, I see still potential for improvements to increase the understanding of the manuscript.*

We would like to thank Björn Guse for his very constructive review and the valuable comments to improve the quality of the manuscript. We appreciate the positive feedback on the manuscript. In the following, we addressed each comment individually. The comments made by Björn Guse are printed in *serif, italic font*. Our replies to the comments are written in black, non serif font and our suggestions to revise the manuscript according to a comment are highlighted with the colors blue for insertions and red for deletions.

Major comments

*From my perspective, the readability of the manuscript can be increased by a clear separation in the two major points of the article. First, the impact of the input variables is analysed to show which input variables are more relevant for discharge and nitrate. Second, it is analysed how the selection of a scenario or model characteristic controls the target variables (uncertainty analysis). I think that the article would be easier to understand if these two aspects are clearly separated. This comment is mainly related to abstract, introduction and discussion. In contrast, these two aspects are already clearly separated in the conclusion.*

We strived for a consistent structure in the manuscript by presenting the two separate blocks: 1) the sensitivity analysis of the model inputs and the model setup and parametrization; 2) and the uncertainty analysis together with the visual analysis.
With your comment in mind, we see the equivocality in the outline of the manuscript. In the current form the manuscript outline can be interpreted as if the performed sensitivity analysis was the actual focus of this work and the visual analysis of the uncertainties are a mere by-product (e.g. p.1L15ff in Abstract, p.5L7-16 in the Introduction).
We agree, that emphasizing the value of the visual analysis and treating it as an individual part of this work increases structure and consistency of the manuscript (e.g. being consistent to section 2.6 p.12L18ff where we already clearly separate the two goals of the performed analyses).

We suggest the make following changes in the updated version of the manuscript:
Exemplary suggested changes in the abstract, p.1L15ff:

The analysis of the 7000 generated model combinations of both case studies had two main goals; i) to identify the dominant controls on the simulation of discharge and $NO_3^- - N$ loads in the two case studies and ii) to assess how the considered inputs control the simulation of discharge and $NO_3^- - N$ loads.  To  assess the impact of the input scenarios , the model setup and the parametrization on the simulation of discharge and $NO_3^- - N$ loads we employed methods of global sensitivity analysis (GSA). The uncertainties in the simulation of discharge and $NO_3^- - N$ loads that resulted from the 7000 SWAT model combinations were evaluated visually. We present approaches for the visualization of the simulation uncertainties that proved to be a powerful diagnostic tool in this study to assess how the analyzed inputs affected the simulations.

Following the suggested changes for the abstract we suggest to apply the same ideas for revising the introduction. Here we will focus on the sections p.3L21-29 and p.4L7-16 in particular.
Concerning the discussion we would prefer to keep the approach we currently follow in the manuscript, in which we discussed our findings and related them to other literature. Clearly separating the findings concerning the sensitivity analysis and the uncertainty analysis is difficult to facilitate for the larger part of the discussion. A separation might lead to a lot of repetition in the discussion.

*P.11, L: 1: Is it correct that you have identified 43 and 52 behavioural parameter sets out of 100.000 model simulations? If it is true than the number of behavioural parameter sets is rather low. How is than the impact on the sensitivity analysis meaning that most of the parameter sets are unbehavioural?*

The decision whether a parameter set is considered to be behavioral is highly subjective, as the objective criteria that are applied to evaluate the model simulations and the thresholds for these objective criteria that define a simulation as "good" or "insufficient" are individual decisions. The decisions made in the presented work are listed on p.10L6-10.
We agree, that a different definition of a behavioral parameter set would affect the influence that the model parametrization has on the analyzed model outputs. The effect of the assumptions made in such an impact assessment were therefore discussed in section 4.2. Thus, the low number of behavioral parameter sets does not per se affect the sensitivity of the model outputs on the model parametrization.
The low number of behavioral parameter sets in this specific case results from the study design. A model parameter set was considered as a behavioral parameter set, if the simulations performed with **all** model setups using that parameter set fulfilled the applied objective criteria (which means a model parameter set implemented in the Raab case study had to meet the thresholds for the objective criteria in all six implemented model setups).

This design decision may influence the resulting sensitivities of the model outputs, as the impact of the model setup and the model parametrization combined can be greater than the effect resulting from the illustrated study design (All individual model setups together with their model parametrizations may result in behavioral model/parameter setups, that are not considered here). The implemented setup however, isolates the effect of the model setups and the effect of the model parametrizations for model setups that were calibrated for a reference period and are applied for future changing conditions. In the context of an environmental impact study, to assess their individual effects is highly relevant in our opinion.

As a result of your comment (and similar comments in other reviews) we see a requirement to clarify the the evaluation of the parameter sets on p.10L6ff.

*Figure 4: It is very hard to understand this figure. In my understanding the results from Fig. 3 are shown again and in addition to that the variations evoked by changes in land use or point emissions. Is it maybe better to present this as relative change to the lines in Figure 3? Or only as line and not as coloured area?*

The Figures 4 to 8 follow the same pattern in the analysis that they illustrate. The Figures 4 to 8 indeed present the results shown in Figure 2 in a modified way. While Figure 3 shows the uncertainty bands of the 7000 simulations performed in each case study implementing the different combinations of input scenarios and model setups, the following figures separate the resulting uncertainty bands with respect to the discrete realizations of the individual model inputs and model setups. During the compilation of the manuscript we tested different ways to communicate the information we wanted to convey (e.g. plot all 7000 simulations as lines, analyze relative changes, etc.). We concluded however that the selected visualizations were the most suitable ones that supported our findings best. Thus, we prefer to remain with the presented figures. We see however the need to clarify the explanations regarding the Figures 3 to 8.

*Discussion: One idea is to add a table or figure as an overview in the discussion to show which of the five criteria has a dominant impact on discharge and nitrate and which criteria are uncertain. I think that the article would benefit from a clear and easy understandable presentation at the end as a kind of take-home-message. I have in mind a figure which summarize all results in relative values. To understand the overall idea of summary figures see for example Figure 9 in Herman et al., 2013.*

We highly appreciate this comment and thank you for the link to the publication by Herman et al. (2013). Herman et al. (2013) used the summary figure as a very effective tool to summarize their findings. We were discussing how to implement this tool to summarize our findings in the manuscript. So far however, we were not able to come up with a good solution that would add value to the manuscript and facilitate interpretation for the reader. Thus, unless we come up with an appropriate illustration during the revision of the manuscript, we prefer to not add an additional figure.

**Specific comments**

*P.1, L. 8: I suggest to modify to: "In impacts studies in two Austrian catchments, ...*

We prefer your suggestion over the phrase in the manuscript. The text will be changed accordingly.

*P. 1, L.13: I suggest to write "for each catchments" instead of "for both catchments".*

Together with other changes the section p.L11-14 will be updated as follows:
We developed scenarios of future changes for land use, point source emissions, and climate. The developed input  scenarios were implemented in  SWAT model setups with different spatial aggregations and employing different model parametrizations that were able to adequately reproduce historical observations of discharge and $NO_3^- - N$ loads.  In total 7000 combinations of scenarios and model setups were used to  simulate daily discharge and $NO_3^- - N$ loads at the catchment outlets of each catchment.

*P.2, L: 17: I suggest to add: "using a set of different climate input data for hydrological models" at the end of this sentence (or a similar statement).*

In this particular section we wanted to keep the statement more general (the statement is also true for land use change, or any other change process expressed with discrete scenarios). Thus, we prefer to keep the general phrase with the following example of climate change scenarios, as written.

*P.2, L.27: The discussion on equifinality is not well motivated. I miss a sentence to relate both paragraphs.*

We suggest the following modification of this section in the updated version of the manuscript:
To simulate the development of hydrological variables under changing conditions, the developed scenarios are implemented in hydrological models that are calibrated for historic conditions. Yet, often different model setups and different sets of parameters in a model can perform equally well to reproduce historical observations of the variables of interest. Equifinality is a well-known issue in hydrologic modeling that has been extensively addressed in the literature...

*P.3, L. 5: I suggest to add a sentence at the beginning of the paragraph similar to "Sensitivity analysis can be used to derive the impact of different input variables on hydrological target variables" to make clear why you have selected this method.*

The sentence will be added accordingly.

*P. 6, L. 11: fertilizer*

This will be corrected accordingly.

*P. 8, L. 5: Please avoid one-sentence-paragraphs*

The sentence will be added to the previous paragraph.

*P. 9, L: 14: I suggest to write: "applied a GSA on discharge and nitrate...".*

To consider your suggestion and the suggestions made by other reviewers on this sentence it will be changed as follows:
In a  parameter screening, we applied a GSA to the simulations of discharge and $NO_3^- - N$ at the catchment outlets using all SWAT model setups  to identify the relevant model parameters.

*Table 3: Is the sensitivity related to discharge or nitrate or both?*

We appreciate this comment and think that this is valuable information for the reader. Thus, we suggest to modify Table 3 and differentiate between parameters that were influential for discharge related processes and $NO_3^- - N$ related processes.

*P.15, L. 14: You may add that this result could be expected since the model structure is known to be of higher importance for low flows since high flows are strongly driven by the precipitation (observations).*

We addressed this issue already in the discussion to some extent. We suggest however to stress this issue more and to clarify this point in the discussion. In contrast to your statement, the study design (that tried to assess the individual effects of the model setup and the parametrization) clearly show that the model setup has a stronger influence on large and medium discharges, whereas the model parametrization greatly affects the low flow.

*P. 15, L. 31-34: For me, it seems to be that in Fig. 3, spring is the dominant season in the upper left subplot.*

We agree with your observation that spring is little more dominant in Schwechat. Thus, we will mention this fact as well.

*Figure 4: The legend needs to be explained in the figure caption.*

We agree that the used abbreviations are not self explaining in this figure. Hence, we will add   an explanation of the abbreviations in the figure caption.

*P. 19, L. 5: Could you add in which subplot you can see this drastic change?*

Actually, that finding is supported by all subplots. The anomalies in precipitation affect the simulated long term monthly averages of discharge and $NO_3^- - N$ loads as well as the different segments of the illustrated flow duration curves of both catchments.

*P. 19, L. 11: Have you an explanation for this?*

We think that the precipitation anomalies explain these findings, to a large extent. Increases in mean annual precipitation increase the discharge and $NO_3^- - N$ loads, while a simulated reduction of mean annual precipitation in the future results in a reduction of discharge and

$NO_3^- - N$ loads. This explanation is, in our opinion, provided in the text of the manuscript. We suggest however to revise this section and try to specify the statement more precisely.

*P. 25, L. 3: I suggest to add "The selection of" before "climate scenarios".*

This will be changed accordingly.

*P. 26, section 4.2: You may add a statement similar to "This analysis shows again that a clear description of the selected scenarios is mandatory for impact studies."*

We appreciate this comment and such a statement will be added to the text.

References

*Herman, J.D.; Kollat, J.B.; Reed, P.M.; Wagener, T. (2013): From maps to movies: high resolution time-varying sensitivity analysis for spatially distributed watershed models, Hydrol. Earth Syst. Sci., 17, 5109-5125.*

---

## Author Comment (AC2) · 9 Nov 2018

**Reply to the reviewer comments RC2: 'modelling of hydrology and nitrate export from catchment' by Anonymous Referee #2**

*This manuscript by Schürz et al. gives a detailed sensitivity and uncertainty analysis for modelling of hydrology and nitrate export in two medium-size catchments. The sensitivity analysis is elaborated for three groups of input scenarios (land use, point sources, climate) and alternatives of model setup and model parameters. The uncertainty of the modelled flow and nitrate exports is done separately for these five model-specific groups, which enabled evaluations of their influence on the reliability of modelling outputs.*

*I like the study. It shows a well-designed example how to transparently present modelling results. The methods are sound, using contemporary approaches, and sufficiently described. The results are suitably visualized and a discussed, and support conclusions.*

We would like to thank the Anonymous Referee #2 for their positive and supportive feedback on this manuscript. In the following, we addressed each comment made by Anonymous Referee #2. The initial comments made are printed in *serif, italic font*. Our replies to the comments are written in black, non serif font and our suggestions to revise the manuscript according to a comment are highlighted with the colors blue for insertions and red for deletions.

Major comments

*From my view, more credibility can be given to the parametrization of model (which shows very high impact to simulated results and uncertainty) when the selected parameter values that were used in the uncertainty analysis are given, at least in the Appendix.*

Based on this comment and comments made by other reviewers, we propose to add the following information to provide further detail on the model parameters used.

To show a clustering of model parameter values of the selected parameters and to identify parameter interactions we add the following figure in the Appendix of the manuscript:

[Figure]

*Catchment* ● Schwechat ● Raab

Figure caption:

Coordinate plot of the 43 and 52 behavioral SWAT model parameters that were used with the model setups of the Schwechat and the Raab, respectively. Each panel illustrates the connection of two model parameters for the Schwechat in red (below the diagonal) and the Raab in blue (above the diagonal). The x and y axes of each panel show the range of the respective parameter plotted along the x or y dimension. The corresponding parameter ranges for all illustrated parameters are provided in Table XX (Reference to table below).

Due to the limited space in the figure we avoided plotting axes and axis labels. The figure however illustrates the clustering and interaction of model parameters. We additionally suggest to add parameter ranges and the type of change of the model parameters in an additional table:

Table caption:
SWAT model parameters calibrated in the model setups of the Schwechat and the Raab catchments. The type of change indicates whether the model parameters were replaced by absolute values, modified by adding absolute values to the predefined model parameters or, changed by a relative fraction of the predefined model parameter. Illustrated are the initial ranges of the model parameters and the ranges of the final behavioral parameter sets of the model setups of the Schwechat and the Raab catchments.

| Parameter | Type of change | Initial parameter range | Parameter change range Schwechat | Raab |
|---|---|---|---|---|
| SFTMP | replace value | [-1.00, 1.00] | [-0.69, 0.93] | [-0.98, 0.88] |
| SNOCOVMX | replace value | [100.0, 500.0] | [0.9, 177.0] | [100.8, 447.5] |
| SNO50COV | replace value | [0.20, 0.50] | [0.21, 0.49] | |
| SURLAG | replace value | [0.00, 0.50] | [0.02, 0.99] | [0, 0.1] |
| GW_DELAY | replace value | [0.0, 300.0] | [5.5, 25.0] | [2.1, 283.3] |
| GW_REVAP | replace value | [0.02, 0.20] | [0.05, 0.15] | [0.02, 0.20] |
| GWQMN | replace value | [0, 3000] | [566, 2472] | [109, 2925] |
| RCHRG_DP | replace value | [0.00, 1.00] | [0.31, 0.69] | [0.13, 0.97] |
| SOL_K | relative change | [-0.90, 10.00] | [0, 0.97] | [-0.79, 9.76] |
| SOL_AWC | relative change | [-0.90, 2.00] | [-0.86, 1.49] | [0, 1.98] |
| SLSOIL | replace value | [0.0, 150.0] | [0.9, 27.6] | [14.7, 148.2] |
| CANMX | relative change | [0.00, 0.25] | [0.34, 2.40] | |
| ESCO | replace value | [0.00, 0.90] | [0.05, 0.90] | [0.05, 0.89] |
| LAT_TTIME | replace value | [0.0, 180.0] | [0.8, 6.8] | [5.5, 176.3] |
| OV_N | absolute change | [-0.09, 0.60] | | [0.07, 0.58] |
| CNOP_till | relative change | [-0.20, 0.10] | [-0.29, -0.06] | [-0.18, 0.01] |
| RCN | replace value | [2.00, 10.00] | [5.05, 9.97] | [2.3, 8.45] |
| NPERCO | replace value | [0.00, 1.00] | [0.24, 0.99] | [0.18, 0.7] |
| CDN | replace value | [0.00, 1.50] | [0.01, 1.44] | |
| SDNCO | replace value | [0.00, 0.50] | [0.02, 0.49] | |

Specific comments

*p.5, l. 25: Shouldn't be the Raab catchment area 988 km2?*

Thank you for identifying that typo. According to Table A2 p.32 the total delineated area of the Raab catchment is 98815.9 ha. The value in the text on p.5 L25 will be changed accordingly from  to 988 km2.

*p.19, l. 12-13: I suggest to join the sentences: "While a grouping of the individual climate scenarios with respect to their temperature anomalies shows a more indefinite picture, all climate scenarios simulated an increase in temperature."*

This will be changed accordingly.

---

## Author Comment (AC3) · 9 Nov 2018

**Reply to the reviewer comments RC3: 'Review of the manuscript by Schürz et al.' by**

**Anonymous Referee #3**

*I provide my comments below according to the HESS review criteria. Given some of my major comments below, it does not seem necessary to provide a more detailed line by line annotation at this point.*

We want to thank the Anonymous Referee #3 for their detailed review of the manuscript and the valuable comments made to improve the quality of the manuscript. In particular the critical comments on the methodology helped us to reassess the results and the conclusions drawn in this work. The comments made by Anonymous Referee #3 are printed in *serif, italic font* below. Our replies to the comments are written in black, non serif font and our suggestions to revise the manuscript according to a comment are highlighted with the colors blue for insertions and red for deletions.

*1. Does the paper address relevant scientific questions within the scope of HESS? Yes. Trying to quantify and attribute uncertainty from various sources in "eco-hydrological" modelling in the context of climate and environmental change.*

We appreciate the positive feedback on the relevance of our manuscript.

*2. Does the paper present novel concepts, ideas, tools, or data? I found the way figures 2 to 7 very informative. I particularly found figure 2 very appealing in presenting SA results.*

We appreciate the positive feedback on the visualization of our findings.

*3. Are substantial conclusions reached? Given some of the discussions provided on the methodology below in NO.4, I am not sure if we can say conclusions are substantial.*

A detailed reply to the specific comments can be found below (4 a) and b)).

*4. Are the scientific methods and assumptions valid and clearly outlined? I very much liked how the manuscript tries to do a systematic and comprehensive approach, step-by-step, to set up the models, define scenarios, conduct SA/UA experiments, visualize (for better communication of) the results, and reach to conclusions. However, I have some major concerns about some of the methods and tools used in this study that I explain below:*

*a) Discrete PAWN SA: My most fundamental concern is related the way the main SA with PAWN is performed in this work, which also led to main conclusions in the paper. I strongly feel that the PAWN SA results (Figure 2) is largely impacted by the NUMBER of discrete realizations in each category (Table 4) and not by their CONTENT. In other words, it is intuitive that in this design of SA experiments, by default, the category with a higher number of members will always show a higher influence, because parameters sampled here will naturally have a much higher variability with respect to those categories. And this is exactly what we see in SA results and why some results are rather counter intuitive (e.g. negligible or small influence from land use changes or model setup, and very large influence from Climate and parameters). This is a fundamental issue that needs to be addressed by authors as it is the foundation for all conclusions.*

We disagree with the argument that the influence of a model input / model setup depends on the number of realizations of that respective input/setup. Indeed, the model parametrization and the climate scenarios had the strongest impact on most of the analyzed processes and were represented by a substantially larger number of realizations compared to the other inputs. A counterexample to the statement that the sensitivity is per design impacted by the number of realizations of an input is illustrated by the influence of the point source scenarios in the Raab catchment for medium and low nitrate-nitrogen ($NO_3^- - N$) loads and $NO_3^- - N$ concentrations for medium and low discharges in this study. The calculated PAWN indices for these measures were substantially larger for the point sources compared to, for instance, the climate scenarios. Yet, only four point source scenarios (and only two industrial emission scenarios that eventually were responsible for the large sensitivities) were used, while 22 climate simulations were implemented.

We want to clearly point out however, that the number of discrete realizations of a model input can affect the calculation of a sensitivity index indirectly. In the case of the PAWN index a distance is calculated between the unconditional and the conditional cumulative distribution function (CDF) of a target variable (Pianosi and Wagener (2015) for example suggest to use the Komogorov-Smirnov test statistics). The unconditional CDF can also be estimated from all simulation that were performed (where all model inputs are perturbed), while to estimate the conditional CDF only simulations are used that used one discrete realization of the input of interest (this means all other inputs are perturbed, while the input of interest is kept constant). The distance measure is calculated for all realizations of a model input accordingly. The calculated distances for all conditional CDFs (keeping the model input constant at every respective realization) do have a certain distribution. To infer the PAWN sensitivity index, the calculated distances are summarized employing any summary statistics (Pianosi and Wagener (2015) for example suggests to use the median or the maximum). The choice of summary statistics can however strongly affect the comparability of the calculated sensitivity indices of the individual model inputs if the distance measure distributions for the model inputs substantially differ. As a consequence, we employed the maximum statistics in this study, as we were primarily interested in the maximum possible

influence an input has on an analyzed target variable. Different summary statistics, but also different methods for global sensitivity analysis (GSA, e.g. the method of Sobol (1993) that analyzes an average influence of a model input) were tested and evaluated during the compilation of this study. The outlined effects were observed in these analyses (yet not shown in this manuscript).

Further, the calculated sensitivities are well supported by the analysis of the simulation uncertainties. Inputs that showed a large influence on an analyzed process also showed a strong effect on the simulation uncertainty bands of that respective process.

Finally, we disagree with the statement that the negligible or small influence from land use changes or model setup are counter intuitive findings. In our opinion these findings were substantially discussed in section 4.1. Other literature cited in section 4.1 strongly supports the findings (e.g. Wagner et al. (2017), Guse et al. (2015), Mehdi et al. (2015a, 2015b), or Bieger et al. (2013) for the impact of land use change, or Jha (2014) for the model setups).

*b) Design of Experiments: Authors do a great job particularly in explaining a rather careful and detailed procedure to setup the model, process the required data, define HRUs, and layout future land, pollution, and climate scenarios. This is extensive amount of work. However, I feel that this breadth has caused insufficient scientific depth in places in the manuscript. For example, it is unclear to me why certain various metrics are chosen in the SA analysis with VARS? How are these metrics really different from each other from an SA perspective (in particular, NSE and RSR are directly related, so why both are used?), Why this choice is not consistent with the metrics used in the next steps (e.g. what happened to KGE or RSR)? Perhaps strategically reducing some of the metrics can help in a more efficient way of conducting SA and presenting its results (e.g. some of the quantile classes presented in Figure 2 in each signature measure can be removed).*

It is correct that different measures were used as objective criteria in the GSA to identify influential model parameters and in the model calibration (identification of behavioral parameter combinations). The purpose of the GSA was to screen the model parameters. This screening had an inclusive character, which means that the parameter had to be influential for at least one of the selected criteria. Consequently, the similarity of criteria did not affect the results of the parameter screening (if the measures are similar then the same parameters are influential for these objective criteria.). Contrary, the selection of behavioral parameters was exclusive. Thus, only criteria were used that describe the aspects of a simulated time series that we explicitly wanted to evaluate. In the selection of the criteria for the model calibration we referred to literature such as Pfannerstill et al. (2014).

We agree that the measures NSE and RSR are strongly related in their calculation. Yet, both measures differed completely in their application in this study. While the NSE was applied to the simulated and observed times series of a variable, the RSR was applied to various segments of the flow duration curves (FDC). Thus, the resulting NSE values also accounted for the timing of simulated values of a variable, whereas the RSR values of the FDC segments did only account for the distribution of simulated values of a variable letting aside the temporal occurrence of a value.

We fully understand that the Fig. 2 can overwhelm the reader, as we try to present a lot of information in one figure. Nevertheless, we think that all segments of a FDC characterize different processes of the water or the nutrient cycles (in this case). Further, the large number of analyzed segments of the FDCs visually support the gradual shifts of sensitivities

of a target variable from one model input to others. There is a chance that this information is lost, when removing too many of the analyzed FDC segments from the figure.

*Or for example, what is the scientific reference or justification for the way UA is conducted here at the end using 7K simulations out of all possible combinations? Wouldn't a Latin Hypercube Sampling be a more effective choice than random sampling? These methods and choices (and other similar ones) must to be clearly justified in the manuscript.*

As briefly mentioned above, other methods for GSA were tested as well (while not shown in the manuscript). A preceding analysis employed the Sobol method (Sobol, 1993) for GSA using a sampling design proposed by Saltelli (2002) that requires $N(k+2)$ samples, where $N$ is the "base sample" (Saltelli, 2008) that was defined with 1000 in this study and k is the number of inputs (in this case 5).

As we identified issues with the average sensitivity that is expressed by the sensitivities calculated using the Sobol method (see also the reply 4a)) we utilized the random sample that was drawn for the Sobol method to calculate PAWN indices. Pianosi and Wagener (2018) outline how to estimate PAWN indices from any generic sampling. For this study the proposed concept was applied to discrete model inputs in this study.

We see however from this and other reviews on that matter, that the sampling and the confidence in the GSA results require greater attention in the manuscript. Thus, we suggest to revise the section of the input factor sampling in the revised version of the manuscript. Further, as proposed by Francesca Pianosi in her review, we plan to perform a bootstrapping (as presented in Sarrazin et al. (2016)) to calculate confidence intervals for the PAWN indices. This will greatly improve the results of the manuscript.

*5. Are the results sufficient to support the interpretations and conclusions? Please see my comments above in NO.4.*

We tried to clarify issues raised concerning the methodology that was applied to derive the results illustrated in the manuscript in 4 a) and b). Please find our replies to these comments below the respective sections 4 a) and b).

*6. Is the description of experiments and calculations sufficiently complete and precise to allow their reproduction by fellow scientists (traceability of results)? No. Details of SA/UA experiments are missing. In particular, I found description of the VARS method somewhat short and there are important details that are missing (a more careful description from the original papers or some of newer applications is recommended). Another very important information that is missing is the ranges used for parameters, and an explanation of how these ranges are determined. These ranges can impact all the SA/UA results. Or it is unclear how parameters are tied to HRUs, and how all different setups, with different NO. of HRUs, in different basins have the same number of parameters (42) when doing SA with VARS?*

We agree that the explanations concerning the parameter sensitivity analysis are rather short, as we intended to focus on the actual sensitivity study. Yet, you are right that the working steps in the parameter sensitivity analysis and the model parametrization affect the results of the following study.

We suggest to elaborate the parameter sensitivity analysis with greater detail. For the updated version of the manuscript adding a table is planned that provides information on the

initial parameter boundaries, the boundaries of the final behavioral parameter sets and the type of change that was applied to the model parameters (whether the parameters were replaced by a single value globally or the spatially distributed parameter field was changed by a fraction of the parameter value or changed by adding/subtracting an absolute value).

*7. Do the authors give proper credit to related work and clearly indicate their own new/original contribution? Yes.*

Thank you

*8. Does the title clearly reflect the contents of the paper? Yes for the most part.*

Based on the assessment of the manuscript title we see no possibility to improve the title to more precisely reflect the contents of the manuscript.

*9. Does the abstract provide a concise and complete summary? Yes.*

Thank you

*10. Is the overall presentation well-structured and clear? Yes for the most part.*

Thank you

*11. Is the language fluent and precise? I feel the language needs to be modified a bit. Both in terms of English grammar (double check usage of "the" and "comma"), and in terms of being scientifically more precise (e.g. using "pollution" instead of "emission"; or using "most influential input" instead of "most relevant"; or page 3 line 4; or page 3 line 26). I recommend a more careful review of the manuscript in this regard.*

Thank you for the feedback on the language of the manuscript. Based on this comment and comments made by other reviewers we plan to carefully review the language in a revised version of the manuscript.

*12. Are mathematical formulae, symbols, abbreviations, and units correctly defined and used? Yes.*

We appreciate your evaluation.

*13. Should any parts of the paper (text, formulae, figures, tables) be clarified, reduced, combined, or eliminated? Some of the quantile classes presented in Figure 2 in each signature measure can be removed.*

We outlined our thoughts on reducing the number of quantile classes in our reply on comment 4 a). Please see our reply above.

*14. Are the number and quality of references appropriate? Yes.*

We appreciate your evaluation.

*15. Is the amount and quality of supplementary material appropriate? Yes.*

We appreciate your evaluation.

References

Bieger, K., Hörmann, G., and Fohrer, N. (2013). The impact of land use change in the Xiangxi Catchment (China) on water balance and sediment transport, Regional Environmental Change, 15, 485–498.

Guse, B., Pfannerstill, M., and Fohrer, N. (2015). Dynamic Modelling of Land Use Change Impacts on Nitrate Loads in Rivers, Environmental Processes, 2, 575–592.

Mehdi, B., Lehner, B., Gombault, C., Michaud, A., Beaudin, I., Sottile, M.-F., and Blondlot, A. (2015a). Simulated impacts of climate change and agricultural land use change on surface water quality with and without adaptation management strategies, Agriculture, Ecosystems & Environment, 213, 47–60.

Mehdi, B., Ludwig, R., and Lehner, B. (2015b). Evaluating the impacts of climate change and crop land use change on streamflow, nitrates and phosphorus: A modeling study in Bavaria, Journal of Hydrology: Regional Studies, 4, 60–90.

Pfannerstill, M., Guse, B., and Fohrer, N. (2014). Smart low flow signature metrics for an improved overall performance evaluation of hydrological models, Journal of Hydrology, 510, 447–458.

Pianosi, F. and Wagener, T.(2015). A simple and efficient method for global sensitivity analysis based on cumulative distribution functions, Environmental Modelling & Software, 67, 1–11.

Sarrazin, F., Pianosi, F., and Wagener, T. (2016) Global Sensitivity Analysis of environmental models: Convergence and validation, Environmental Modelling & Software, 79, 135–152.

Sobol, I. M. (1993). Sensitivity analysis for nonlinear mathematical models, Mathematical Modelling and Computational Experiments, 4, 407–414.

---

## Author Comment (AC4) · 9 Nov 2018

**Reply to the reviewer comments RC4: 'Review of the manuscript by Schürz et al.' by**

**Francesca Pianosi**

*The manuscript presents an interesting application of uncertainty and sensitivity analysis to the SWAT model. The aim is to assess the dominant controls of long-term discharge and nitrate-nitrogen load predictions under climate and land use change, while also taking into account the intrinsic uncertainty in the model, i.e. parameter and setup uncertainty. The analysis is solid and provides interesting insights about the model behaviour. Although the specific findings are only relevant to the investigated model and case studies, their discussion is interesting for the wider community of SWAT users and in general users of environmental impacts assessment models, as it demonstrates the type of findings yielded by GSA and their implications for the refinement and use of the model. The visual analysis introduced in Figure 4-8 is a simple and yet effective complement to quantitative GSA approaches.*

*Overall the paper is well structured and well written, and I think it should be accepted for publication.*

*Below are some points that could be addressed to improve the manuscript clarity before publication.*

We would like to thank Francesca Pianosi for her constructive review and the valuable comments made. We appreciate the general positive feedback on the manuscript. It is a pleasure for us, that one of the developers of the PAWN sensitivity index evaluated this manuscript. Below, we addressed each comment made. We hope to clarify and discuss all points of concern sufficiently in the following document. The referee comments are printed in *serif, italic font*. Our replies to the comments are written in black, non serif font and our suggestions to revise the manuscript are highlighted with the colors blue for insertions and red for deletions.

Major comments

*[1] Language is at times unclear - some examples are given below as Minor points. I also have a general comment about the use of the term "sensitive". The authors use it as interchangeable with "influential" however I find this confusing, because "sensitivity" is an attribute of the output, not of the inputs. I would say that "input x1 is influential on the output" or "the output is sensitive to input x1" but I would not say that "input x1 is sensitive" - this is confusing. Some examples of these unclear occurrences are also given below under Minor points, however if the authors accept my remark they should check the entire manuscript.*

We highly appreciate this comment and agree that the example provided above identifies the correct use of the terminology. We accept the remark and will improve the updated version of the manuscript to correctly use the terms "sensitive" and "influental" accordingly.

*[2] The definition and use of the behavioural parameter sets is slightly unclear. I think the confusion started on P. 10 L. 6-7 with the sentence "For all SWAT model setups of the Schwechat and the Raab catchments we identified non-unique parameter sets that adequately simulated daily observation of discharge and NO3-N loads".*

*Does it mean that you identified one behavioural parameter set for **each** model setup, or that you identified one behavioural parameter set to be applied **in all** the setups? If the former, then how is the dependency between parameterisations and model setups accounted for in the GSA? If the latter, then the underlying assumption is that the same parameter values can effectively represent processes at different aggregation scales (ie for different definitions of the subbasins and HRUs)? This should be clarified. On a parallel note, I find it interesting that out of 100,000 sampled parameterisations only 43 and 52 where found behavioural. This is not uncommon in calibration of complex hydrological models but still worth highlighting. It would also be interesting to see whether these behavioural parameterisations are clustered in specific regions of the parameter space or if they are scattered across the sampled ranges, which would indicate a certain amount of interactions between the parameters. This could be illustrated for example through a parallel coordinate plot.*

We agree that the paragraph as stated leaves room for an ambiguous interpretation of the performed simulation and analysis steps. We designed the study the following way: For the different model setups of the Raab (6 different setups) and the Schwechat (4 different setups) we analyzed the model parameter sensitivities employing global sensitivity analysis (GSA). Thus, we performed six individual GSAs for the Raab catchment and four for the Schwechat catchment. The individual parameter sensitivity analyses resulted in the same sets of influential model parameters for the Raab catchment and the Schwechat catchment, respectively. As a consequence, we selected the same model parameters for all model setups of the Raab catchment and for all setups of the Schwechat catchment. For each case study, we drew 100 000 realizations of parameter combinations from the influential sets of model parameters. The simulations were performed with all model setups involving all drawn parameter combinations.

To answer your first question, a parameter set was eventually considered as a behavioral parameter set, if the simulations performed with **all** model setups involving that parameter set fulfilled the applied objective criteria stated on p.10L7-10.

This design decision was necessary, to treat the model setups and model parametrizations as individual inputs in the GSA (as you have indicated in the first part of your question). We agree with your comment that the selected layout implies the assumption that a parameter combination represents the analyzed processes at different aggregation scales. The individual models (i.e. a model setup with a specific spatial aggregation together with a selected model parametrization) were not analyzed and compared at the subasin or the HRU level. Yet, all models are capable of adequately simulating discharge and nitrate-nitrogen loads at the catchment outlets in the reference period. A drawback of this design decision is that it does not consider parameter combinations that that would result in satisfactory simulations when employed in one or several model setups but do not give good results with **all** model setups. Thus, the influence of the model setup and the model parametrization combined can be greater than the effect resulting from the illustrated study design. Nevertheless, we think that the presented results reveal relevant insights in their

current form, as they isolate the effect of the model setups that were calibrated for a reference period and are now applied for future changing conditions (the same applies to the parametrizations).

The selected experimental design is also a major reason for the low number of resulting behavioral parameter sets in the two case studies.

All of the explanations outlined above are in our opinion not sufficiently addressed in the current version of the manuscript. In particular, we think that the design decisions we made have to be conveyed and highlighted clearly in the methodology. Thus, we suggest to revise the concerning sections p.9L9 to p.11L2 accordingly.

According to the suggested visualization of the model parameters in a coordinate plot, we propose to add the figure below in the Appendix of the manuscript. We omitted any axes and tick labels due to very limited plotting space and the large number of parameters to visualize. The figure however illustrates any clustering and interaction of model parameters. We additionally suggest to add parameter ranges and the type of change of the model parameters in Table 3 on page 10.

[Figure]

*Catchment* ● Schwechat ● Raab

Figure caption:

Coordinate plot of the 43 and 52 behavioral SWAT model parameters that were used with the model setups of the Schwechat and the Raab respectively. Each panel illustrates the connection of two model parameters for the Schwechat in red (below the diagonal) and the Raab in blue (above the diagonal). The x and y axes of each panel show the range of the respective parameter plotted along the x or y dimension. The corresponding parameter ranges for all illustrated parameters are provided in Table XX (Reference to table below).

Table caption:
SWAT model parameters calibrated in the model setups of the Schwechat and the Raab catchments. The type of change indicates whether the model parameters were replaced by absolute values, modified by adding absolute values to the predefined model parameters or, changed by a relative fraction of the predefined model parameter. Illustrated are the initial ranges of the model parameters and the ranges of the final behavioral parameter sets of the model setups of the Schwechat and the Raab catchments.

| Parameter | Type of change | Initial parameter range | Parameter change range | |
| | | | Schwechat | Raab |
| --- | --- | --- | --- | --- |
| SFTMP | replace value | [-1.00, 1.00] | [-0.69, 0.93] | [-0.98, 0.88] |
| SNOCOVMX | replace value | [100.0, 500.0] | [0.9, 177.0] | [100.8, 447.5] |
| SNO50COV | replace value | [0.20, 0.50] | [0.21, 0.49] | |
| SURLAG | replace value | [0.00, 0.50] | [0.02, 0.99] | [0, 0.1] |
| GW_DELAY | replace value | [0.0, 300.0] | [5.5, 25.0] | [2.1, 283.3] |
| GW_REVAP | replace value | [0.02, 0.20] | [0.05, 0.15] | [0.02, 0.20] |
| GWQMN | replace value | [0, 3000] | [566, 2472] | [109, 2925] |
| RCHRG_DP | replace value | [0.00, 1.00] | [0.31, 0.69] | [0.13, 0.97] |
| SOL_K | relative change | [-0.90, 10.00] | [0, 0.97] | [-0.79, 9.76] |
| SOL_AWC | relative change | [-0.90, 2.00] | [-0.86, 1.49] | [0, 1.98] |
| SLSOIL | replace value | [0.0, 150.0] | [0.9, 27.6] | [14.7, 148.2] |
| CANMX | relative change | [0.00, 0.25] | [0.34, 2.40] | |
| ESCO | replace value | [0.00, 0.90] | [0.05, 0.90] | [0.05, 0.89] |
| LAT_TTIME | replace value | [0.0, 180.0] | [0.8, 6.8] | [5.5, 176.3] |
| OV_N | absolute change | [-0.09, 0.60] | | [0.07, 0.58] |
| CNOP_till | relative change | [-0.20, 0.10] | [-0.29, -0.06] | [-0.18, 0.01] |
| RCN | replace value | [2.00, 10.00] | [5.05, 9.97] | [2.3, 8.45] |
| NPERCO | replace value | [0.00, 1.00] | [0.24, 0.99] | [0.18, 0.7] |
| CDN | replace value | [0.00, 1.50] | [0.01, 1.44] | |
| SDNCO | replace value | [0.00, 0.50] | [0.02, 0.49] | |

*[3] GSA was applied using 7000 samples of the input factors. How was this number chosen? Did the authors checked the adequacy of this sample size? The fact that the ranking based on the sensitivity indices in Figure 2 is confirmed by the visual analysis of Figure 4-8 is reassuring, yet formal methods exist to assess the robustness of the GSA results to the chosen sample size (for example, using bootstrapping confidence intervals as in Sarrazin et al. 2016 or a dummy parameter as in Zadeh et al 2017, both cited in the manuscript). It would be good to include more discussion of this point in the manuscript.*

The number of input factor samples used in this study results from previous analyses performed using the present input factor data basis. A preceding analysis employed the Sobol method (Sobol, 1993) for global sensitivity analysis (GSA) using a sampling design proposed by Saltelli (2002) that requires $N(k+2)$ samples, where $N$ is the "base sample" (Saltelli, 2008) that was defined with 1000 in this study and k is the number of inputs (in this case 5).

Similar to the generic sampling strategy that you suggested in a presentation at the EGU 2018 (Pianosi and Wagener, 2018a, that is now published in Pianosi and Wagener, 2018), we utilized a model input sample that was initially drawn for a GSA applying a different method (in our case the Sobol method) and employed it to estimate PAWN indices.

Besides the PAWN Index that is presented in this manuscript, we also tested the Sobol method and a modified version of the STAR-VARS method (Razavi and Gupta, 2016a, 2016b) in the course of this work (results for the latter two analyses are not shown in the manuscript). All experiments expressing the maximum sensitivity of the used target variables to the analyzed input factors showed strongly overlapping results. Thus, we were confident regarding the soundness of the GSA results. We agree however that it would be beneficial to the reader of the manuscript to provide any measure of confidence with the results of the GSA. Thus we suggest to perform bootstrapping (as demonstrated in Sarrazin et al. (2016)) to provide confidence intervals together with the calculated PAWN indices in section 3.2. p.15ff.

*[4] The PAWN method was applied using a sampling scheme different from the one originally presented in Pianosi and Wagener (2015), in order to handle discrete-valued input factors. I understand the idea is to consider as fixed points $x_i^j$ all the possible values that the discrete input factor $x_i$ can take. Hence, for each input factor, the number of fixed points coincides with the number of possible values ($n_i$) that the input can take. If my interpretation is correct, then the text is misleading when it says (P. 13 L. 28) that "a generic random sample of the size $N$ was drawn and subsetted with $N/n_i$ subsets for all $x_i^j$" as the generic sample is divided into $n_i$ (and not $N/n_i$) subsets. Is this right?*

The term "generic" was used in the present context, as the sampling in all input factor dimensions was done randomly, although restricted by the number of fixed values each input can have. The separation of the total sample into $N/n_i$ subsamples is then a required step to calculate the Kolmogorov-Smirnov distance for the input factor $x_i$ at each location $x_i^j$. We understand however, that the term "generic sample" might be interpreted as a random sampling of continuous variables that is not the case here. Thus, instead of using the term "generic random sample" we suggest to specify the performed sampling in the following way (p.13L22-25):

 Pianosi and Wagener (2015) introduced the PAWN sensitivity method using a specifically tailored sampling design to infer the PAWN indices $T_i$ for continuous model inputs $x_i$. The  proposed sampling strategy suggests to draw $N_c$ conditional samples at $n$ randomly sampled points of each influencing variable $x_i$, where xi is fixed at a value $x_i = x_i^j$ while all others are perturbed. Recently, Pianosi and Wagener (2018) extended the applicability of the PAWN sensitivity method to estimate $T_i$ from a generic random sample of continuous model inputs. To approximate $T_i$ the generic sample $N$ is split into $n$ segments along each model input dimension resulting in conditional samples $N_c$ with an approximate size of $N/n$. We employed the proposed updated sampling strategy and adapted it for the use with discrete model inputs.  A sample of the size $N$ was

drawn. For each model input combination every model input was sampled randomly from its discrete realizations.  $N/n_i$  $x_i^j$  To infer  $KS_j(x_i)$ for all discrete values $x_i^j$ of a model input $x_i$ the sample $N$ was split into subsets for all $n_i$ discrete values, resulting in subsets of the size $N/n_i$ on average. It is important to consider, that the subset size depends on the number of discrete values $n_i$ of a model input $x_i$, while the subsets of the sampling scheme proposed by Pianosi and Wagener (2018) were on average $N/n$ for all model inputs $x_i$.

*Also, if I understand the strategy correctly, then the inputs with small number of possible values (for instance the land use scenario) are associated with conditional distributions based on a very large number of samples (around $N/n_i$ =7,500/2 in the case of land use scenarios), while the inputs with large number of possible values (for instance the parameterisation) are associated with conditional distributions based on much smaller number of samples (around 7,000/43). Do you think using such different sample sizes could have had an impact on the estimation of the KS values and hence of the PAWN sensitivity indices?*

Your assumption concerning the subset sizes is correct. We admit that the manuscript does not convey this information clearly. Thus, we suggest to update this section of the manuscript as proposed in the reply above. In the current version of the manuscript we did not analyze the effect of the strong differences in the subset sizes on the confidence intervals of the calculated sensitivities. We did however compare the results derived with the PAWN method to the results inferred from an adapted version of the STAR-VARS method (Razavi and Gupta; 2016a, 2016b) that was not affected by the different numbers of discrete values for each model input due to its sampling design. We observed only minor differences between these methods and hence assumed that the effect of the different subset sizes is low. We suggest however for the updated version of the manuscript to consider that point in the bootstrapping. If that assumption is correct, we expect that if the impact of the different subset sizes is low when the confidence intervals remain in a comparable range for different numbers of subsets of the individual model inputs.

*Finally, a new sampling strategy was recently proposed for PAWN (Pianosi and Wagener, 2018). While this new strategy is still designed for continuous inputs, and hence could not be used here, it would be good to mention its existence for readers who may want to apply PAWN in the future (as for the case of continuous inputs this would be recommended over the strategy in the 2015 paper).*

The publication will be considered in the updated version of the manuscript, as suggested in the updated section above.

*[5] I think the discussion in Section 4.2 is interesting but potentially slightly misleading. The authors clarify that "several assumptions were made in the development of scenarios that are highly subjective". I understand the importance of highlighting the subjectivity inherent in the scenario definition if the goal of this study was to make projections of the future evolution of the two catchments. However, this is not the objective when doing GSA. GSA answers the question: "how much output variation do we get if we vary the inputs within certain ranges?" The answer yielded by GSA (i.e. the sensitivity indices, the input ranking, etc.) is certainly conditioned upon the chosen ranges, however this is "intrinsic" to the question asked, regardless of how the choice is made - be it an "objective" calibration exercise (as done for the parameterisations) or a "storylines" approach. In other words, I think the point is to justify why certain scenarios are considered for the impacts assessment study; once they have been selected for that purpose, it follows that they would be used in the GSA too if one wants to know their relative influence with respect to other input factors of the model. So, I do not agree with the sentence (P. 26 L. 13-14) "For the SA of the simulated variables the diversity of the developed scenarios is essential.": diversity may be important for the impacts assessment (is it?) but not necessarily for the GSA. If a limited set of scenarios were selected for the impacts assessment, I would use that set for the GSA even if it is not diverse.*

We agree with the comment, that the goal of a GSA is to attribute the variations in simulated outputs to variations in model inputs, rather than simulating possible futures for a catchment. This was not the message we wanted to convey with this study. We intended to point out how GSA and an analysis of the uncertainty bands as illustrated can complement any impact study in understanding the sources of the uncertainties in simulating future conditions.

Maybe this is again, an issue of terminology. What we specifically wanted to address in this section was the fact that the subjective decisions we as modelers make in developing future scenarios will, no doubt, affect the simulation of a variable of interest. Further, when the developed discrete scenarios for a model input result in a wide range of a simulated output this will also affect the sensitivity of the output variable to the respective model input.

The analogue example for a single continuous model parameter would be to change the interval of that parameter in which it can vary for an assessment of the its influence on the model output. The selection of the parameter range is apparently highly subjective as well. Yet, while increasing the interval of a continuous property to cover more extreme regions of the model input space is a simple concept, the impact on the simulation of an output variable caused by any assumptions made in the development of model input scenarios is not always entirely clear in the scenario development. As this issue is not always addressed appropriately in environmental impact studies (e.g. by only using a few climate scenarios in an impact assessment), we saw a high need for this important discussion.

With the term "diverse" we wanted to express to represent a wide range of possible future representations of a model input. The addressed sentence seems however to be redundant as the following sentence repeats the argument. Thus, we suggest to change this section as follows:

  Scenarios must cover a broad range of possible futures...

Minor points

*P. 1 L. 15: "scenario inputs" should be "input scenarios"*

This will be changed accordingly in the updated version of the manuscript.

*P. 2 L. 5: "the precipitation of the climate scenarios" sounds a bit odd, maybe "precipitation projections"*

We agree with this comment and suggest to change the sentence as follows:
Additionally, the visual analysis of the uncertainty bands illustrated that the anomalies in precipitation of the different climate scenarios dominated the changes in simulation outputs, rather than changes while the differences in air temperature in both case studies showed no considerable impact.

*P. 3 L. 3-4: "An assessment is only as good as the dominant contributors of uncertainty in such a modeling chain." Unclear. Something seems to be missing in this sentence: an environmental impacts assessment is only good if dominant contributors of uncertainty are... what? Identified? removed? ...?*

The sentence actually does not contribute much information. Thus, we rather suggest to delete it in the updated version of the manuscript.

*P. 3 L. 11-12: "model computations" should be "model evaluations" (or "runs" or "executions")*

The phrase will be changed to: ... from certain a number of model computations evaluations.

*P. 3 L. 19: "Most applications utilize GSA to identify and rank continuous model parameters". Unclear: GSA does not "identify parameters" at most "identify influential parameters"*

Will be changed to:
Most applications utilize GSA to identify influential model parameters and to rank continuous model parameters according to their influence on model outputs. Model parameters are usually continuous model inputs.

*P. 3 L. 21: "Although," Comma should be removed*

This will be changed accordingly.

*P. 3 L. 26-27: "An OAT analysis however presumes linear models and non-correlated inputs". Not sure OAT requires a linear model, for instance the Morris method uses a OAT approach and yet is typically applied to nonlinear models. More generally, why should GSA be applied to a linear model at all? If the model is linear than the effect of each input on the model output is simply proportional to the input variation, no need to do GSA to know that.*

In this section we highlighted the equivalency of the standard procedure performed in impact assessments an **local** "one-at-a-time" (OAT) analyses. We agree that in this specific sentence we used the more general term "OAT" instead of referring specifically to "local OAT". The presumptions of OAT such as linearity of the model or independence of model inputs was also addressed by Baroni and Tarantola (2014) or Saltelli (2010). To infer the (global) sensitivity of a model output from a delta change of a model input presumes that the

same delta change of the model input at another position in the input space has the same effect on the model output and is not influenced by any other model input (linearity and independence). We further did not suggest to apply sensitivity analysis to a model where a linear relationship between the inputs and outputs is a-priori known. Contrary to that, we state that applying an OAT analysis to infer sensitivities of model outputs implies model linearity and the independence of model inputs.

Concerning terminology we referred to the terms as they are used in Saltelli and Annoni (2010), where OAT was considered to be performed from the same "nominal point" and the analogous analysis performed at various points in the input space was termed "radial" elementary effects (EE). We do however agree that methods such as the Morris method or EE also employ OAT sampling designs while inferring global estimates of the output sensitivities.

Thus, we suggest to modify the commented sentence and add "local" to specifically address the issues with local OAT analysis.

*P. 4 L. 4: "complex". Unclear. What is the definition of a "complex" input?*

The term "complex" was already used earlier in the manuscript (e.g. p.3L21, p.3L30). Yet, we do not provide an explanation of that term at any point in the manuscript. Further, the term "complex" apparently does not clearly convey what is meant here, where maybe "composite" might be a more precise term to use. Thus we suggest to change the term "complex" to "composite" and further add examples in p.3L21:

Although, it is possible to implement  composite model inputs (e.g. climate scenarios that affect several climate variables at the same time, or land use scenarios that can impact the model setup) in GSA...

*P. 4 L. 5: "No study is known to us that takes advantage of GSA in the scope of environmental impact studies." What is the definition of "environmental impact studies" here? I would say that GSA has been applied to such studies before, e.g. Anderson et al (2014); Butler et al (2014); Le Cozannet et al (2015)*

We agree that if you consider "environmental impact studies" in their actual broad context that GSA has been applied in several studies. Thus, the sentence is misleading and will be deleted. The publications mentioned here as examples should rather be acknowledged and mentioned in the introduction. We will adapt the introduction accordingly

*P. 4 L. 13-16: Very long sentence, consider splitting into two.*

Based on the GSA and the visual analysis of the simulated uncertainties we are able to draw conclusion on the simulation of discharge and $NO_3^- - N$ loads as impacted by the model setup, model parametrization and the future scenarios of land use, point source emissions and climate. These conclusions are of course limited to assumptions made in the model setup and in the development of the scenarios.

*P. 8 L. 3: "Although," Comma should be removed.*

This will be changed accordingly.

*P. 8 L. 14-15 "The SWAT model setups for the Raab and the Schwechat involved decisions for the selected number of subbasins of a model setup and the definition of the HRUs." Convoluted sentence.*

This section will be modified as follows:

A SWAT model setup requires the modeler to determine an "appropriate" number of subbasins and to make decisions for the HRUs (such as eliminating "insignificantly" small HRUs from the setup). The SWAT model setups for the for the Raab and for the Schwechat had different numbers of subbasins and defined HRU differently.

*P. 8 L. 15: "Both modifications": which modifications? Unclear*

Please see the changes suggested in the comment above.

*P. 9 L. 2: "involving". Unclear. Maybe "which requires"?*

The text will be changed accordingly.

*P. 9 L. 11: "to define of the thresholds". Remove "of"? In general, the entire sentence is a bit unclear. How is the "aggregation error" defined? Error in which variable, and with respect to what "correct" value?*

Thank you for finding the typo. As this section requires further explanations this section will be modified as follows:

In total, we set up four SWAT models, two with 3 and two with 14 subbasins for the Schwechat catchment and six setups for the Raab catchments with two each of 4, 29, and 54 subbasins. We kept the resulting HRUs of full HRU setups unmodified. The numbers of HRUs in the reduced HRU setups were modified by applying thresholds for land use, soil, and slope classes. HRUs with an area below the defined thresholds were eliminated from a model setup.  We employed the R package topHRU (Strauch et al., 2016) to  determine optimum thresholds for land use, soil, and slope classes  that result in a maximum aggregation error of 5% of the total area of the HRUs when comparing the changes of land use, soil, and slope classes of the full HRU setup and the reduced HRU setup with the same numbers of subbasins. Table 2 gives an overview of the final baseline model setups for both case studies.

*P. 9 L. 14: "In a pre-analysis step," In the GSA literature, this kind of "pre-analysis step" is often called a "screening" analysis, as it aims at screening out the non-influential parameters. Maybe worth mentioning the term as it would be familiar to many readers.*

We appreciate the suggestion. The text will be changed accordingly.

*P. 9 L. 14: "relevant parameters". Relevant to what? Maybe better "influential"*

The term will be changed accordingly.

*P. 9 L. 21: "FDCs". Explain the acronym*

The acronym "FDC" was introduced on p.4L11:
...as well as flow duration curves (FDCs) of daily discharge and daily $NO_3^- - N$ loads...

*P. 13 L. 5: "To identify the impact of" maybe better "To measure the relative importance of"*

We prefer your suggestion. Thus, we will implement it accordingly.

*P. 13 L. 7: "PAWN involves". Unclear what "involves mean. Maybe better "PAWN uses"*

The text will be changed to PAWN employs...

*P. 13 L. 11: "the sensitivity of a model input x for a target variable y". Sensitivity is an attribute of the output, not the input. I would rephrase as "the sensitivity of a target variable y to a model input x".*

As mentioned in our reply to the major comment [1] the entire manuscript will be modified to meet this suggestion. Thus, we also implement this suggestion.

*P. 13 Eq. (1) and (2). The mathematical notation could be made clearer. I find it odd that in Eq. (1) $KS$ takes as subscript the index of the fixed point ( $j$ ) while its argument remains the generic input $x_i$. This choice also makes it more difficult to understand how maximisation occurs in Eq. (2). I think using the notation $KS(x_i^j)$ in both equations would make things much clearer.*

We appreciate this comment and will change the equations accordingly.

*P. 13 L. 23: "possible states". Why "states"? The term was never used with this meaning before. I would rather say "possible values".*

This is a remnant of a previous version of the manuscript and will be changed accordingly.

*P. 13 L. 24: "a lower sensitivity of the input xi on the target variable y". Again, rephrase as either "a lower sensitivity of the target variable y to the input $x_i$ " or "a lower influence of the input $x_i$ on the target variable $y$ ".*

This suggestion will be implemented.

*P. 13 L. 28: "subsetted with" Not sure "subset" can be used as a verb. Maybe better "divided into"*

Your statement is correct. The verb "subset" is not listed in any dictionary. Thus, we will rephrase it as suggested.

*P. 13 L. 29. "... were used for the sensitivity assessment". I would link this to the mathematical notation just introduced above and say: "... were used as target variable y".*

We appreciate the suggestion and will implement it accordingly.

*P. 14 L. 10-12: "In this study, we consider all execute model setups to be plausible..." I do not understand this clarification. What other approach would have been possible? To discard some simulations because deemed not plausible? And how would you define then what is plausible and what is not? Please clarify.*

We agree that the phrasing sounds odd. Thus, this sentence will be rephrased to:

 All executed model setups  represent plausible realizations of the future conditions in both catchments to simulate future discharge and $NO_3^- - N$ loads.

*P. 14 L. 17: "low number of each input". Unclear. Do the authors mean "low number of inputs" (i.e. 5 inputs) or "low number of possible values taken by each input"?*

Here we meant the latter. We will update that phrase to:

 The low number of possible values taken by each input allowed...

*P. 14 L. 24-28. This sounds like a repetition of what just said in the methodology section, I do not think is needed. I would rather use this opportunity to explain how to read*

*Figure 2 (what is the difference between the panels and how to read each circle plot).*

We will consider to replace this section with an explanation of Figure 2 in the updated version of the manuscript. As further modifications will be added to the figure (e.g. confidence intervals, etc.) we do not suggest modifications to the text at this point.

*P. 15 L. 17: "highly sensitive" replace by "highly influential"*

This will be changed accordingly in the revised manuscript.

*P. 15 L. 25-26: "their overall sensitivities follow the general trend of the climate scenarios to a large extent". Unclear, please rephrase.*

We will rephrase this section as follows:

 The model setups yielded insignificantly low PAWN indices for the majority of signature measures with values below 0.1 in the Raab case study, indicating that the model setup  had a low influence on most analyzed processes.  The pattern of the resulting PAWN indices of the model setups closely follows however the pattern of the PAWN indices that were calculated for the climate scenarios.

*P. 15 L. 33: "difference that is visible for the two". Unclear, do you mean "difference that is visible between the two"?*

This sentence will be rephrased to:

 A notable difference between the two case studies is how  the simulations of long term monthly discharges and $NO_3^- - N$ loads in the reference period compare to the ranges of future simulations.

*P. 15 L. 33-34: "how the reference period relates to the uncertainty bands in amplitude". Unclear what this means.*

See our suggestion for the modification of that phrase above.

*P. 16, caption of Fig. 2: "Model input sensitivities for signature measures". Replace by "sensitivities of signature measures to model inputs". And later on "sensitivities of" should be replaced by "sensitivities to"*

The caption will be changed accordingly in the updated version of the manuscript.

*P. 17, text and Figure 3: what does "specific discharge" mean? Why "specific"?*

I am not sure how well established this term is used in the hydrologic community. The specific discharge relates a discharge (given in e.g. $m^3s^{-1}$) to the catchment area that produces the runoff and sums it for a specific time interval (in this case on monthly basis). We decided to use the specific discharge for a better comparison of the two catchments with substantially different catchment sizes.

*P. 17 L. 6: "show a difference". Does this mean "show an increase"? If so, I would use "increase", it makes it easier for the reader to follow.*

We prefer your suggestion. This will be changed accordingly in the updated manuscript.

*P. 19 L. 12: "While a grouping...". Remove "While".*

We think that "while" is essential for the meaning of the sentence. To consider other reviewer comments the section will be modified as follows:
While a grouping of the individual climate scenarios with respect to their temperature anomalies shows a more indefinite picture., Aall climate scenarios simulated an increase in temperature.

*P. 24 L. 20: caused future land use change" Maybe "caused **by** future..." ?*

This will be changed in the updated version of the manuscript.

*P. 26 L. 31-32: "The application of sampling strategies for SA usually do not account for the circumstances that one model input constrains any other model input". I do not fully agree. There is an increasing literature on GSA methods applicable to the case of dependent inputs, see for instance Mara and Tarantola (2012; 2017).*

We appreciate the comment and I think it is worth to mention and acknowledge these publications. While it is a more straight forward approach to constrain a continuous property by another continuous property it might be not a straight forward procedure to identify all plausible scenario combinations for multiple model inputs (e.g. some future climate settings might make future agricultural practices implemented in a land use scenario impossible). In the context of the present work we were referring to the latter case. We suggest to clarify this in the manuscript and to acknowledge the substantial work that was done to constrain dependent continuous variables in GSA.

*P. 27 L. 17-18: "by a factor of up to 5... up to 8". Do these numbers come out of a comparison of PAWN indices? If so, I am not sure I would draw such quantitative comparison. PAWN indices are (maximum) KS values: what is the practical interpretation of "a factor of 5" between KS values? I find it difficult to imagine.*

We appreciate your feedback on that section. As a consequence we will remove that comparison in the updated version of the manuscript.

*P. 28 L. 3: "the lack of tool that allow the practitioners access to such methods". Not sure I understand what the authors mean here. Several GSA software tools are available (some are reviewed for example in Pianosi et al 2015). So what is the problem here? That they are not "friendly" enough for practitioners to use them? Or that they are not sufficiently tailored to SWAT applications? Pls clarify.*

We agree that the sentence is too vague. Software, toolboxes and libraries to perform GSA are available for many different programming languages, for instance the SAFE toolbox (Pianosi et al., 2015) for matlab, SPOTPY (Houska et al., 2015) for python, or R packages such as sensitivity (Iooss et al., 2015), or fast (Reusser, 2015).

From a practitioner's perspective the challenge is to assemble such a large number of models and to perform thousands of model simulations for a large number of model input combinations, instead of performing the status quo procedure of implementing single scenarios into a calibrated model. To generalize such analysis for the application in environmental impact studies we suggest to come up with frameworks that support the practitioner in this laborious working steps of a case study.

Thus we suggest to specify the section p.28L2-4. We suggest to mention the tool boxes, packages, etc. that are available to perform GSA, but highlight the challenge of of assembling, executing and evaluating such a large number of model setups instead.

**References**

*Anderson, B., Borgonovo, E., Galeotti, M., Roson, R., 2014. Uncertainty in climate change modeling: can global sensitivity analysis be of help? Risk Anal. 34 (2).*

*Butler, M.P., Reed, P.M., Fisher-Vanden, K., Keller, K., Wagener, T., 2014. Identifying parametric controls and dependencies in integrated assessment models using global sensitivity analysis. Environ. Model. Softw. 59.*

*Le Cozannet et al (2015) Evaluating uncertainties of future marine flooding occurrence as sea-level rises, Environmental Modelling Software, 73.*

*Mara and Tarantola, 2012, Variance-based sensitivity indices for models with dependent inputs, Reliability Engineering and System Safety*

*Tarantola and Mara, 2017, VARIANCE-BASED SENSITIVITY INDICES OF COMPUTER MODELS WITH DEPENDENT INPUTS: THE FOURIER AMPLITUDE SENSITIVITY TEST, International Journal for Uncertainty Quantification*

*Pianosi and Wagener (2018), Distribution-based sensitivity analysis from a generic input-output sample, Environmental Modelling Software*

*Pianosi, F., Sarrazin, F., Wagener, T. (2015), A Matlab toolbox for Global Sensitivity*

Baroni, G. and Tarantola, S. (2014). A General Probabilistic Framework for uncertainty and global sensitivity analysis of deterministic models: A hydrological case study, Environmental Modelling & Software, 51, 26–34.

Houska, T., Kraft, P., Chamorro-Chavez, A., Breuer, L. (2015). SPOTting model parameters using a ready-made python package. PloS one, 10(12), e0145180.

Iooss, B., , Janon, A., Pujol, G., (with contributions from Boumhaout, K., Da Veiga, S., Delage, T., Fruth, J., Gilquin, L., Guillaume, J., Le Gratiet, L., Lemaitre, p., Nelson, B., Monari, F., Oomen, R., Rakovec, O., Ramos, B., Roustant, O., Song, E., Staum, J., Sueur, R., Touati T., and Weber, F. (2018). sensitivity: Global Sensitivity Analysis of Model Outputs. R package version 1.15.2. https://CRAN.R-project.org/package=sensitivity

Pianosi, F. and Wagener T. (2018b). A new implementation of the PAWN method to perform density-based sensitivity analysis from a generic sample, EGU General Assembly 2018, Geophysical Research Abstracts Vol. 20, EGU2018-10784.

Razavi, S. and Gupta, H. V. (2016a). A new framework for comprehensive, robust, and efficient global sensitivity analysis: 1. Theory, Water Resources Research, 52, 423–439.

Razavi, S. and Gupta, H. V. (2016b). A new framework for comprehensive, robust, and efficient global sensitivity analysis: 2. Application, Water Resources Research, 52, 440–455.

Reusser, D. (2015). fast: Implementation of the Fourier Amplitude Sensitivity Test (FAST). R package version 0.64. https://CRAN.R-project.org/package=fast

Saltelli, A. (2002). Making best use of model valuations to compute sensitivity indices. Computer Physics Communications 145, 280–297.

Saltelli, A., Ratto, M., Andres, T., Campolongo, F., Cariboni, J., Gatelli, D., Saisana, M., and Tarantola, S. (2008). Global Sensitivity Analysis. The Primer, John Wiley & Sons, Ltd, Chichester, UK.

Saltelli, A. and Annoni, P. (2010). How to avoid a perfunctory sensitivity analysis, Environmental Modelling & Software, 25, 1508–1517

Sarrazin, F., Pianosi, F., and Wagener, T. (2016) Global Sensitivity Analysis of environmental models: Convergence and validation, Environmental Modelling & Software, 79, 135–152.

Sobol, I. M. (1993). Sensitivity analysis for nonlinear mathematical models, Mathematical Modelling and Computational Experiments, 4, 407–414.

---

## Editor Decision (ED1)

Hess-2018-375 – Revised version

Evaluation of the revised version

Dear Dr. C. Schürz

Thank you for the revised version of the manuscript and the effort you have put into it. In general, I consider the questions to be properly addressed. Nevertheless, some aspects call for clarification before I can accept the manuscript for publication.

Below I list a number of questions related to your response to the reviewers:

- Fig. A3: You mention in the response that the figure shall show any clustering of model parameters and possible parameter interactions. This figure is indeed very valuable for that purpose and illustrates a few interesting patterns. For example:
    o There is one case of a strong correlation between two parameters (N concentration in rainfall, N percolation coefficient in the Raab catchment).
    o some parameters were not relevant in one of the catchments (e.g., denitrification rate in the Raab catchment).
    o Some parameters have strongly bi-modal distributions (e.g., available water capacity in soils in the Raab catchment.
    o Some parameters have very different distributions for the two catchments (e.g., subsoil hillslope length).
    However, you do not discuss any of these aspects (why to keep highly correlated parameters for example).

    Can you please comment on
        o Whether these observations are plausible (why is denitrification important in the Schwechat but not in the Raab catchment)?
        o How these patterns may have affected the outcome (how important for example are the three SLSSOIL parameter values for the resulting overall uncertainty)?

- Reviewer 3 commented on some apparently counter-intuitive results such as the limited effect of land use change on the N concentrations. You argue that this is supported by findings in the literature and you also provide evidence that it was not an artifact of the number of realizations of the respective input/setup. While I agree with your arguments I still think it was worth mentioning that land use change might cause much larger effects on nitrate levels for example in situations where larger fractions of the catchments were affected or changes were more dramatic (see e.g., Honti et al. (2017) as a quick example to illustrate the point (not because I think you have to cite it)). I suggest you clarify this aspect more clearly.

- Reviewer 4 asked for clarifications about the GSA sample size (N = 7000). Your explanation seems clear to me except for the determination of the base sample $N_{base}$ = 1000. Was this an arbitrary decision or was there a further argument behind? Please clarify.

Below I list a number of questions related to revised manuscript:

p. 2, L. 1: I suggest to skip "proved to" (how did you proof it?). I think the statement "We present approaches for the visualization of the simulation uncertainties that support the diagnosis … " is clear enough.

p. 2, L. 7: What do you mean by "anomalies"? Please clarify.

p.7, L. 18: "where" should be corrected to "were", I assume.

p. 8, L. 4: space missing in front of parenthesis.

p. 9, L. 21: Probably, one should insert "maximum" before "5%".

p. 9, L. 22: How were the 42 parameters selected? If you argue that these parameters are frequently used for calibration, can you support this by a reference? Which fraction of the total number of parameters is covered by them?

p. 10, L. 5: It would be also informative to list those parameters that were not influential.

p. 10, L. 20: How can one calculate the target variables using the Nash Sutcliffe criterion? Please clarify.

p. 11, L. 19: What are these observable trends? Based on which data, references? Please clarify.

p. 12, L. 28: The N = 7000 represents almost half of all combinations (46%) for the Schwechat catchment (according to Tab. 3) and 13% for the Raab watershed, is this correct? Perhaps this information might be useful.

p. 12, L. 29: It is not clear what you mean "quasi random sampling".

p. 13, L. 10: The target variable should be "y", I assume.

p. 14, L: 9 – 10: What is meant by the "generic random sample"?

p. 15, L. 10: "Subsetting" isn't a verb, I assume.

p. 15, L. 12: "temperature or precipitation anomalies": how did you define these terms?

p. 19, L. 11 – 13: "In comparison to the reference period (dashed line), wetter future climate scenarios (blue) simulated larger discharge and NO3 -N loads, while dryer future conditions lead to a drastic reduction in discharge and NO3 -N loads." With these changes, what are the implications for the N-balances: will N accumulate under drier conditions? Would one not expect feedback mechanisms to get activated? Can you comment on that?

Fig. A2: A log-scale for the y-axes would allow for a much better comparison of the observations and the simulations.

Please respond to these comments.

Sincerely

Dr. Christian Stamm
Editor HESS

**References:**

Honti, M., J. Rieckermann, N. Schuwirth and C. Stamm (2017). "Can integrative catchment management mitigate future water quality issues caused by climate change and socio-economic development?" Hydrological and Earth System Sciences **21**: 1593–1609.

---

## Author Response (AR2)

**Reply to the editor comments: 'Hess-2018-375 - Revised version'**

*February 8, 2019*

Dear Dr. Christian Stamm,

we would like to thank you for the positive feedback and your very detailed comments on our revisions of the manuscript hess-2018-375,. In the following we will address the points you raised that require further clarification. Below are your comments printed in *serif, italic font*. We added our replies in black, non serif font and highlighted our changes in the manuscript according to a comment with the colors blue for insertions and red for deletions.
We hope the revision of the manuscript considers all of your comments. If there are any further questions or issues, please contact me and we will try to clear them as soon as possible.

Sincerely,
Christoph Schürz

*Below I list a number of questions related to your response to the reviewers:*

- *Fig. A3: You mention in the response that the figure shall show any clustering of model parameters and possible parameter interactions. This figure is indeed very valuable for that purpose and illustrates a few interesting patterns. For example:*
  - *There is one case of a strong correlation between two parameters (N concentration in rainfall, N percolation coefficient in the Raab catchment).*
  - *some parameters were not relevant in one of the catchments (e.g., denitrification rate in the Raab catchment). Some parameters have strongly bi‑modal distributions (e.g., available water capacity in soils in the Raab catchment.*
  - *Some parameters have very different distributions for the two catchments (e.g., subsoil hillslope length).*

  *However, you do not discuss any of these aspects (why to keep highly correlated parameters for example).*

  *Can you please comment on*

  - *Whether these observations are plausible (why is denitrification important in the Schwechat but not in the Raab catchment)?*

  - *How these patterns may have affected the outcome (how important for example are the three SLSSOIL parameter values for the resulting overall uncertainty)?*

We agree that we do not provide any discussion for the patterns that are visible in the parameter cross plot that we added in the revised version of the manuscript. Thus, we added an interpretation of the model parametrizations at two points in the manuscript. (1) We added a brief discussion for the differences of the model parameter sensitivities in section p.10L32ff. (2) We also added a discussion on the parameter correlations and clustering in section p.11L14ff:

(1):
The majority of parameters were identified as influential parameters in the Schwechat and the Raab case study. The parameters SNO50COV, CANMX, CDN, and SDNCO were only relevant for the model setups in the Schwechat and the parameter OV_N was only influential for in the Raab. For the majority of these parameters it is a matter of the selected threshold that defines a parameter to be influential or not. The most dominant parameters were however identified as highly relevant in both case studies.

(2):
The majority of parameters are scattered randomly and do not show any clustering or interaction with other parameters. The parameters RCN and NPERCO in the Schwechat catchment show a clear inverse relationship. This implies that the parameters compensate each other in the behavioral model setups. This finding seems plausible for the Schwechat catchment where the NO3-N transport into the receiving waters is strongly groundwater driven and a surplus of NO3-N input is reduced by a decrease in NO3-N percolation. The parameters SLSOIL, SURLAG, and SOL_AWC show a clear bimodal pattern for the Raab

catchment. The bimodal patterns of these parameters are strongly related and a compensation effect between these parameters is visible. Model setups with increased slope values (SLSOIL) and longer lag-times of the surface runoff (SURLAG) together with an increased soil available water content (SOL_AWC) resulted in behavioral model and were able to reproduce historic discharge and NO3-N records, similar to the model setups where such clear relationship is not visible.

- *Reviewer 3 commented on some apparently counter‑intuitive results such as the limited effect of land use change on the N concentrations. You argue that this is supported by findings in the literature and you also provide evidence that it was not an artifact of the number of realizations of the respective input/setup. While I agree with your arguments I still think it was worth mentioning that land use change might cause much larger effects on nitrate levels for example in situations where larger fractions of the catchments were affected or changes were more dramatic (see e.g., Honti et al. (2017) as a quick example to illustrate the point (not because I think you have to cite it)). I suggest you clarify this aspect more clearly.*

We agree that we treated this point too much from one side and focused on literature that mainly supports our findings. To balance the discussion of land use change here, we added arguments that should emphasize the fact that land use change of course can be relevant, depending on the defined scenario boundaries. The paragraph on p.24L33ff was changed the following:

...Mehdi et al. (2015)  however found that including  agricultural land use change into the impact assessment of a southern German watershed strongly increased the NO3-N and total phosphorus loads. Teshager et al. (2016) support the findings of Mehdi et al. (2015) and also found that corn intensive scenarios lead to an increase in discharge and significant water quality problems while an extensive scenario where mainly switchgrass is planted lead to water quality improvements under future climate change.  Consequently, the low impact of land use change found in the present study seems reasonable with respect to other literature, particularly as no extreme scenarios were implemented. This does however not generally imply a low importance of land use change in environmental impact assessments. Land use change or changes in the management can be the most relevant input, particularly when strong future changes, such as possible bans of emittents are considered (Honti et al.; 2017).

- *Reviewer 4 asked for clarifications about the GSA sample size (N = 7000). Your explanation seems clear to me except for the determination of the base sample $N_{base}$ = 1000. Was this an arbitrary decision or was there a further argument behind? Please clarify.*

Thank you for that comment. The decision we made here was not arbitrary, but was made based on examples given by Saltelli et al. (2008) in practical applications. Although, the selected base sample numbers strongly vary in literature (see e.g. Sarrazin et al. (2016) for a review) we selected this number of base samples in previous experiments.
In the manuscript we modified the section p.12L29-32 and added the reference:

The number of combinations results from previous experiments that applying the SA method of Sobol (results not shown) using the sampling strategy proposed by Saltelli and Tarantola

(2002).  A base sample size of $N_b$ = *1000* was used to meet the suggestions shown in Saltelli et al. (2008).  Thus, the total sample size of 7000 is defined as $N = N_b(k+2)$, where $k$ is the number of model inputs *(k = 5)*. . Although Sarrazin et al. (2016) report publications that required substantially larger base sample sizes (e.g. $N_b$ = *12000* in Nossent et al. (2011), or $N_b$ = 8192 in Tang et al. (2007)) for convergence of the ranking of influential continuous model parameters, a sample size of 7000 includes 46% and 12% of all possible model input combinations in the Raab and the Schwechat case studies, respectively.

*Below I list a number of questions related to revised manuscript:*

*p. 2, L. 1: I suggest to skip "proved to" (how did you proof it?). I think the statement "We present approaches for the visualization of the simulation uncertainties that support the diagnosis … " is clear enough.*

The change was done accordingly. A similar phrasing in the conclusions (p.29L2-3) was also changed accordingly.

*p. 2, L. 7: What do you mean by "anomalies"? Please clarify.*

Apparently this phrase is more common in the German language. Hence, the term "anomalies" was rephrased to "deviations to historic records" in the entire manuscript (p.2L2, p.15L12, p.19L10, p.19L20, and captions Fig.5 and Fig.6)

*p.7, L. 18: "where" should be corrected to "were", I assume.*

The change was done accordingly.

*p. 8, L. 4: space missing in front of parenthesis.*

The change was done accordingly.

*p. 9, L. 21: Probably, one should insert "maximum" before "5%".*

The change was done accordingly.

*p. 9, L. 22: How were the 42 parameters selected? If you argue that these parameters are frequently used for calibration, can you support this by a reference? Which fraction of the total number of parameters is covered by them?*

We agree that the statement in its current form is sound enough. We added literature to support the parameter selection:

Initially, 42 model parameters were selected that are frequently calibrated in SWAT model setups to simulate discharge and NO3-N loads (see e.g. Arnold et al. (2012) and Abbaspour et al. (2007) for a general overview of relevant SWAT model parameters, Mehdi et al. (2018) and Haas et al. (2016) for parameters controlling the water balance and nutrient cycles, or Haas et al. (2015) for a review on the dominant nitrogen parameters).

*p. 10, L. 5: It would be also informative to list those parameters that were not influential.*

We included all non-influential parameters in Table A1.

*p. 10, L. 20: How can one calculate the target variables using the Nash Sutcliffe criterion? Please clarify.*

We agree that the phrasing sounds odd. We modified the section as follows:

The Nash Sutcliffe Efficiency criterion (NSE, Nash and Sutcliffe, 1970), the Kling Gupta Efficiency criterion (KGE), including its three components (Gupta et al., 2009), and a refined version of the Index of Agreement (Willmott et al., 2012) were used to evaluate the simulated time series of daily mean discharge and daily sums NO3 -N loads. Additionally, we applied the ratio of the root mean square error and standard deviation (RSR, (Moriasi et al., 2007)) to evaluate different segments of the FDCs of daily discharge and daily NO3-N load simulations (Pfannerstill et al., 2014; Haas et al., 2016). All calculated criteria were included in the parameter sensitivity analysis as target variables. A model parameter was considered to be sensitive if it showed a relative sensitivity of 10% compared to the most sensitive parameter with respect to a specific objective criterion for at least one of the employed objective criteria.

*p. 11, L. 19: What are these observable trends? Based on which data, references? Please clarify.*

We specified the sentences the following way:
A "business-as-usual" scenario extrapolates the  trends that we determined for the dominant crops in the time period 1970 - 2010 (Statistik Austria, 2015b)  to the future (2071 to 2100), while a second "extensive" scenario assumes an extensification of agricultural activities and other intensive land uses in both catchments (Table A5).

*p. 12, L. 28: The N = 7000 represents almost half of all combinations (46%) for the Schwechat catchment (according to Tab. 3) and 13% for the Raab watershed, is this correct? Perhaps this information might be useful.*

We highly appreciate this comment. We think this information is very valuable. Particularly, when considering the comment of Reviewer 4 (and your additions to that comment). We included this information. Therefore, see text modifications in the reply to the comment of reviewer 4.

*p. 12, L. 29: It is not clear what you mean "quasi random sampling".*

Quasi-random sampling is a group of sampling strategies that cover the input space more evenly compared to pure random sampling. Saltelli (2002; 2008) suggest to use quasi-random sampling for the implementation of variance-based sensitivity analysis. Thus, we used such sampling strategy in our applications as well. In the text we now refer to Saltelli (2008).

*p. 13, L. 10: The target variable should be "y", I assume.*

Thank you for identifying that typo. This will be changed accordingly.

*p. 14, L: 9 – 10: What is meant by the "generic random sample"?*

The term was adopted from Pianosi and Wagener (2018). "Generic" here expresses that the group of random sampling strategies is meant in general. This term is common in parameter sampling and we would prefer to keep that terminology here.

*p. 15, L. 10: "Subsetting" isn't a verb, I assume.*

We agree with that. The term will be replaced by "grouping".

*p. 15, L. 12: "temperature or precipitation anomalies": how did you define these terms?*

Please see the reply to the comment on p.L7 above.

*p. 19, L. 11 – 13: "In comparison to the reference period (dashed line), wetter future climate scenarios (blue) simulated larger discharge and NO3 -N loads, while dryer future conditions lead to a drastic reduction in discharge and NO3 -N loads." With these changes, what are the implications for the N‑balances: will N accumulate under drier conditions? Would one not expect feedback mechanisms to get activated? Can you comment on that?*

We tried to specify the text and added more detailed results that should reflect the differences in the dynamics in the NO3-N cycle between dry and wet climate scenarios. Therefore we added the following text on p.19L13ff:

...These findings further imply that NO3-N applied in fertilizers will remain in the upper soil layers and be transformed (mineralized or immobilized or denitrified) instead of being transported to the receiving waters. A comparison of the NO3-N budgets of simulations with dry and wet climate scenarios for the Raab shows a difference of up to +27% of NO3-N accumulated in the soil, as well as a decrease of 43% and 38% in NO3-N yield in the fast and slow runoff, respectively.

*Fig. A2: A log‑scale for the y‑axes would allow for a much better comparison of the observations and the simulations.*

During the compilation of the manuscript (and the first revisions) we also tried to plot the time series on a logarithmic scale. We concluded then, that the logarithmic scale did not improve the interpretability very much.

[revised manuscript text omitted]